## OPEN

# New insights into the genetic etiology of Alzheimer's disease and related dementias

Characterization of the genetic landscape of Alzheimer's disease (AD) and related dementias (ADD) provides a unique opportunity for a better understanding of the associated pathophysiological processes. We performed a two-stage genome-wide association study totaling 111,326 clinically diagnosed/'proxy' AD cases and 677,663 controls. We found 75 risk loci, of which 42 were new at the time of analysis. Pathway enrichment analyses confirmed the involvement of amyloid/tau pathways and highlighted microglia implication. Gene prioritization in the new loci identified 31 genes that were suggestive of new genetically associated processes, including the tumor necrosis factor alpha pathway through the linear ubiquitin chain assembly complex. We also built a new genetic risk score associated with the risk of future AD/dementia or progression from mild cognitive impairment to AD/dementia. The improvement in prediction led to a 1.6- to 1.9-fold increase in AD risk from the lowest to the highest decile, in addition to effects of age and the *APOE* ε4 allele.

A D is the most common form of dementia. The heritability is high, estimated to be between 60% and 80%[1]. This strong genetic component provides an opportunity to determine the pathophysiological processes in AD and to identify new biological features, new prognostic/diagnostic markers and new therapeutic targets through translational genomics. Characterizing the genetic risk factors in AD is therefore a major objective; with the advent of high-throughput genomic techniques, a large number of putative AD-associated loci/genes have been reported[2]. However, much of the underlying heritability remains unexplained. Hence, increasing the sample size of genome-wide association studies (GWASs) is an obvious solution that has already been used to characterize new genetic risk factors in other common, complex diseases (e.g., diabetes).

### GWAS meta-analysis

The European Alzheimer & Dementia Biobank (EADB) consortium brings together the various European GWAS consortia already working on AD. A new dataset of 20,464 clinically diagnosed AD cases and 22,244 controls has been collated from 15 European countries. The EADB GWAS results were meta-analyzed with a proxy-AD GWASs of the UK Biobank (UKBB) dataset. The UKBB's proxy-AD designation is based on questionnaire data in which individuals are asked whether their parents had dementia. This method has been used successfully in the past[3] but is less specific than a clinical or pathological diagnosis of AD; hence, we will refer to these cases as proxy AD and related dementia (proxy-ADD). EADB stage I (GWAS meta-analysis) was based on 39,106 clinically diagnosed AD cases, 46,828 proxy-ADD cases (as defined in the Supplementary Note), 401,577 controls (Supplementary Tables 1 and 2) and 21,101,114 variants that passed our quality control (Fig. 1; see Supplementary Fig. 1 for the quantile–quantile plot and genomic inflation factors). We selected all variants with a $P$ value below $1 \times 10^{-5}$ in stage I. We defined nonoverlapping regions around these variants, excluded the region corresponding to *APOE* and examined the remaining variants in a large follow-up sample that included AD cases and controls from the ADGC, FinnGen and CHARGE consortia (stage II; 25,392 AD cases and 276,086 controls). A signal was considered as significant on the genome-wide level if it (1) was nominally associated ($P \leq 0.05$) in stage II, (2) had the same direction of association in the stage I and II analyses and (3) was associated with the ADD risk with

$P \leq 5 \times 10^{-8}$ in the stage I and stage II meta-analysis. Furthermore, we applied a PLINK clumping procedure[4] to define potential independent hits within the stage I results (Methods). After validation by conditional analyses (Supplementary Note and Supplementary Tables 3 and 4), this approach enabled us to define 39 signals in 33 loci already known to be associated with the risk of developing ADD[3,5–10] and identify 42 loci defined as new at the time of analysis (Tables 1 and 2, Supplementary Table 5 and Supplementary Figs. 2–29). Of the 42 new loci, 17 had $P \leq 5 \times 10^{-8}$ in stage I and 25 were associated with $P \leq 5 \times 10^{-8}$ after follow-up (stage I and stage II meta-analysis, including the ADGC, CHARGE and FinnGen data). We also identified 6 loci with $P \leq 5 \times 10^{-8}$ in the stage I and stage II analysis but with $P > 0.05$ in stage II (Supplementary Table 6). It is noteworthy that the magnitude of the associations in stage I did not change substantially if we restricted the analysis to clinically diagnosed AD cases (Supplementary Table 7 and Supplementary Fig. 30). Similarly, none of the signals observed appeared to be especially driven by the UKBB data (Supplementary Table 7 and Supplementary Figs. 2–29). Nine of these loci (*APP*, *CCDC6*, *GRN*, *LILRB2*, *NCK2*, *TNIP1*, *TMEM106B*, *TSPAN14* and *SHARPIN*) were recently reported in three articles using part of the GWAS data included in our study[11–13]. We also generated a detailed analysis of the human leukocyte antigen (*HLA*) locus on the basis of the clinically diagnosed AD cases (Supplementary Tables 8 and 9, Supplementary Figs. 31 and 32 and Supplementary Note).

### Genetic overlap with other neurodegenerative diseases

We tested the association of the lead variants within our new loci with the risk of developing other neurodegenerative diseases or AD-related disorders (Supplementary Fig. 33 and Supplementary Tables 10–12). We also performed more precise colocalization analyses (using Coloc R package, https://cran.r-project.org/web/packages/coloc/index.html) for five loci known to be associated with Parkinson's disease (*IDUA* and *CTSB*), types of frontotemporal dementia (*TMEM106B* and *GRN*) and amyotrophic lateral sclerosis (*TNIP1*) (Supplementary Tables 13 and 14). The *IDUA* signal for Parkinson's disease was independent of the signal in ADD (coloc posterior probability (PP)3 = 99.9%), but we were not able to determine whether the *CTSB* signals colocalized. The *TMEM106B* and *GRN* signals in frontotemporal lobar degeneration with TAR DNA-binding protein (TDP-43) inclusions (frontotemporal lobar

A full list of author and affiliations appears at the end of the paper.

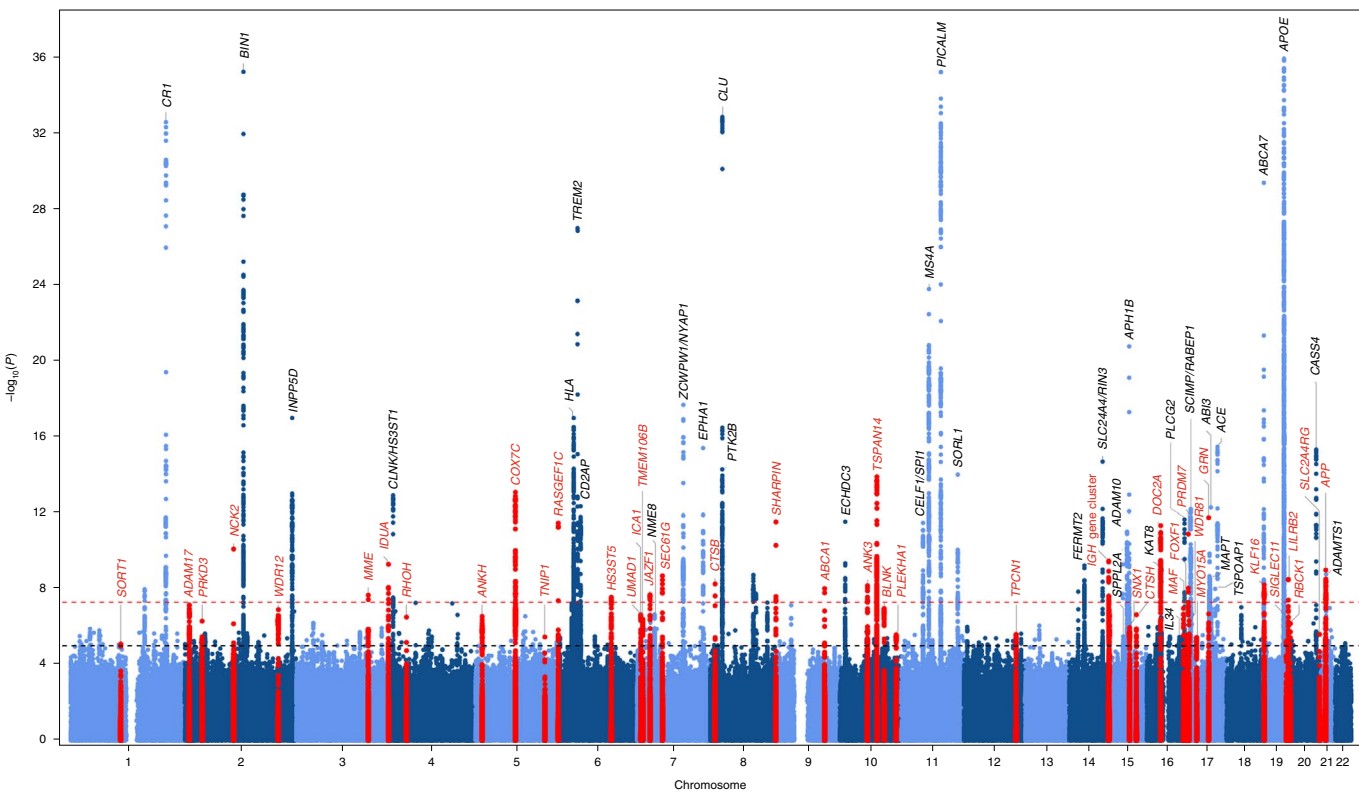

**Fig. 1 | Manhattan plot of the stage I results.** *P* values are two-sided raw *P* values derived from a fixed-effect meta-analysis. Variants with a *P* value below $1\times10^{-36}$ are not shown. Loci with a genome-wide significant signal are annotated (known loci in black and new loci in red). Variants in new loci are highlighted in red. The red dotted line represents the genome-wide significance level ($P=5\times10^{-8}$), and the black dotted line represents the suggestive significance level ($P=1\times10^{-5}$).

degeneration TDP) probably share causal variants with ADD (coloc PP4 = 99.8% and coloc PP4 = 80.1%, respectively). Lastly, we were not able to determine whether the *TNIP1* signals colocalized for ADD and amyotrophic lateral sclerosis.

## Pathway analyses

Next, we sought to perform a pathway enrichment analysis on the stage I association results to gain better biological understanding of this newly expanded genetic landscape for ADD. Ninety-three gene sets were still statistically significant after correction for multiple testing ($q\leq0.05$; Methods and Supplementary Table 15). As described previously, the most significant gene sets are related to amyloid and tau[5]; other significant gene sets are related to lipids, endocytosis and immunity (including macrophage and microglial cell activation). When restricting this analysis to the meta-analysis based on the clinically diagnosed AD cases, 54 gene sets were significant ($q\leq0.05$). Of these 54 gene sets, 33 reached $q\leq0.05$ in the stage I analysis and all reached $P\leq0.05$. This indicates that the inclusion of proxy-ADD cases does not cause disease-relevant biological information to be missed and underlines the additional power of this type of analysis.

We next performed a single-cell expression enrichment analysis by using the average gene expression per nucleus (Av. Exp.) data in the human Allen Brain Atlas (49,495 nuclei from 8 human brains). Only the microglial expression reached a high level of significance ($P=1.7\times10^{-8}$; Supplementary Table 16); greater expression corresponded to a more significant association with ADD. After adjusting for microglial Av. Exp., the remaining associations became nonsignificant; this indicates that microglial Av. Exp. drives all the other cell-type associations. These results were observed whatever the brain region studied (Supplementary Table 16). A similar result

was observed using a mouse single-cell dataset[14] (Supplementary Table 17 and Supplementary Note).

Lastly, we looked at whether the relationship between an elevated microglia Av. Exp. and a genetic association with the ADD risk was specific to particular biological processes (Supplementary Table 18) by analyzing the interaction between microglia Av. Exp. and pathway membership in MAGMA[15]. Of the five most significant interaction signals ($q\leq10^{-3}$), two were directly associated with endocytosis processes (GO:0006898 and GO:0031623); this suggested a functional relationship between microglia and endocytosis, which is known to be involved in phagocytosis (Supplementary Table 18). It is noteworthy that we also detected an interaction between GO:1902991 (regulation of amyloid precursor protein (APP) catabolic process) and the gene expression level in microglia ($q=1.4\times10^{-3}$; Supplementary Table 18). Even though these data suggest a functional relationship between microglia and APP/amyloid beta (Aβ) peptide pathways, this observation reinforces the likely involvement of microglial endocytosis in AD, a mechanism that is also strongly involved in APP metabolism[16]. Of note, there are overall similarities in the interaction effects of human and mouse microglia expression with genes in biological pathways of relevance to the AD genetic risk (Supplementary Table 18 and Supplementary Note).

## Gene prioritization

We next attempted to identify the genes most likely to be responsible for the association signal with ADD at each new locus. To this end, we studied the downstream effects of ADD-associated variants on molecular phenotypes (i.e., expression, splicing, protein expression, methylation and histone acetylation) in various *cis*-quantitative trait locus (*cis*-QTL) catalogues from AD-relevant tissues, cell types and brain regions. We investigated the genetic

**Table 1 | Summary of association results in the stage I and stage II analysis for known loci with a genome-wide significant signal**

| Variant[a] | Chromosome | Position[b] | Gene[c] | Known locus | Minor/major allele | MAF[d] | OR[e] | 95% CI | P value |
|---|---|---|---|---|---|---|---|---|---|
| rs679515 | 1 | 207577223 | CR1 | CR1 | T/C | 0.188 | 1.13 | 1.11–1.15 | $7.2 \times 10^{-46}$ |
| rs6733839 | 2 | 127135234 | BIN1 | BIN1 | T/C | 0.389 | 1.17 | 1.16–1.19 | $6.1 \times 10^{-118}$ |
| rs10933431 | 2 | 233117202 | INPP5D | INPP5D | G/C | 0.234 | 0.93 | 0.92–0.95 | $3.6 \times 10^{-18}$ |
| rs6846529 | 4 | 11023507 | CLNK | CLNK/HS3ST1 | C/T | 0.283 | 1.07 | 1.05–1.08 | $2.2 \times 10^{-17}$ |
| rs6605556 | 6 | 32615322 | HLA-DQA1 | HLA | G/A | 0.161 | 0.91 | 0.90–0.93 | $7.1 \times 10^{-20}$ |
| rs10947943 | 6 | 41036354 | UNC5CL | TREM2 | A/G | 0.142 | 0.94 | 0.93–0.96 | $1.1 \times 10^{-9}$ |
| rs143332484 | 6 | 41161469 | TREM2 | TREM2 | T/C | 0.013 | 1.41 | 1.32–1.50 | $2.8 \times 10^{-25}$ |
| rs75932628 | 6 | 41161514 | TREM2 | TREM2 | T/C | 0.003 | 2.39 | 2.09–2.73 | $2.5 \times 10^{-37}$ |
| rs60755019 | 6 | 41181270 | TREML2 | TREM2 | G/A | 0.004 | 1.55 | 1.33–1.80 | $2.1 \times 10^{-8}$ |
| rs7767350 | 6 | 47517390 | CD2AP | CD2AP | T/C | 0.271 | 1.08 | 1.06–1.09 | $7.9 \times 10^{-22}$ |
| rs6966331 | 7 | 37844191 | EPDR1 | NME8 | T/C | 0.349 | 0.96 | 0.94–0.97 | $4.6 \times 10^{-10}$ |
| rs7384878 | 7 | 100334426 | SPDYE3 | ZCWPW1/NYAP1 | C/T | 0.31 | 0.92 | 0.91–0.94 | $1.1 \times 10^{-26}$ |
| rs11771145 | 7 | 143413669 | EPHA1 | EPHA1 | A/G | 0.348 | 0.95 | 0.93–0.96 | $3.3 \times 10^{-14}$ |
| rs73223431 | 8 | 27362470 | PTK2B | PTK2B | T/C | 0.369 | 1.07 | 1.06–1.08 | $4.0 \times 10^{-22}$ |
| rs11787077 | 8 | 27607795 | CLU | CLU | T/C | 0.392 | 0.91 | 0.90–0.92 | $1.7 \times 10^{-44}$ |
| rs7912495 | 10 | 11676714 | USP6NL | ECHDC3 | G/A | 0.462 | 1.06 | 1.05–1.08 | $9.7 \times 10^{-19}$ |
| rs10437655 | 11 | 47370397 | SPI1 | CELF1/SPI1 | A/G | 0.399 | 1.06 | 1.04–1.07 | $5.3 \times 10^{-14}$ |
| rs1582763 | 11 | 60254475 | MS4A4A | MS4A | A/G | 0.371 | 0.91 | 0.90–0.92 | $3.7 \times 10^{-42}$ |
| rs3851179 | 11 | 86157598 | EED | PICALM | T/C | 0.358 | 0.9 | 0.89–0.92 | $3.0 \times 10^{-48}$ |
| rs74685827 | 11 | 121482368 | SORL1 | SORL1 | G/T | 0.019 | 1.19 | 1.13–1.25 | $2.8 \times 10^{-11}$ |
| rs11218343 | 11 | 121564878 | SORL1 | SORL1 | C/T | 0.039 | 0.84 | 0.81–0.87 | $1.4 \times 10^{-21}$ |
| rs17125924 | 14 | 52924962 | FERMT2 | FERMT2 | G/A | 0.089 | 1.1 | 1.07–1.12 | $8.3 \times 10^{-16}$ |
| rs7401792 | 14 | 92464917 | SLC24A4 | SLC24A4/RIN3 | G/A | 0.371 | 1.04 | 1.02–1.05 | $4.8 \times 10^{-8}$ |
| rs12590654 | 14 | 92472511 | SLC24A4 | SLC24A4/RIN3 | A/G | 0.328 | 0.93 | 0.92–0.95 | $4.2 \times 10^{-21}$ |
| rs8025980 | 15 | 50701814 | SPPL2A | SPPL2A | G/A | 0.345 | 0.96 | 0.94–0.97 | $1.3 \times 10^{-8}$ |
| rs602602 | 15 | 58764824 | MINDY2 | ADAM10 | A/T | 0.28 | 0.94 | 0.93–0.96 | $2.1 \times 10^{-15}$ |
| rs117618017 | 15 | 63277703 | APH1B | APH1B | T/C | 0.144 | 1.11 | 1.09–1.13 | $2.2 \times 10^{-25}$ |
| rs889555 | 16 | 31111250 | BCKDK | KAT8 | T/C | 0.281 | 0.95 | 0.94–0.97 | $2.0 \times 10^{-11}$ |
| rs4985556 | 16 | 70660097 | IL34 | IL34 | A/C | 0.115 | 1.07 | 1.05–1.09 | $6.0 \times 10^{-10}$ |
| rs12446759 | 16 | 81739398 | PLCG2 | PLCG2 | G/A | 0.403 | 0.95 | 0.94–0.96 | $1.2 \times 10^{-13}$ |
| rs72824905 | 16 | 81908423 | PLCG2 | PLCG2 | G/C | 0.008 | 0.74 | 0.68–0.81 | $8.5 \times 10^{-12}$ |
| rs7225151 | 17 | 5233752 | SCIMP | SCIMP/RABEP1 | A/G | 0.124 | 1.08 | 1.05–1.10 | $4.1 \times 10^{-13}$ |
| rs199515 | 17 | 46779275 | WNT3 | MAPT | G/C | 0.219 | 0.94 | 0.93–0.96 | $9.3 \times 10^{-13}$ |
| rs616338 | 17 | 49219935 | ABI3 | ABI3 | T/C | 0.012 | 1.32 | 1.23–1.42 | $2.8 \times 10^{-14}$ |
| rs2526377 | 17 | 58332680 | TSPOAP1 | TSPOAP1 | G/A | 0.445 | 0.95 | 0.94–0.97 | $1.6 \times 10^{-12}$ |
| rs4277405 | 17 | 63471557 | ACE | ACE | C/T | 0.384 | 0.94 | 0.93–0.95 | $8.8 \times 10^{-20}$ |
| rs12151021 | 19 | 1050875 | ABCA7 | ABCA7 | A/G | 0.336 | 1.1 | 1.09–1.12 | $1.6 \times 10^{-37}$ |
| rs6014724 | 20 | 56423488 | CASS4 | CASS4 | G/A | 0.09 | 0.89 | 0.87–0.91 | $4.1 \times 10^{-21}$ |
| rs2830489 | 21 | 26775872 | ADAMTS1 | ADAMTS1 | T/C | 0.281 | 0.95 | 0.94–0.97 | $1.7 \times 10^{-10}$ |

P values are two-sided raw P values derived from a fixed-effect meta-analysis. CI, confidence interval; OR, odds ratio; MAF, minor allele frequency. [a]Reference single-nucleotide polymorphism (SNP) (rs) number, according to dbSNP build 153. [b]GRCh38 assembly. [c]Nearest protein-coding gene according to GENCODE release 33. [d]Weighted average MAF across all discovery studies. [e]Approximate OR calculated with respect to the minor allele.

colocalization between association signals for the ADD risk and those for the molecular phenotypes and the association between the ADD risk and these phenotypes by integrating cis-QTL information into our ADD GWAS. Moreover, we considered the lead variant annotation (the allele frequency, protein-altering effects and nearest protein-coding gene) and a genome-wide, high-content short interfering RNA screen for APP metabolism[17]. Based on this evidence, we developed a systematic gene prioritization strategy

that yielded a total weighted score of between 0 and 100 for each gene (Supplementary Fig. 34 and Supplementary Note). This score was used to compare and prioritize genes in the new loci within 1 Mb upstream and 1 Mb downstream of the lead variants. Genes either were ranked as tier 1 (greater likelihood of being the causal risk gene responsible for the ADD signal) or tier 2 (lower likelihood and the absence of a minimum level of evidence as a causal risk gene) or were not ranked.

**Table 2 | Summary of association results in the stage I and stage II analysis for new loci at the time of analysis with a genome-wide significant signal**

| Locus number | Variant[a] | Chromosome | Position[b] | Gene[c] | Minor/major allele | MAF[d] | OR[e] | 95% CI | P value |
|---|---|---|---|---|---|---|---|---|---|
| 1 | rs141749679 | 1 | 109345810 | SORT1 | C/T | 0.004 | 1.38 | 1.24–1.54 | $7.5 \times 10^{-9}$ |
| 2 | rs72777026 | 2 | 9558882 | ADAM17 | G/A | 0.144 | 1.06 | 1.04–1.08 | $2.7 \times 10^{-8}$ |
| 3 | rs17020490 | 2 | 37304796 | PRKD3 | C/T | 0.145 | 1.06 | 1.04–1.08 | $3.3 \times 10^{-9}$ |
| 4 | rs143080277 | 2 | 105749599 | NCK2 | C/T | 0.005 | 1.47 | 1.33–1.63 | $2.1 \times 10^{-13}$ |
| 5 | rs139643391 | 2 | 202878716 | WDR12 | T/TC | 0.131 | 0.94 | 0.92–0.96 | $1.1 \times 10^{-8}$ |
| 6 | rs16824536 | 3 | 155069722 | MME | A/G | 0.054 | 0.92 | 0.89–0.95 | $3.6 \times 10^{-8}$ |
| 6 | rs61762319 | 3 | 155084189 | MME | G/A | 0.026 | 1.16 | 1.11–1.21 | $2.2 \times 10^{-11}$ |
| 7 | rs3822030 | 4 | 993555 | IDUA | G/T | 0.429 | 0.95 | 0.94–0.96 | $8.3 \times 10^{-12}$ |
| 8 | rs2245466 | 4 | 40197226 | RHOH | G/C | 0.343 | 1.05 | 1.03–1.06 | $1.2 \times 10^{-9}$ |
| 9 | rs112403360 | 5 | 14724304 | ANKH | A/T | 0.073 | 1.09 | 1.06–1.12 | $2.3 \times 10^{-9}$ |
| 10 | rs62374257 | 5 | 86927378 | COX7C | C/T | 0.23 | 1.07 | 1.05–1.09 | $1.4 \times 10^{-15}$ |
| 11 | rs871269 | 5 | 151052827 | TNIP1 | T/C | 0.326 | 0.96 | 0.95–0.97 | $8.7 \times 10^{-9}$ |
| 12 | rs113706587 | 5 | 180201150 | RASGEF1C | A/G | 0.11 | 1.09 | 1.07–1.12 | $2.2 \times 10^{-16}$ |
| 13 | rs785129 | 6 | 114291731 | HS3ST5 | T/C | 0.35 | 1.04 | 1.03–1.06 | $2.4 \times 10^{-9}$ |
| 14 | rs6943429 | 7 | 7817263 | UMAD1 | T/C | 0.42 | 1.05 | 1.03–1.06 | $1.0 \times 10^{-10}$ |
| 15 | rs10952097 | 7 | 8204382 | ICA1 | T/C | 0.114 | 1.07 | 1.05–1.10 | $6.8 \times 10^{-9}$ |
| 16 | rs13237518 | 7 | 12229967 | TMEM106B | A/C | 0.411 | 0.96 | 0.94–0.97 | $4.9 \times 10^{-11}$ |
| 17 | rs1160871 | 7 | 28129126 | JAZF1 | G/GTCTT | 0.222 | 0.95 | 0.93–0.97 | $9.8 \times 10^{-9}$ |
| 18 | rs76928645 | 7 | 54873635 | SEC61G | T/C | 0.103 | 0.93 | 0.91–0.95 | $1.6 \times 10^{-10}$ |
| 19 | rs1065712 | 8 | 11844613 | CTSB | C/G | 0.053 | 1.09 | 1.06–1.12 | $1.9 \times 10^{-9}$ |
| 20 | rs34173062 | 8 | 144103704 | SHARPIN | A/G | 0.081 | 1.13 | 1.09–1.16 | $1.7 \times 10^{-16}$ |
| 21 | rs1800978 | 9 | 104903697 | ABCA1 | G/C | 0.13 | 1.06 | 1.04–1.08 | $1.6 \times 10^{-9}$ |
| 22 | rs7068231 | 10 | 60025170 | ANK3 | T/G | 0.403 | 0.95 | 0.94–0.96 | $3.3 \times 10^{-13}$ |
| 23 | rs6586028 | 10 | 80494228 | TSPAN14 | C/T | 0.196 | 0.93 | 0.91–0.94 | $2.0 \times 10^{-19}$ |
| 24 | rs6584063 | 10 | 96266650 | BLNK | G/A | 0.043 | 0.89 | 0.86–0.92 | $6.7 \times 10^{-11}$ |
| 25 | rs7908662 | 10 | 122413396 | PLEKHA1 | G/A | 0.467 | 0.96 | 0.95–0.97 | $2.6 \times 10^{-9}$ |
| 26 | rs6489896 | 12 | 113281983 | TPCN1 | C/T | 0.076 | 1.08 | 1.05–1.10 | $1.8 \times 10^{-9}$ |
| 27 | rs7157106 | 14 | 105761758 | IGH gene cluster | A/G | 0.36 | 1.05 | 1.03–1.07 | $2.0 \times 10^{-8}$ |
| 27 | rs10131280 | 14 | 106665591 | IGH gene cluster | A/G | 0.133 | 0.94 | 0.92–0.96 | $4.3 \times 10^{-10}$ |
| 28 | rs3848143 | 15 | 64131307 | SNX1 | G/A | 0.22 | 1.05 | 1.04–1.07 | $8.4 \times 10^{-11}$ |
| 29 | rs12592898 | 15 | 78936857 | CTSH | A/G | 0.133 | 0.94 | 0.92–0.96 | $4.2 \times 10^{-9}$ |
| 30 | rs1140239 | 16 | 30010081 | DOC2A | T/C | 0.379 | 0.94 | 0.93–0.96 | $2.6 \times 10^{-13}$ |
| 31 | rs450674 | 16 | 79574511 | MAF | C/T | 0.373 | 0.96 | 0.95–0.98 | $3.2 \times 10^{-8}$ |
| 32 | rs16941239 | 16 | 86420604 | FOXF1 | A/T | 0.029 | 1.13 | 1.08–1.17 | $1.3 \times 10^{-8}$ |
| 33 | rs56407236 | 16 | 90103687 | PRDM7 | A/G | 0.069 | 1.11 | 1.08–1.14 | $6.5 \times 10^{-15}$ |
| 34 | rs35048651 | 17 | 1728046 | WDR81 | T/TGAG | 0.214 | 1.06 | 1.04–1.08 | $7.7 \times 10^{-11}$ |
| 35 | rs2242595 | 17 | 18156140 | MYO15A | A/G | 0.112 | 0.94 | 0.92–0.96 | $1.1 \times 10^{-9}$ |
| 36 | rs5848 | 17 | 44352876 | GRN | T/C | 0.289 | 1.07 | 1.06–1.09 | $2.4 \times 10^{-20}$ |
| 37 | rs149080927 | 19 | 1854254 | KLF16 | G/GC | 0.48 | 1.05 | 1.04–1.07 | $5.1 \times 10^{-10}$ |
| 38 | rs9304690 | 19 | 49950060 | SIGLEC11 | T/C | 0.24 | 1.05 | 1.03–1.07 | $4.7 \times 10^{-9}$ |
| 39 | rs587709 | 19 | 54267597 | LILRB2 | C/T | 0.325 | 1.05 | 1.04–1.07 | $3.6 \times 10^{-11}$ |
| 40 | rs1358782 | 20 | 413334 | RBCK1 | A/G | 0.246 | 0.95 | 0.94–0.97 | $1.6 \times 10^{-8}$ |
| 41 | rs6742 | 20 | 63743088 | SLC2A4RG | T/C | 0.221 | 0.95 | 0.93–0.97 | $2.6 \times 10^{-9}$ |
| 42 | rs2154481 | 21 | 26101558 | APP | C/T | 0.476 | 0.95 | 0.94–0.97 | $1.0 \times 10^{-12}$ |

P values are two-sided raw P values derived from a fixed-effect meta-analysis. [a]rs number, according to dbSNP build 153. [b]GRCh38 assembly. [c]Nearest protein-coding gene according to GENCODE release 33. [d]Weighted average MAF across all discovery studies. [e]Approximate OR calculated with respect to the minor allele.

From all newly identified loci, this gene prioritization yielded 31 tier 1 genes and 24 tier 2. The 55 prioritized genes, the details of the analyses and the supporting evidence are summarized in Fig. 2a and the Supplementary Note (Supplementary Tables 19–30 and Supplementary Figs. 35–45). Among the 31 tier 1 genes, we observed that 25 of these genes were the only prioritized gene in their respective locus. For the remaining 6 tier 1 genes, we also found tier 2 genes in their respective locus. We also identified five loci containing several tier 2 prioritized genes. In one of these loci, locus 39 (L39), the tier 2 prioritized gene *LILRB2* had strong additional support from published literature (Supplementary Note). In five loci, our prioritization score did not identify sufficient molecular evidence to prioritize genes with exception of being the nearest gene (L10, L12, L13, L14 and L32). Finally, we excluded the complex IGH cluster (L27) from gene prioritization analyses due to genomic complexity of the telomeric locus as a consequence of known fusion events[18].

We highlight two examples, L18 and L23. In L18, the lead variant, rs76928645 (MAF = 10%), is intergenic and is located more than 100 kb downstream or upstream of the two nearest protein-coding genes (*SEC61G* and *EGFR*, respectively). Our gene prioritization analyses suggested that *EGFR* was the only risk gene (Fig. 3). We found that both the lead variant (rs76928645) and the other nearby variants in linkage disequilibrium (LD) are significant expression QTLs (eQTLs) for regulating *EGFR* expression downstream. The eQTL signals in brain strongly colocalized with the GWAS signal (with eQTL coloc PP4s of 98.3% in the temporal cortex (TCX) and 99.5% in the dorsolateral prefrontal cortex (DLPFC)). Accordingly, the fine-mapped expression transcriptome-wide association study (eTWAS) associations (Fine-mapping Of CaUsal gene Sets (FOCUS) posterior inclusion probability (PIP) = 1; eTWAS $P = 6.9 \times 10^{-9}$, eTWAS $Z = +5.8$ in the TCX; eTWAS $P = 3.1 \times 10^{-11}$, eTWAS $Z = +6.6$ in the DLPFC) indicated that genetic downregulation of *EGFR* expression is associated with a lower ADD risk (Fig. 3; Supplementary Tables 22, 24 and 26; and Supplementary Figs. 36a, 39 and 41).

In L23, we observed numerous eQTL-GWAS and methylation QTL (mQTL)-GWAS hits for *TSPAN14* that support the hypothesis that increased brain expression of *TSPAN14* is associated with increased ADD risk. We also identified several splice junctions in *TSPAN14* whose genetic regulation signals in lymphoblastoid cell lines (LCLs) and brain colocalized with the ADD association signal. These splice junctions were also associated with ADD risk (Fig. 4, Supplementary Tables 22–28 and Supplementary Figs. 36–42 and 44c). As three of these splice junctions were related to new complex cryptic splicing events that were predicted to result in two cryptic exons not previously described in known *TSPAN14* transcripts (based on GENCODE v38), we designed a long-read single-molecule (Nanopore) sequencing experiment (Supplementary Note) to validate these cryptic exons on a total of 93 complementary DNA (cDNA) samples derived from LCLs, frontal cortex and hippocampus and consequently validated those cryptic exons (Fig. 4). All three of the validated cryptic splicing events occur within the ADAM10-interacting domain of TSPAN14. Cryptic exon 1 is at least 45 bp long, and cryptic exon 2 is 118 bp long.

Lastly, we used STRING v11 (ref. [19]) to analyze protein–protein interaction for (1) previously known AD genes from GWASs, (2) our prioritized new genes (tier 1 in Fig. 2a and Supplementary Table 20) and (3) a combination of the two (Supplementary Note). The largest networks contained 14, 8 and 30 proteins, respectively (Supplementary Fig. 46). These networks were larger than would be expected by chance (respectively, $P < 2 \times 10^{-5}$, $P = 2.8 \times 10^{-3}$ and $P < 2 \times 10^{-5}$ based on comparison with 50,000 randomly simulated protein lists matched for the number of proteins and the total number of interactions for each protein). Notably, the number of interactions between our prioritized genes and previously known genes is also significantly greater than would be expected ($P < 1 \times 10^{-4}$), indicating that the newly prioritized genes are biologically relevant in AD. No such enrichment ($P = 0.88$) was observed for the remaining genes in the new loci, again highlighting the value of our prioritization approach.

We next performed a pathway enrichment analysis of the tier 1 genes using STRING. We found that several gene sets linked to the immune system remained statistically significant after correction for multiple testing (Fig. 2b and Supplementary Table 31), especially regulation of the tumor necrosis factor (TNF)-mediated signaling pathway (GO:0010803). We report the potential genetic implication of the linear ubiquitin chain assembly complex (LUBAC), which is a major regulator of the aforementioned signaling pathway[20]. Two of the LUBAC's three complements are encoded by the new tier 1 prioritized genes *SHARPIN* and *RBCK1*, and the complex's function is directly regulated by *OTULIN* (also a new tier 1 prioritized gene).

## GRS

We next looked at whether the genetic ADD burden (as measured by a genetic risk score (GRS)) generated from our genome-wide significant variants ($n = 83$, excluding *APOE*; Supplementary Table 32) might influence the rate of conversion to AD in (1) individuals from several prospective, population-based cohorts and (2) patients with mild cognitive impairment (MCI) in prospective memory clinic studies (Supplementary Table 33). We used Cox regression models to assess the association after adjustment for age at baseline, sex, the number of APOE-ε4 and APOE-ε2 alleles, and genetic principal components (PCs).

In population-based cohorts with clinically diagnosed AD cases, the GRS was significantly associated with conversion to AD; this was shown in a fixed-effect meta-analysis (hazard ratio (HR) (95%CI) per average risk allele = 1.076 (1.064–1.088), $P = 9.2 \times 10^{-40}$; Fig. 5 and Supplementary Table 34). Likewise, the GRS was significantly associated with AD conversion in patients with MCI (HR = 1.056 (1.040–1.072), $P = 2.8 \times 10^{-12}$; Fig. 5 and Supplementary Table 35). Furthermore, we found that the GRS association increased significantly when the new variants discovered in the present study were added to the previously described variants (Supplementary Table 36) for both population-based studies (HR = 1.052 (1.037–1.068), $P = 1.5 \times 10^{-11}$) and MCI cohorts (HR = 1.034 (1.013–1.055), $P = 1.4 \times 10^{-3}$).

**Fig. 2 | Gene prioritization. a**, Summary of weighted scores for each evidence category for the prioritized genes in the 42 new genome-wide-significant loci. Using our gene prioritization method, we considered the genes within 1 Mb of each new lead variant and prioritized a total of 55 genes in 42 new loci at two different confidence levels (31 tier 1 genes and 24 tier 2 genes). The leftmost squares indicate the new locus index number. The different types of evidence are colored according to the seven different domains to which they belonged. Weighted scores for each evidence category are rescaled to a 0–100 scale, and the proportions of mean human brain cell-type-specific expression for each gene are also rescaled to a 0–100 scale; darker colors represent higher scores or higher expression proportions. Tier 1 genes are shown in dark green, and tier 2 genes are shown in light green. Only tier 1 and tier 2 genes are shown for each locus. Supplementary Fig. 35 shows full results. MAFs and CADD (v1.6) PHRED scores for rare and/or protein-altering rare variants are labeled in white within the respective squares. **b**, Pathway enrichment analysis based on the tier 1 gene list. Only the ten strongest associations (according to STRING software) are presented here. coloc, colocalization; eQTL, expression QTL; eTWAS, expression transcriptome-wide association study; GO, Gene Ontology; haQTL, histone acetylation QTL; Mon. Mac., monocytes and macrophages; sTWAS, splicing transcriptome-wide association study; m/haQTL, methylation/histone acetylation QTL; sQTL, splicing QTL; FDR, false discovery rate.

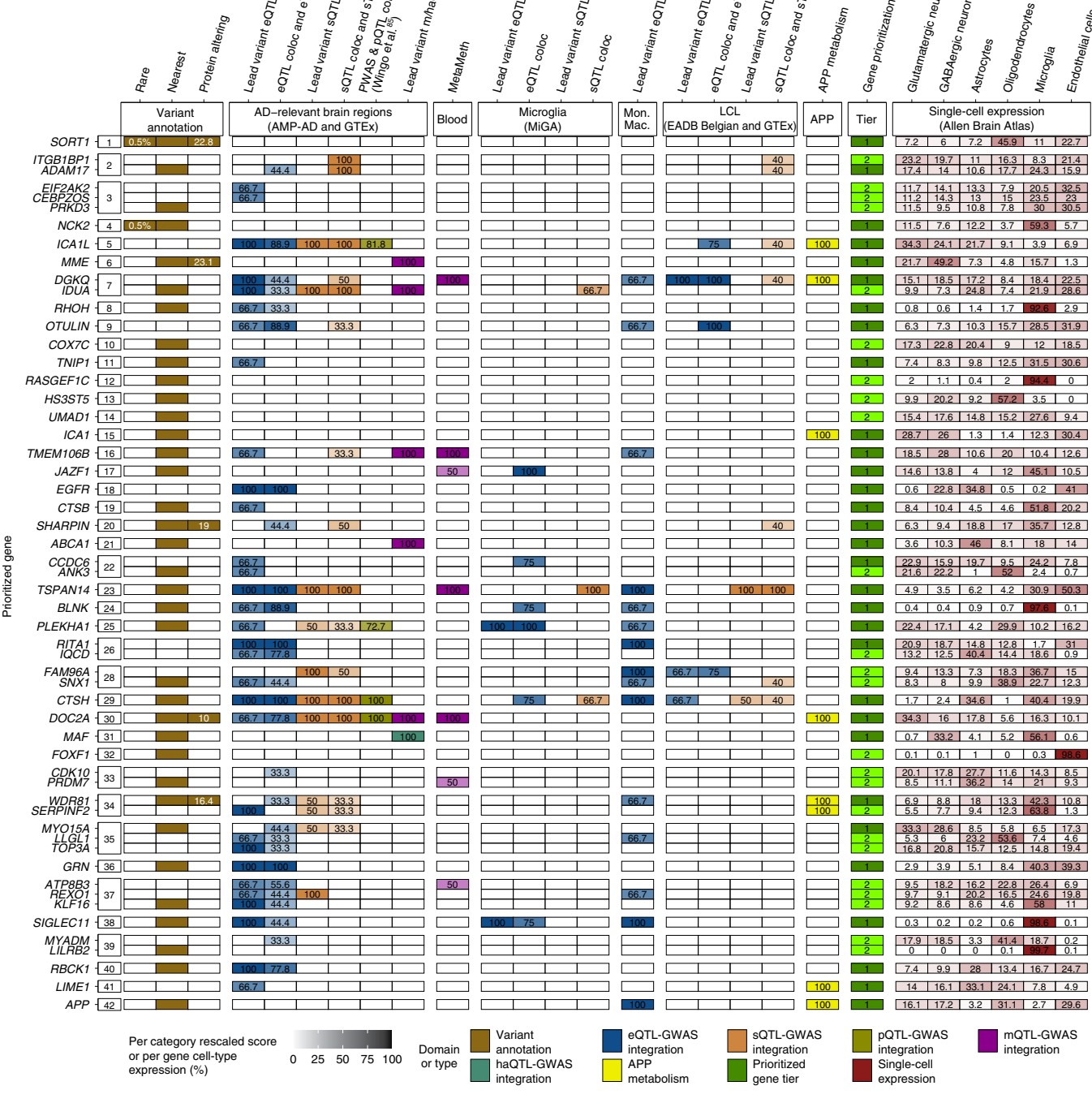

**a**

**b**

| GO term | Term description | Number of tier 1 genes | Number of genes in pathway | FDR | Matching proteins in the network |
|---|---|---|---|---|---|
| GO:0001775 | Cell activation | 12 | 1,024 | $4.09 \times 10^{-5}$ | GRN,CTSH, BLNK, NCK2, DGKQ, EGFR, APP, ADAM17, CTSB, RHOH, TSPAN14, MME |
| GO:0002376 | Immune system process | 15 | 2,370 | $2.60 \times 10^{-4}$ | GRN,CTSH, BLNK, NCK2, OTULIN, APP, LIME1, ADAM17, TNIP1, CTSB, RBCK1, PLEKHA1, RHOH, TSPAN14, MME |
| GO:0009966 | Regulation of signal transduction | 17 | 3,033 | $2.60 \times 10^{-4}$ | GRN,CTSH, BLNK, NCK2, DGKQ, EGFR, OTULIN, APP, ADAM17, TNIP1, RBCK1, PLEKHA1, ABCA1, RHOH, SHARPIN, TSPAN14, RITA1 |
| GO:0010646 | Regulation of cell communication | 18 | 3,327 | $2.60 \times 10^{-4}$ | GRN,CTSH, BLNK, NCK2, DGKQ, EGFR, OTULIN, APP, ADAM17, TNIP1, RBCK1, PLEKHA1, ABCA1, RHOH, ICA1, SHARPIN, TSPAN14, RITA1 |
| GO:0023051 | Regulation of signaling | 18 | 3,360 | $2.60 \times 10^{-4}$ | GRN,CTSH, BLNK, NCK2, DGKQ, EGFR, OTULIN, APP, ADAM17, TNIP1, RBCK1, PLEKHA1, ABCA1, RHOH, ICA1, SHARPIN, TSPAN14, RITA1 |
| GO:0045321 | Leukocyte activation | 10 | 894 | $2.60 \times 10^{-4}$ | GRN, CTSH, BLNK, NCK2, APP, ADAM17, CTSB, RHOH, TSPAN14, MME |
| GO:0048583 | Regulation of response to stimulus | 19 | 3,882 | $2.60 \times 10^{-4}$ | GRN,CTSH, BLNK, NCK2, DGKQ, EGFR, OTULIN, APP, LIME1, ADAM17, TNIP1, RBCK1, PLEKHA1, ABCA1, RHOH, SHARPIN, TSPAN14, SIGLEC11, RITA1 |
| GO:0048584 | Positive regulation of response to stimulus | 14 | 2,054 | $2.60 \times 10^{-4}$ | CTSH, BLNK, NCK2, EGFR, OTULIN, APP, LIME1, ADAM17, TNIP1, RBCK1, PLEKHA1, SHARPIN, TSPAN14 |
| GO:0010803 | Regulation of TNF-mediated signaling pathway | 4 | 56 | $4.30 \times 10^{-4}$ | OTULIN, ADAM17, RBCK1, SHARPIN |
| GO:0042058 | Regulation of EGFR signaling pathway | 4 | 80 | $1.50 \times 10^{-3}$ | NCK2, EGFR, APP, ADAM17 |

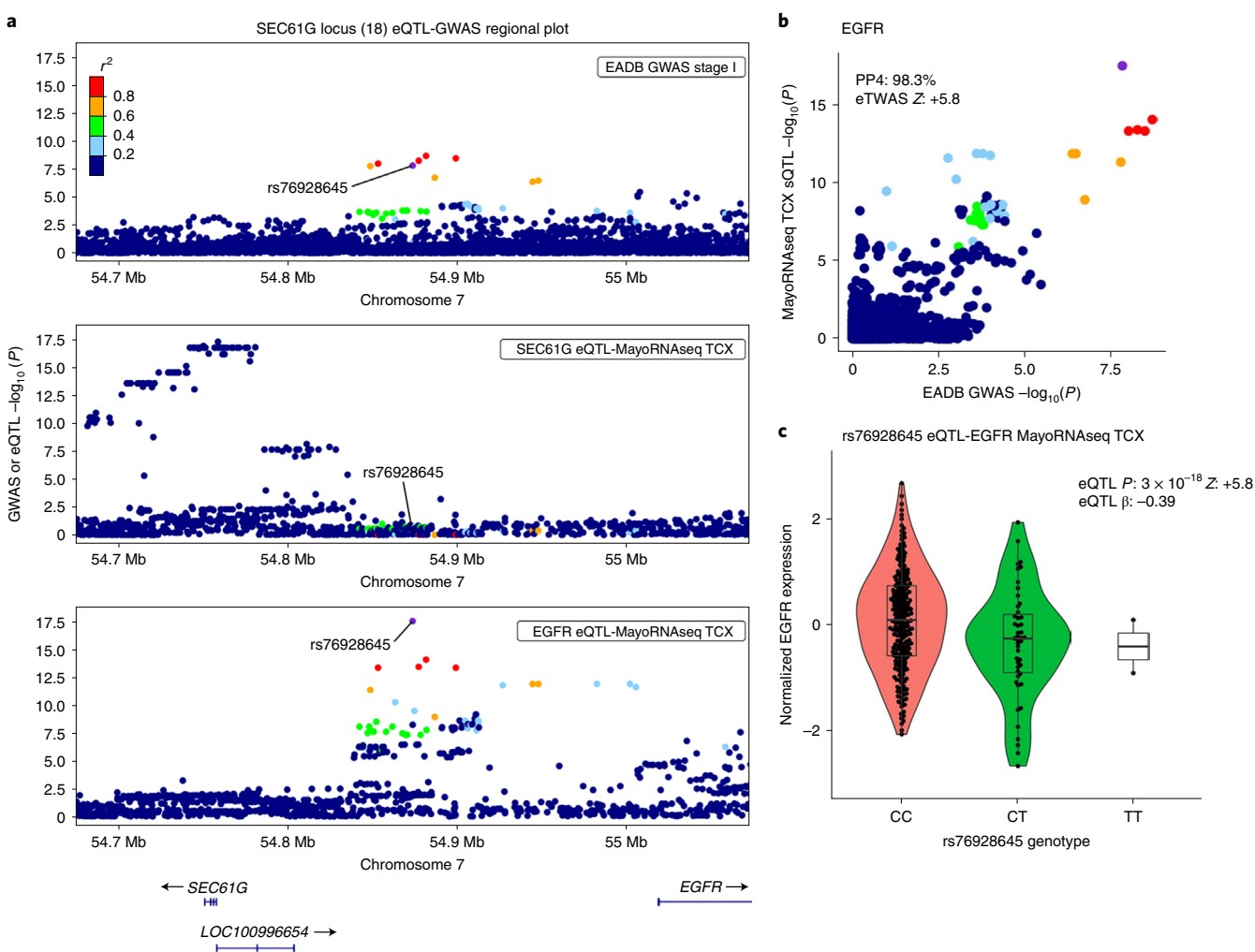

**Fig. 3 | Regulation of EGFR expression by the ADD-risk-associated and colocalized brain eQTLs within the intergenic SEC61G locus. a**, The regional plot of the new *SEC61G* locus (L18) shows the EADB GWAS stage I ($n = 487,511$) ADD association signal within 200 kb of the intergenic lead variant, rs76928645 (the two closest protein-coding genes, *SEC61G* and *EGFR*, are more than 100 kb from the lead variant), together with the eQTLs in the same region identified for *SEC61G* and *EGFR* expression separately in the TCX (MayoRNAseq TCX eQTL catalog based on $n = 259$ individuals). The rs7692864 lead variant is shown in purple, and LD $r^2$ values (calculated for the EADB Trans-Omics for Precision Medicine (TOPMed) dataset ($n = 42,140$) with respect to the lead variant) are indicated on a color scale. *y* axis, $-\log_{10}$ for the GWAS or eQTL *P* value; *x* axis, hg38 genomic position on chromosome 7. **b**, Colocalization between the *EGFR* eQTL signal (MayoRNAseq TCX, $n = 259$ individuals) and the EADB GWAS stage I ($n = 487,511$) signal (eQTL coloc PP4 = 98.3%); with the significant eTWAS association (eTWAS $P = 6.9 \times 10^{-9}$ and eTWAS $Z = 5.8$) and fine-mapped (FOCUS PIP = 1) eTWAS association in the same catalog. *y* axis, eQTL $-\log_{10}(P)$ value; *x* axis, GWAS $-\log_{10}(P)$ value. LD $r^2$ values and color scales are as in **a**. **c**, The eQTL violin plot shows a significant association (eQTL $P = 3 \times 10^{-18}$) between the rs76928645 lead variant genotype and *EGFR* expression in the TCX (MayoRNAseq TCX, $n = 259$ individuals), where the protective allele T is associated with lower EGFR expression (eQTL β, −0.39). Each data point represents a sample whose normalized *EGFR* expression value is indicated on the *y* axis, and the rs76928645 genotype information is indicated on the *x* axis. The lower and upper hinges of the boxes represent the first and third quantiles, the whiskers extend 1.5 times the interquartile range from the hinges and the line represents the median.

Importantly, the results of our meta-analysis suggest that the risk of conversion to AD rises with the number of risk alleles from non-APOE risk variants in the GRS by 1.9-fold in population-based cohorts (HR = 1.93 (1.75–2.13); Fig. 5) and 1.6-fold in MCI cohorts (HR = 1.63 (1.42–1.87); Fig. 6) on top of effects of age and the *APOE* ε4 allele. These observations result from the comparison of hypothetical individuals with a GRS value at the first decile of the distribution versus those with a GRS value at the ninth decile (Fig. 6). With regard to *APOE*, carrying an additional APOE-ε4 allele was associated with a slightly higher increase in the AD risk in population-based cohorts (HR = 2.19 (2.03–2.37)) and MCI cohorts (HR = 1.90 (1.73–2.07)). There was no interaction between the GRS and the number of *APOE*-ε4 alleles (Supplementary Table 37).

In an MCI cohort setting, this effect of the GRS corresponds to a median AD conversion probability within 3 years of 21.9% in patients with a GRS below the first decile (range, 4.1–34.9%) and 37.5% (range, 10.8–56.2%) in patients with a GRS above the ninth decile. There was a consistent increase in probability between these deciles in all cohorts (median (range), 13.8% (6.6–25.0%); Supplementary Table 38).

To better define the GRS discriminative ability regarding AD conversion, we assessed the improvements in three indices of predictive performance after adding the GRS to a Cox model containing age, sex, PCs and the number of *APOE*-ε4 and *APOE*-ε2 alleles as covariates (Supplementary Tables 34 and 35). We found a small but consistent increase in the discrimination between AD converters and nonconverters, as indicated by the concordance index (C-index) in

population-based cohorts ($\Delta_{5\text{years}}$-C-index$_{\text{fixed-effects}} = 0.002$ (0.0004–0.004)) and MCI cohorts ($\Delta_{3\text{years}}$-C-index$_{\text{fixed-effects}} = 0.007$ (0.001–0.012)). This finding was further supported by small-to-moderate increases in the continuous NRI (net reclassification improvement) index in population-based cohorts (NRI$_{5\text{year-fixed-effects}} = 0.248$ (0.159–0.336)) and MCI cohorts (NRI$_{3\text{year-fixed-effects}} = 0.232$ (0.140–0.325)); this indicates that the risk assignment is more appropriate to individuals when the GRS is taken into account[21]. Furthermore, an increase in the index of prediction accuracy (IPA) was observed in all of the population-based cohorts (average $\Delta_{5\text{years}}$-IPA$_{\text{fixed-effects}} = 0.29\%$ (0.23%–0.35%)) and all but one of the MCI cohorts (average $\Delta_{3\text{years}}$-IPA$_{\text{fixed-effects}} = 1.53\%$ (1.31%–1.76%)), indicating an overall improvement in predictive performance. As expected, the amount of improvement in this index varied greatly from one cohort to another, given its dependency on incidence rates. The value of adding the new genetic variants was emphasized by the fact that effect sizes (as measured by the indices of predictive ability) were lower when only previously known AD risk variants were included in the GRS (Supplementary Table 39).

The results were similar when we (1) computed indices for other follow-up time points, (2) applied a random effects meta-analysis, (3) considered conversion to all-cause-dementia as the outcome and (4) excluded the Framingham Heart Study (FHS), as it was part of the stage II of the GWAS from which ORs for PRS computation were extracted (Supplementary Tables 34–44 and Supplementary Fig. 47).

## Discussion

Our meta-analysis combined a large, new case–control study with previous GWASs. We identified 75 independent loci for ADD; 33 had been reported previously, and 42 correspond to new signals at the time of this analysis. The prioritized genes and their potential impact on the pathophysiology of AD are described in the Supplementary Note.

Our pathway enrichment analyses removed ambiguities concerning the involvement of tau-binding proteins and APP/Aβ peptide metabolism in late-onset AD processes at a much higher level than had been described previously[5]. It is noteworthy that new genetic risk factors are often first evaluated in the context of known pathways; many new research approaches were developed to systematically characterize putative links among APP metabolism, tau function and ADD genetic risk factors[22,23]. This approach can lead to circular reasoning and thus artificial enrichment in specific processes. However, we implicate *ADAM17*, a gene whose protein product is known to carry α-secretase activity as ADAM10 (ref. [24]). This observation suggests that the nonamyloidogenic pathway for APP metabolism might be deregulated in AD. In addition to *APP*, we also identified six highly plausible prioritized (tier 1) genes (*ICA1L*, *DGKQ*, *ICA1*, *DOC2A*, *WDR81* and *LIME1*) that are likely to modulate the metabolism of APP.

These pathway enrichment analyses also confirmed the involvement of innate immunity and microglial activation in ADD (Supplementary Table 15). Our single-cell expression enrichment analysis also highlighted genes expressed in microglia (Supplementary Tables 16 and 17). Indeed, three of our prioritized (tier 1) genes (*RHOH*, *BLNK* and *SIGLEC11*) and two of our tier 2 genes (*LILRB2* and *RASGE1FC*) appeared to be mainly expressed in microglia (>90% relative to the total expression summed across cell types; Fig. 2a and Supplementary Table 45). Importantly, *SIGLEC11* and *LILRB2* have already been linked to Aβ peptides/amyloid plaques[25,26].

Here, we also provide genetic evidence of the LUBAC's potential implication in ADD. Two of the LUBAC's three complements are encoded by *SHARPIN* and *RBCK1*, and the LUBAC is regulated by OTULIN; all three genes were found to be high-confidence, prioritized risk genes in our study. The LUBAC is the only E3 ligase known to form linear ubiquitin chains de novo through ubiquitin's N-terminal methionine. The complex has mostly been studied in the context of inflammation, innate immunity and defense against intracellular pathogens. For instance, the LUBAC is reportedly essential for NLRP3 inflammasome activation[27] and thus acts as a key innate immune regulator[28]. In turn, the NLRP3 inflammasome is essential for the development and progression of Aβ pathology in mice[29] and may drive tau pathology through Aβ-induced microglial activation[30]. The LUBAC is also reportedly involved in autophagy, and linear ubiquitin chain modifications of TDP-43-positive neuronal cytoplasmic inclusions have been described as potential inducers of autophagic clearance[31]. Lastly, the LUBAC has been studied as a regulator of TNF-α signaling in particular[20].

Interestingly, the TNF-α signaling pathway was also flagged by other genetic findings in our study (Supplementary Fig. 48). For example, ADAM17 (also known as TNF-α-converting enzyme) is of pivotal importance in the activation of TNF-α signaling[32]. For *TNIP1*, its gene product (TNF-α-induced protein 3-interacting protein 1) is involved in the inhibition of the TNF-α signaling pathway and nuclear factor κB activation/translocation[33]. Additional signal related to TNF-α is the one found at *SPPL2A* (one of the 33 confirmed loci). The protein encoded by *SPPL2A* is involved in non-canonical shedding of TNF-α[34], and PGRN has been described as a TNF receptor ligand and an antagonist of TNF-α signaling[35]. Several lines of evidence had linked the inhibition of TNF-α signaling with reduction of both Aβ and tau pathologies in vivo[36,37]. Although a potential inflammatory connection has been suggested for TNF-α through the activation of NLRP3 inflammasome[38], the TNF-α signaling pathway is also involved in many other brain physiological functions (e.g., synaptic plasticity in neurons) and pathophysiological processes (e.g., synapse loss) in the brain[39]. Furthermore, the involvement of the TNF-α signaling pathway and the LUBAC might be important in cell types other than microglia in AD. It is important

---

**Fig. 4 | Focus on *TSPAN14* locus. a,** Splicing QTL (sQTL)-GWAS integration results. Known *TSPAN14* transcripts (GENCODE v38; green, coding sequences; gray, noncoding) plotted with $-\log_{10}(P)$ for (1) EADB GWAS stage I ($n = 487,511$) signal (black), (2) sQTL signal for chr10:80509471–80510106 junction (supporting cryptic exon 1) in the EADB Belgian LCL sQTL catalog ($n = 70$ individuals, blue) and (3) sQTL signal for chr10:80512269–80512719 junction in the MayoRNAseq TCX sQTL catalog ($n = 259$ individuals, red); hg38 genomic position is shown above. LCL and brain-based sQTL coloc and sTWAS analyses associate ADD risk with these junctions that suggest cryptic splicing within ADAM10-interacting domain of *TSPAN14* (magenta), which was predicted to result in two cryptic exons. **b,** Long-read sequencing validation of *TSPAN14* cryptic exons. Nanopore sequencing results (Supplementary Note) in the zoomed-in region of chr10:80506973–80516400 (cumulative coverage in $\log_{10}$ scale). Pooled LCL cDNA sample sequenced for cDNA Amplicon2 shown in blue. cDNA Amplicon1 was sequenced on biologically independent hippocampal (HPC; $n = 16$, red), frontal cortex (FC; $n = 18$, pink) and LCL ($n = 59$, orange) cDNA samples. Green, canonical exons (8–12); dotted black lines, canonical splicing; blue, cryptic exon 1 (>45 bp); red, cryptic exon 2 (118 bp). All annotated junctions use canonical splice donor (GT) and acceptor (AG) sites. **c,d,** sQTL-GWAS colocalization plots for chr10:80509471–80510106 (supporting cryptic exon 1) in the EADB Belgian LCL sQTL catalog ($n = 70$ individuals) (**c**) and chr10:80512269–80512719 (supporting cryptic exon 2) in the MayoRNAseq TCX sQTL catalog ($n = 259$ individuals) (**d**). sQTL signals for the two junctions colocalize with ADD signal (PP4s of 98.8% and 97.4%, respectively), and sTWAS associates with increased preference for the cryptic splicing with decreased ADD risk (sTWAS $P = 6.28 \times 10^{-12}$ and $1.6 \times 10^{-13}$, sTWAS $Z = -6.9$ and $-7.4$, respectively). $y$ axis, sQTL $-\log_{10}(P)$; $x$ axis, EADB GWAS stage I $-\log_{10}(P)$. LD $r^2$ values calculated within EADB-TOPMed dataset ($n = 42,140$) based on the lead variant rs6586028 (purple) are indicated on a color scale.

to note that six of our prioritized (tier 1) genes (*ICA1L*, *EGFR*, *RITA1*, *MYO15A*, *LIME1* and *APP*) are expressed at a low level in microglia (<10%, relative to the total expression summed across cell types; Supplementary Table 45), emphasizing that ADD results from complex crosstalk between different cell types in the brain[23,40]. It is also noteworthy that the EGFR pathway is known to interact with the TNF-α signaling pathway[41], which suggests interplay between the two signaling pathways during the ADD development.

A better understanding of the etiology of ADD might also result from the observation that the risks of developing ADD and fronto-temporal dementia are associated with the same causal variants in *GRN* and *TMEM106B*. This association might be due to the misclassification of clinical diagnosis of AD and the presence of proxy-ADD cases in the UKBB. However, *GRN* and *TMEM106B* have also been linked to brain health and many other neurodegenerative diseases. For instance, *GRN* and *TMEM106B* are reportedly potential genetic

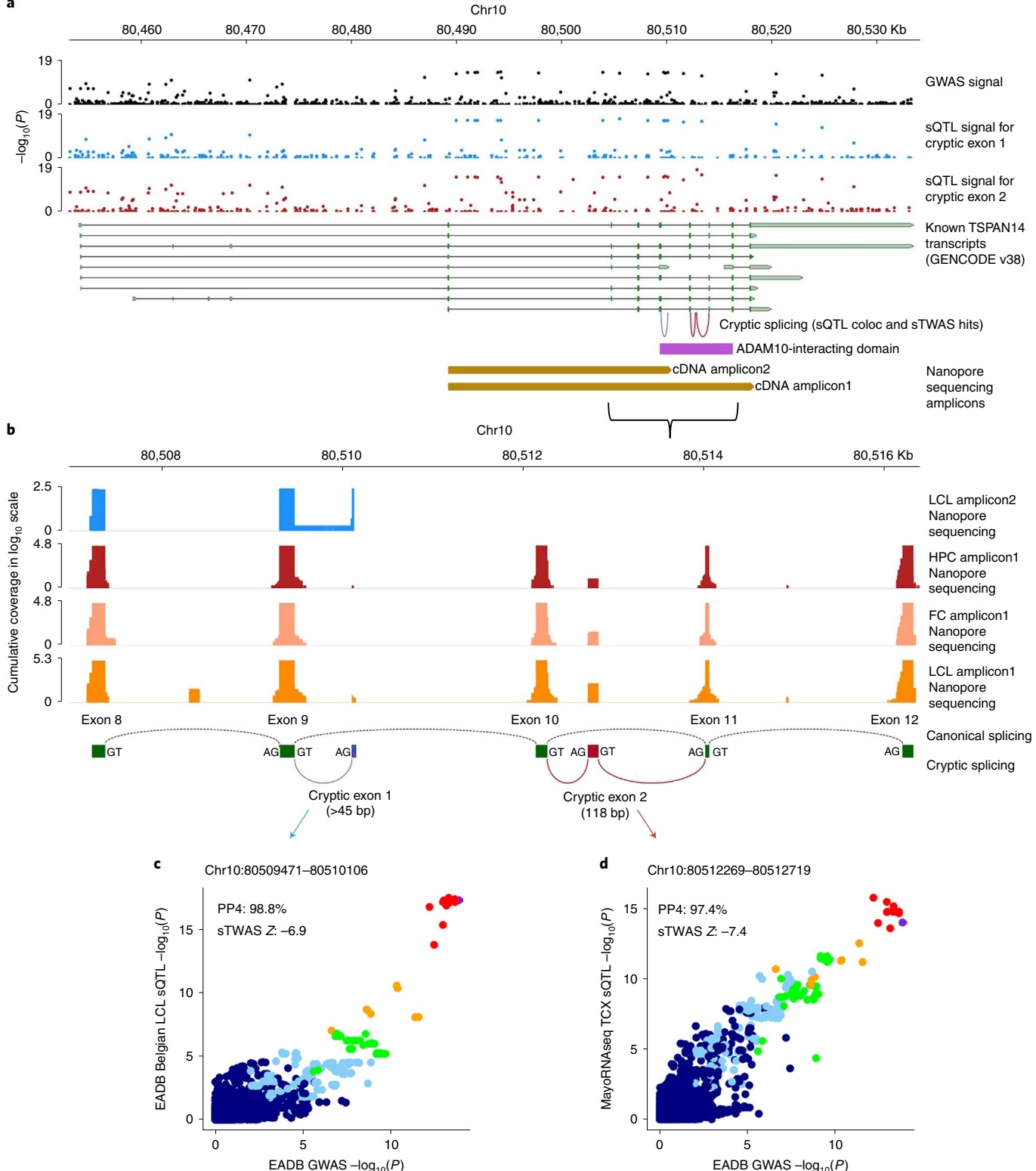

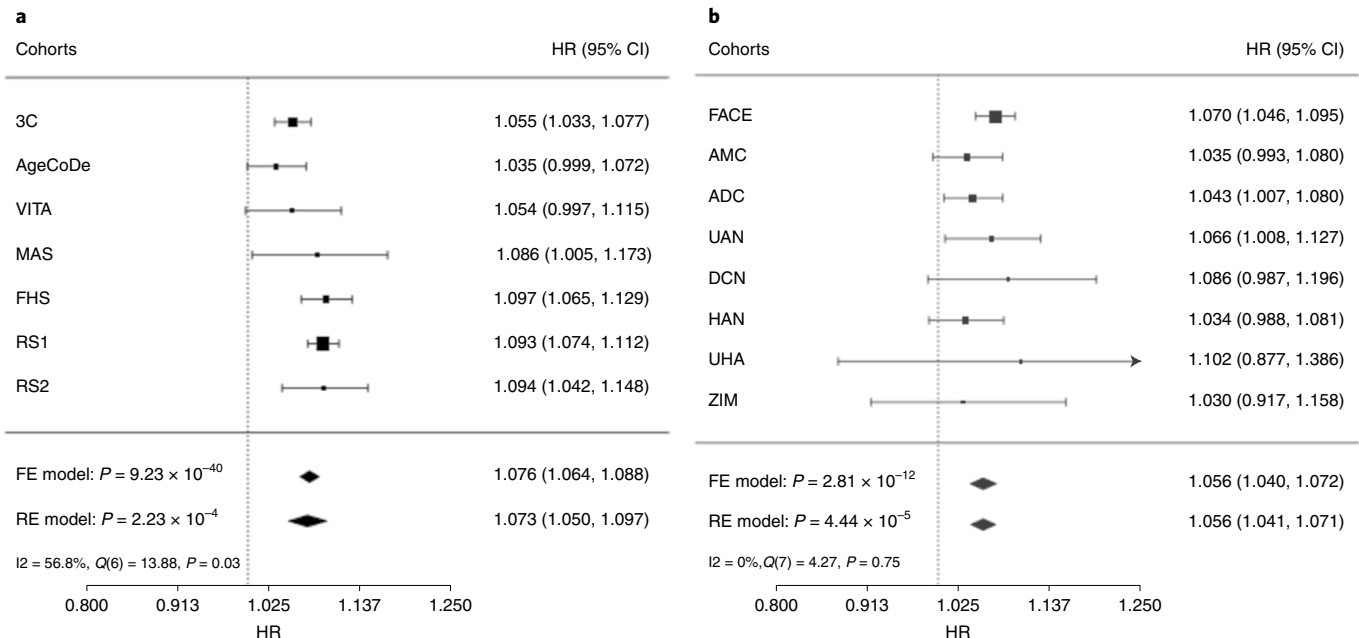

**Fig. 5 | Association between the GRS and the risk of progression to AD. a,b,** Meta-analysis results of the association between the GRS and the risk of progression to AD in population-based cohorts (n = 17,545 independent samples) (**a**) and MCI cohorts (n = 4,114 independent samples) (**b**). Data are presented as HR together with 95% CIs derived from Cox regression analyses for each individual cohort. HRs indicate the effect of the GRS as the increment in the AD risk associated with each additional average risk allele in the GRS. Null hypothesis testing is based on a meta-analysis of individual cohort effects using fixed effects (FE) and random effects (RE) models. Resulting HRs and 95% CIs and the respective Z test and associated two-sided P value are shown at the bottom of the figure. Heterogeneity between cohorts is indicated by the I2 index together with the respective Cochran's Q statistic (distributed as $\chi^2$ statistic), associated degrees of freedom and P value. 3C, Three-City Study; AgeCoDe, German study on aging cognition and dementia; AMC, additional, independent memory clinic cohort from Fundacio ACE; DCN, German Dementia Competence Network study; FACE, Fundacio ACE memory clinic cohort; FHS, Framingham Heart Study; HAN, BALTAZAR multicenter prospective memory clinic study; MAS, Sydney Memory and Ageing Study; RS1, Rotterdam Study first cohort; RS2, Rotterdam Study second cohort; VITA, Vienna Transdanube Aging study; UAN, memory clinic cohort from the Hospital Network Antwerp; UHA, University of Halle memory clinic cohort; ZIM, Heidelberg/Mannheim memory clinic sample.

risk factors for differential aging in the cerebral cortex[42] and cognitive impairment in amyotrophic lateral sclerosis[43] and Parkinson's disease[44,45]. Lastly, both *GRN* and *TMEM106B* have already been associated with neuropathological features of AD[46–48]. Taken as a whole, these data may thus emphasize a potential continuum between neurodegenerative diseases in which common pathological mechanisms are driven by *GRN* and *TMEM106B*. Interestingly, both *GRN* and *TMEM106B* are reported to be involved in defective endosome/lysosome trafficking/function[49,50], a defect that is also observed in AD.

By applying a GRS derived from all the genome-wide-significant variants discovered in this study, we identified an association with the risk of incident AD in prospective population-based cohorts and with the risk of progression over time from MCI to AD (Fig. 5 and Supplementary Table 33). In patients with MCI, previous associations of AD risk with a GRS built on previously known genetic AD risk variants has been inconsistent[51]. It is important to note that the GRS has an impact on the AD risk in addition to that of age and that the GRS's effect is independent of *APOE* status. With a view to translating genetic findings into preventive measures and personalized medicine, we also sought to provide the GRS's added value for risk prediction by calculating the discriminative capacity through three different indices. Overall, the indices suggested that the effect size for the association between the GRS and AD was small but significant. Despite this modest effect, the inclusion of the GRS into the predictive model consistently improved the assignment of the risk of progression, as expressed by the net reclassification improvement (NRI) index[21]. Importantly, the cumulative improvements in risk prediction (due to inclusion of the new variants in the GRS) led

to a 1.6- to 1.9-fold increase in the AD risk from the lowest to the highest decile, in addition to the effects of age and *APOE* status. We also showed that in addition to known risk variants, the new risk variants identified in the present study are significantly associated with progression to AD. The results of future GWASs are expected to further improve AD-risk prediction. Hence, the GRS will help to sharpen the threshold that differentiates between people at risk of progressing to dementia and those who are not.

A recent study estimated that fewer than 100 causal common variants may explain the entire AD risk[52]; if that estimate is correct, then our study might have already characterized a large proportion of this genetic component due to common variants. However, several reasons strongly underscore the need for additional efforts to fully characterize the still-missing AD genetic component. First, it is probable that additional, yet-unknown loci bear common variants modulating the risk for AD. Second, identification of rare variants with very low frequencies is a major challenge for genetic studies, because available samples with sequencing data in AD are underpowered. Notably, almost all the genes with rare variants associated with AD risk also present common variants associated with AD risk (i.e., *TREM2*, *SORL1*, *ABCA7*, *ABCA1*, *PLCγ2* and *ADAM10*)[53]. Third, gene–gene and gene–environment interactions have not yet been studied in detail. Hence, by increasing the GWAS sample size and improving imputation panels, conventional and (above all) more complex analyses will have more statistical power and should enable the characterization of associations with rare/structural variants. Lastly, higher-powered GWASs of multiancestry populations will be particularly welcome for characterizing potential new genetic risk factors, improving fine-mapping approaches and developing

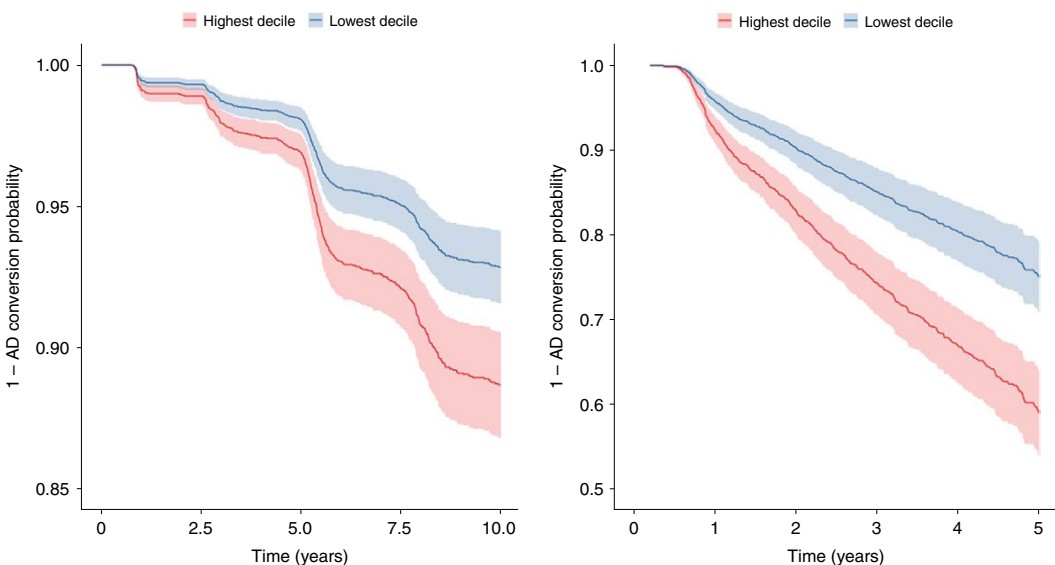

**Fig. 6 | Risk of progression to AD according to the GRS. a,b,** Representative plots of the progression to AD over 10 years in the population-based 3C study (**a**) and the progression from MCI to AD over 5 years in the Fundació ACE cohort (**b**). The figures show the probabilities of conversion (survival probabilities) to AD (*y* axes) for a hypothetical participant with average covariates (mean values for age and PCs, and the mode for sex and *APOE*) and a GRS at the first (lowest) decile (in blue) or a GRS at the ninth (highest) decile (red). The shaded regions correspond to the 95% CI.

specific GRSs (because GRSs developed with European-ancestry populations are known to be less effective with other ancestries).

In conclusion, we have validated 33 previous loci, doubled the total number of genetic loci associated with the ADD risk, expanded our current knowledge of the pathophysiology of ADD, identified new opportunities for the development of GRSs and gene-specific treatments and opened up a pathway to translational genomics and personalized medicine.

## Online content

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

Céline Bellenguez[1,567 ✉], Fahri Küçükali[2,3,4,567], Iris E. Jansen[5,6,567], Luca Kleineidam[7,8,9,567], Sonia Moreno-Grau[10,11,567], Najaf Amin[12,13,567], Adam C. Naj[14,15,567], Rafael Campos-Martin[8,567], Benjamin Grenier-Boley[1], Victor Andrade[7,8], Peter A. Holmans[16], Anne Boland[17], Vincent Damotte[1], Sven J. van der Lee[5,18], Marcos R. Costa[1,19], Teemu Kuulasmaa[20], Qiong Yang[21,22], Itziar de Rojas[10,11], Joshua C. Bis[23], Amber Yaqub[12], Ivana Prokic[12], Julien Chapuis[1], Shahzad Ahmad[12,24], Vilmantas Giedraitis[25], Dag Aarsland[26,27], Pablo Garcia-Gonzalez[10,11], Carla Abdelnour[10,11], Emilio Alarcón-Martín[10,28], Daniel Alcolea[11,29], Montserrat Alegret[10,11], Ignacio Alvarez[30,31], Victoria Álvarez[32,33], Nicola J. Armstrong[34], Anthoula Tsolaki[35,36], Carmen Antúnez[37], Ildebrando Appollonio[38,39], Marina Arcaro[40], Silvana Archetti[41], Alfonso Arias Pastor[42,43], Beatrice Arosio[44,45], Lavinia Athanasiu[46], Henri Bailly[47], Nerisa Banaj[48], Miquel Baquero[49], Sandra Barral[50,51,52], Alexa Beiser[20,53], Ana Belén Pastor[54], Jennifer E. Below[55], Penelope Benchek[56,57], Luisa Benussi[58], Claudine Berr[59], Céline Besse[17], Valentina Bessi[60,61], Giuliano Binetti[58,62], Alessandra Bizarro[63], Rafael Blesa[11,29], Mercè Boada[10,11], Eric Boerwinkle[64,65], Barbara Borroni[66], Silvia Boschi[67], Paola Bossù[68], Geir Bråthen[69,70], Jan Bressler[64,71], Catherine Bresner[16], Henry Brodaty[34,72], Keeley J. Brookes[73], Luis Ignacio Brusco[74,75,76], Dolores Buiza-Rueda[11,77], Katharina Bürger[78,79], Vanessa Burholt[80,81], William S. Bush[82], Miguel Calero[10,54,83], Laura B. Cantwell[84], Geneviève Chene[85,86], Jaeyoon Chung[87], Michael L. Cuccaro[88], Ángel Carracedo[89,90], Roberta Cecchetti[91], Laura Cervera-Carles[11,29], Camille Charbonnier[92], Hung-Hsin Chen[93], Caterina Chillotti[94], Simona Ciccone[45],

Jurgen A. H. R. Claassen[95], Christopher Clark[96], Elisa Conti[38], Anaïs Corma-Gómez[97], Emanuele Costantini[98], Carlo Custodero[99], Delphine Daian[17], Maria Carolina Dalmasso[8], Antonio Daniele[98], Efthimios Dardiotis[100], Jean-François Dartigues[101], Peter Paul de Deyn[102], Katia de Paiva Lopes[103,104,105,106], Lot D. de Witte[106], Stéphanie Debette[101], Jürgen Deckert[107], Teodoro del Ser[54], Nicola Denning[108], Anita DeStefano[20,21,109], Martin Dichgans[78,110], Janine Diehl-Schmid[111], Mónica Diez-Fairen[30,31], Paolo Dionigi Rossi[45], Srdjan Djurovic[46], Emmanuelle Duron[47], Emrah Düzel[112,113], Carole Dufouil[85,86], Gudny Eiriksdottir[114], Sebastiaan Engelborghs[115,116,117,118], Valentina Escott-Price[15,108], Ana Espinosa[10,11], Michael Ewers[78,79], Kelley M. Faber[119], Tagliavini Fabrizio[120], Sune Fallgaard Nielsen[121], David W. Fardo[122], Lucia Farotti[123], Chiara Fenoglio[124], Marta Fernández-Fuertes[97], Raffaele Ferrari[125,126], Catarina B. Ferreira[127], Evelyn Ferri[45], Bertrand Fin[17], Peter Fischer[128], Tormod Fladby[129], Klaus Fließbach[8,9], Bernard Fongang[130], Myriam Fornage[70,71], Juan Fortea[11,29], Tatiana M. Foroud[119], Silvia Fostinelli[58], Nick C. Fox[131], Emlio Franco-Macías[132], María J. Bullido[11,133,134], Ana Frank-García[11,133,135], Lutz Froelich[136], Brian Fulton-Howard[137], Daniela Galimberti[40,124], Jose Maria García-Alberca[11,138], Pablo García-González[10], Sebastian Garcia-Madrona[139], Guillermo Garcia-Ribas[139], Roberta Ghidoni[58], Ina Giegling[140], Giaccone Giorgio[120], Alison M. Goate[137], Oliver Goldhardt[111], Duber Gomez-Fonseca[141], Antonio González-Pérez[142], Caroline Graff[143,144], Giulia Grande[145], Emma Green[146], Timo Grimmer[111], Edna Grünblatt[147,148,149], Michelle Grunin[57], Vilmundur Gudnason[150], Tamar Guetta-Baranes[151], Annakaisa Haapasalo[152], Georgios Hadjigeorgiou[153], Jonathan L. Haines[82], Kara L. Hamilton-Nelson[154], Harald Hampel[155], Olivier Hanon[47], John Hardy[126], Annette M. Hartmann[140], Lucrezia Hausner[136], Janet Harwood[16], Stefanie Heilmann-Heimbach[156], Seppo Helisalmi[157,158], Michael T. Heneka[7,9], Isabel Hernández[10,11], Martin J. Herrmann[107], Per Hoffmann[156], Clive Holmes[159], Henne Holstege[5,18], Raquel Huerto Vilas[42,43], Marc Hulsman[5,18], Jack Humphrey[103,104,105,160], Geert Jan Biessels[161], Xueqiu Jian[130], Charlotte Johansson[143], Gyungah R. Jun[87], Yuriko Kastumata[162], John Kauwe[163], Patrick G. Kehoe[164], Lena Kilander[21], Anne Kinhult Ståhlbom[143], Miia Kivipelto[165,166,167,168], Anne Koivisto[157,169,170], Johannes Kornhuber[171], Mary H. Kosmidis[172], Walter A. Kukull[173], Pavel P. Kuksa[15], Brian W. Kunkle[153], Amanda B. Kuzma[84], Carmen Lage[11,174], Erika J. Laukka[145,175], Lenore Launer[176,177], Alessandra Lauria[63], Chien-Yueh Lee[15], Jenni Lehtisalo[157,178], Ondrej Lerch[179,180], Alberto Lleó[11,29], William Longstreth Jr[181], Oscar Lopez[22], Adolfo Lopez de Munain[11,182], Seth Love[164], Malin Löwemark[21], Lauren Luckcuck[16], Kathryn L. Lunetta[20], Yiyi Ma[18,183], Juan Macías[97], Catherine A. MacLeod[184], Wolfgang Maier[7,9], Francesca Mangialasche[165], Marco Spallazzi[51], Marta Marquié[10,11], Rachel Marshall[16], Eden R. Martin[154], Angel Martín Montes[11,133,135], Carmen Martínez Rodríguez[33], Carlo Masullo[185], Richard Mayeux[50,186], Simon Mead[187], Patrizia Mecocci[91], Miguel Medina[11,54], Alun Meggy[108], Shima Mehrabian[52], Silvia Mendoza[138], Manuel Menéndez-González[33], Pablo Mir[11,188], Susanne Moebus[189], Merel Mol[77], Laura Molina-Porcel[190,191], Laura Montrreal[10], Laura Morelli[192], Fermin Moreno[11,182], Kevin Morgan[193], Thomas Mosley[194], Markus M. Nöthen[156], Carolina Muchnik[74,195], Shubhabrata Mukherjee[196], Benedetta Nacmias[60,197], Tiia Ngandu[178], Gael Nicolas[92], Børge G. Nordestgaard[121,198], Robert Olaso[17], Adelina Orellana[10,11], Michela Orsini[98], Gemma Ortega[10,11], Alessandro Padovani[65], Caffarra Paolo[199], Goran Papenberg[145], Lucilla Parnetti[123], Florence Pasquier[200], Pau Pastor[30,31], Gina Peloso[20,53], Alba Pérez-Cordón[10], Jordi Pérez-Tur[11,201,202], Pierre Pericard[203], Oliver Peters[204,205], Yolande A. L. Pijnenburg[5], Juan A. Pineda[97], Gerard Piñol-Ripoll[42,43],

Claudia Pisanu [206], Thomas Polak[107], Julius Popp[207,208,209], Danielle Posthuma [6], Josef Priller[205,210], Raquel Puerta [10], Olivier Quenez[92], Inés Quintela[89], Jesper Qvist Thomassen [211], Alberto Rábano[11,54], Innocenzo Rainero[66], Farid Rajabli[154], Inez Ramakers[212], Luis M. Real [97,213], Marcel J. T. Reinders [214], Christiane Reitz [186,214,215], Dolly Reyes-Dumeyer[183,215], Perry Ridge[216], Steffi Riedel-Heller [217], Peter Riederer[218], Natalia Roberto[10], Eloy Rodriguez-Rodriguez [11,174], Arvid Rongve [219,220], Irene Rosas Allende[32,33], Maitée Rosende-Roca[10,11], Jose Luis Royo[221], Elisa Rubino[222], Dan Rujescu[140], María Eugenia Sáez [142], Paraskevi Sakka[223], Ingvild Saltvedt [69,224], Ángela Sanabria[10,11], María Bernal Sánchez-Arjona[132], Florentino Sanchez-Garcia[225], Pascual Sánchez Juan [11,174], Raquel Sánchez-Valle[226], Sigrid B. Sando[68,69], Chloé Sarnowski [64], Claudia L. Satizabal[21,109,130], Michela Scamosci[91], Nikolaos Scarmeas[50,227], Elio Scarpini[40,124], Philip Scheltens[5], Norbert Scherbaum[228], Martin Scherer[229], Matthias Schmid[9,230], Anja Schneider[7,9], Jonathan M. Schott [131], Geir Selbæk [129,231], Davide Seripa[232], Manuel Serrano[233], Jin Sha[14], Alexey A. Shadrin [46], Olivia Skrobot[164], Susan Slifer[154], Gijsje J. L. Snijders[106], Hilkka Soininen [157], Vincenzo Solfrizzi [99], Alina Solomon[157,165], Yeunjoo Song [57], Sandro Sorbi[60,197], Oscar Sotolongo-Grau [10], Gianfranco Spalletta [48], Annika Spottke[9,234], Alessio Squassina [235], Eystein Stordal [236], Juan Pablo Tartan[10], Lluís Tárraga[10,11], Niccolo Tesí[5,18], Anbupalam Thalamuthu[34], Tegos Thomas[35,36], Giuseppe Tosto [50,183], Latchezar Traykov[52], Lucio Tremolizzo[38,39], Anne Tybjærg-Hansen[198,211], Andre Uitterlinden [237], Abbe Ullgren[143], Ingun Ulstein[231], Sergi Valero[10,11], Otto Valladares[15], Christine Van Broeckhoven [2,3,238], Jeffery Vance[88], Badri N. Vardarajan[50], Aad van der Lugt[239], Jasper Van Dongen[2,3,4], Jeroen van Rooij[77,239], John van Swieten[77], Rik Vandenberghe[240,241], Frans Verhey[212], Jean-Sébastien Vidal [47], Jonathan Vogelgsang [242,243], Martin Vyhnalek[179,180], Michael Wagner [7,9], David Wallon [244], Li-San Wang[15], Ruiqi Wang[20,21], Leonie Weinhold[230], Jens Wiltfang [242,245,246], Gill Windle[184], Bob Woods [184], Mary Yannakoulia [247], Habil Zare [130], Yi Zhao[15], Xiaoling Zhang [248], Congcong Zhu[248], Miren Zulaica[11,249], EADB, GR@ACE, DEGESCO, EADI, GERAD, Demgene, FinnGen, ADGC, CHARGE, Lindsay A. Farrer [20,87,109], Bruce M. Psaty [22,84,250], Mohsen Ghanbari [12], Towfique Raj [103,104,105,160], Perminder Sachdev [34], Karen Mather [34], Frank Jessen[7,9], M. Arfan Ikram [12], Alexandre de Mendonça[127], Jakub Hort[177,179], Magda Tsolaki [35,36], Margaret A. Pericak-Vance [152], Philippe Amouyel [1], Julie Williams [16,108], Ruth Frikke-Schmidt [198,211], Jordi Clarimon[11,29], Jean-François Deleuze[17], Giacomina Rossi[120], Sudha Seshadri [21,109,130], Ole A. Andreassen [46], Martin Ingelsson[25], Mikko Hiltunen[19,568], Kristel Sleegers [2,3,4,568], Gerard D. Schellenberg[15,568], Cornelia M. van Duijn [12,13,568], Rebecca Sims [16,568], Wiesje M. van der Flier [5,568], Agustín Ruiz[10,11,568], Alfredo Ramirez [7,8,9,130,251,568] and Jean-Charles Lambert [1,568] ✉

[1]Université de Lille, INSERM, CHU Lille, Institut Pasteur Lille, U1167-RID-AGE, Facteurs de risque et déterminants moléculaires des maladies liées au vieillissement, Lille, France. [2]Complex Genetics of Alzheimer's Disease Group, VIB Center for Molecular Neurology, VIB, Antwerp, Belgium. [3]Laboratory of Neurogenetics, Institute Born - Bunge, Antwerp, Belgium. [4]Department of Biomedical Sciences, University of Antwerp, Antwerp, Belgium. [5]Alzheimer Center Amsterdam, Department of Neurology, Amsterdam Neuroscience, Vrije Universiteit Amsterdam, Amsterdam UMC, Amsterdam, the Netherlands. [6]Department of Complex Trait Genetics, Center for Neurogenomics and Cognitive Research, Amsterdam Neuroscience, Vrije University, Amsterdam, the Netherlands. [7]Department of Neurodegenerative Diseases and Geriatric Psychiatry, University Hospital Bonn, Bonn, Germany. [8]Division of Neurogenetics and Molecular Psychiatry, Department of Psychiatry and Psychotherapy, University of Cologne, Medical Faculty, Cologne, Germany. [9]German Center for Neurodegenerative Diseases (DZNE Bonn), Bonn, Germany. [10]Research Center and Memory Clinic Fundació ACE, Institut Català de Neurociències Aplicades, Universitat Internacional de Catalunya, Barcelona, Spain. [11]CIBERNED, Network Center for Biomedical Research in Neurodegenerative Diseases, National Institute of Health Carlos III, Madrid, Spain. [12]Department of Epidemiology, Erasmus MC, Rotterdam, the Netherlands. [13]Nuffield Department of Population Health, Oxford University, Oxford, UK. [14]Department of Biostatistics, Epidemiology, and Informatics, Penn Neurodegeneration Genomics Center, University of Pennsylvania Perelman School of Medicine, Philadelphia, PA, USA. [15]Department of Pathology and Laboratory Medicine, University of Pennsylvania Perelman School of Medicine, Philadelphia, PA, USA. [16]MRC Centre for Neuropsychiatric Genetics and Genomics, Division of Psychological Medicine and Clinical Neuroscience, School of Medicine, Cardiff University, Cardiff, UK. [17]CEA, Centre National de Recherche en Génomique Humaine,

Université Paris-Saclay, Evry, France. [18]Section Genomics of Neurodegenerative Diseases and Aging, Department of Human Genetics Amsterdam UMC, Vrije Universiteit Amsterdam, Amsterdam UMC, Amsterdam, the Netherlands. [19]Brain Institute, Federal University of Rio Grande do Norte, Natal, Brazil. [20]Institute of Biomedicine, University of Eastern Finland, Kuopio, Finland. [21]Department of Biostatistics, Boston University School of Public Health, Boston, MA, USA. [22]Framingham Heart Study, Framingham, MA, USA. [23]Cardiovascular Health Research Unit, Department of Medicine, University of Washington, Seattle, WA, USA. [24]LACDR, Leiden, the Netherlands. [25]Department of Public Health and Carins Sciences/Geriatrics, Uppsala University, Uppsala, Sweden. [26]Centre of Age-Related Medicine, Stavanger University Hospital, Stavanger, Norway. [27]Institute of Psychiatry, Psychology & Neuroscience, London, UK. [28]Department of Surgery, Biochemistry and Molecular Biology, School of Medicine, University of Málaga, Málaga, Spain. [29]Department of Neurology, II B Sant Pau, Hospital de la Santa Creu i Sant Pau, Universitat Autònoma de Barcelona, Barcelona, Spain. [30]Fundació Docència i Recerca MútuaTerrassa and Movement Disorders Unit, Department of Neurology, University Hospital MútuaTerrassa, Terrassa, Spain. [31]Memory Disorders Unit, Department of Neurology, Hospital Universitari Mutua de Terrassa, Terrassa, Spain. [32]Laboratorio de Genética, Hospital Universitario Central de Asturias, Oviedo, Spain. [33]Servicio de Neurología, Hospital Universitario Central de Asturias- Oviedo and Instituto de Investigación Biosanitaria del Principado de Asturias, Oviedo, Spain. [34]Centre for Healthy Brain Ageing, School of Psychiatry, Faculty of Medicine, University of New South Wales, Sydney, New South Wales, Australia. [35]First Department of Neurology, Medical School, Aristotle University of Thessaloniki, Thessaloniki, Greece. [36]Alzheimer Hellas, Thessaloniki, Greece. [37]Unidad de Demencias, Hospital Clínico Universitario Virgen de la Arrixaca, Murcia, Spain. [38]School of Medicine and Surgery, University of Milano-Bicocca, Milano, Italy. [39]Neurology Unit, San Gerardo Hospital, Monza, Italy. [40]Fondazione IRCCS Ca'Granda, Ospedale Policlinico, Milan, Italy. [41]Department of Laboratory Diagnostics, III Laboratory of Analysis, Brescia Hospital, Brescia, Italy. [42]Unitat Trastorns Cognitius, Hospital Universitari Santa Maria de Lleida, Lleida, Spain. [43]Institut de Recerca Biomedica de Lleida (IRBLleida), Lleida, Spain. [44]Department of Clinical Sciences and Community Health, University of Milan, Milan, Italy. [45]Geriatric Unit, Fondazione Cà Granda, IRCCS Ospedale Maggiore Policlinico, Milan, Italy. [46]NORMENT Centre, University of Oslo, Oslo, Norway. [47]EA 4468, Université de Paris, APHP, Hôpital Broca, Paris, France. [48]Laboratory of Neuropsychiatry, Department of Clinical and Behavioral Neurology, IRCCS Santa Lucia Foundation, Rome, Italy. [49]Servei de Neurologia, Hospital Universitari i Politècnic La Fe, Valencia, Spain. [50]Taub Institute on Alzheimer's Disease and the Aging Brain, Department of Neurology, Columbia University, New York, NY, USA. [51]Unit of Neurology, University of Parma and AOU, Parma, Italy. [52]Clinic of Neurology, UH 'Alexandrovska', Medical University - Sofia, Sofia, Bulgaria. [53]Boston University and the NHLBI's Framingham Heart Study, Boston, MA, USA. [54]CIEN Foundation/Queen Sofia Foundation Alzheimer Center, Madrid, Spain. [55]Vanderbilt Brain Institute, Vanderbilt University, Nashville, TN, USA. [56]Cleveland Institute for Computational Biology, Case Western Reserve University, Cleveland, OH, USA. [57]Department of Population and Quantitative Health Sciences, Case Western Reserve University, Cleveland, OH, USA. [58]Molecular Markers Laboratory, IRCCS Istituto Centro San Giovanni di Dio Fatebenefratelli, Brescia, Italy. [59]Neuropsychiatry: Epidemiological and Clinical Research, PSNREC, Université de Montpellier, INSERM U1061, Montpellier, France. [60]Department of Neuroscience, Psychology, Drug Research and Child Health, University of Florence, Florence, Italy. [61]Azienda Ospedaliero-Universitaria Careggi, Florence, Italy. [62]MAC - Memory Clinic, IRCCS Istituto Centro San Giovanni di Dio Fatebenefratelli, Brescia, Italy. [63]Geriatrics Unit, Fondazione Policlinico A. Gemelli IRCCS, Rome, Italy. [64]Human Genetics Center, School of Public Health, University of Texas Health Science Center at Houston, Houston, TX, USA. [65]Human Genome Sequencing Center, Baylor College of Medicine, Houston, TX, USA. [66]Centre for Neurodegenerative Disorders, Department of Clinical and Experimental Sciences, University of Brescia, Brescia, Italy. [67]Department of Neuroscience "Rita Levi Montalcini", University of Torino, Torino, Italy. [68]Experimental Neuro-psychobiology Laboratory, Department of Clinical and Behavioral Neurology, IRCCS Santa Lucia Foundation, Rome, Italy. [69]Department of Neurology and Clinical Neurophysiology, University Hospital of Trondheim, Trondheim, Norway. [70]Department of Neuromedicine and Movement Science, Norwegian University of Science and Technology, Trondheim, Norway. [71]School of Public Health, University of Texas Health Science Center at Houston, Houston, TX, USA. [72]Dementia Centre for Research Collaboration, School of Psychiatry, University of New South Wales, Sydney, New South Wales, Australia. [73]Biosciences, School of Science and Technology, Nottingham Trent University, Nottingham, UK. [74]Centro de Neuropsiquiatría y Neurología de la Conducta (CENECON), Facultad de Medicina, Universidad de Buenos Aires (UBA), C.A.B.A., Buenos Aires, Argentina. [75]Departamento Ciencias Fisiológicas UAII, Facultad de Medicina, UBA, C.A.B.A., Buenos Aires, Argentina. [76]Hospital Interzonal General de Agudos Eva Perón, San Martín, Buenos Aires, Argentina. [77]Department of Neurology, Erasmus MC, Rotterdam, the Netherlands. [78]Institute for Stroke and Dementia Research, Klinikum der Universität München, Ludwig Maximilians Universität (LMU), Munich, Germany. [79]German Center for Neurodegenerative Diseases (DZNE, Munich), Munich, Germany. [80]Faculty of Medical & Health Sciences, University of Auckland, Auckland, New Zealand. [81]Wales Centre for Ageing & Dementia Research, Swansea University, Wales, New Zealand. [82]Department of Population & Quantitative Health Sciences, Case Western Reserve University, Cleveland, OH, USA. [83]UFIEC, Instituto de Salud Carlos III, Madrid, Spain. [84]Department of Pathology and Laboratory Medicine, University of Pennsylvania, Philadelphia, PA, USA. [85]INSERM, Bordeaux Population Health Research Center, UMR 1219, ISPED, CIC 1401-EC, Université de Bordeaux, Bordeaux, France. [86]Pole Santé Publique, CHU de Bordeaux, Bordeaux, France. [87]Medicine Biomedical Genetics Boston University School of Medicine, Boston, MA, USA. [88]Dr. John T. Macdonald Foundation Department of Human Genetics, University of Miami, Miami, FL, USA. [89]Grupo de Medicina Xenómica, Centro Nacional de Genotipado (CEGEN-PRB3-ISCIII), Universidade de Santiago de Compostela, Santiago de Compostela, Spain. [90]Fundación Pública Galega de Medicina Xenómica- CIBERER-IDIS, University of Santiago de Compostela, Santiago de Compostela, Spain. [91]Institute of Gerontology and Geriatrics, Department of Medicine and Surgery, University of Perugia, Perugia, Italy. [92]Department of Genetics and CNR-MAJ, Normandie University, UNIROUEN, INSERM U1245, CHU Rouen, Rouen, France. [93]Division of Genetic Medicine, Vanderbilt University, Nashville, TN, USA. [94]Unit of Clinical Pharmacology, University Hospital of Cagliari, Cagliari, Italy. [95]Radboudumc Alzheimer Center, Department of Geriatrics, Radboud University Medical Center, Nijmegen, the Netherlands. [96]Institute for Regenerative Medicine, University of Zürich, Schlieren, Switzerland. [97]Unidad Clínica de Enfermedades Infecciosas y Microbiología, Hospital Universitario de Valme, Sevilla, Spain. [98]Department of Neuroscience, Catholic University of Sacred Heart, Fondazione Policlinico Universitario A. Gemelli IRCCS, Rome, Italy. [99]University of Bari, "A. Moro", Bary, Italy. [100]School of Medicine, University of Thessaly, Larissa, Greece. [101]Bordeaux Population Health Research Center, University Bordeaux, INSERM, Bordeaux, France. [102]Department of Neurology, University Medical Center Groningen, Groningen, the Netherlands. [103]Ronald M. Loeb Center for Alzheimer's Disease, Icahn School of Medicine at Mount Sinai, New York, NY, USA. [104]Department of Genetics and Genomic Sciences & Icahn Institute for Data Science and Genomic Technology, Icahn School of Medicine at Mount Sinai, New York, NY, USA. [105]Estelle and Daniel Maggin Department of Neurology, Icahn School of Medicine at Mount Sinai, New York, NY, USA. [106]Department of Psychiatry, Icahn School of Medicine at Mount Sinai, New York, NY, USA. [107]Department of Psychiatry, Psychosomatics and Psychotherapy, Center of Mental Health, University Hospital, Wuerzburg, Germany. [108]UKDRI@ Cardiff, School of Medicine, Cardiff University, Cardiff, UK. [109]Department of Neurology, Boston University School of Medicine, Boston, MA, USA. [110]Munich Cluster for Systems Neurology (SyNergy), Munich, Germany. [111]Klinikum rechts der Isar, Department of Psychiatry and Psychotherapy, Technical University of Munich, School of Medicine, Munich, Germany. [112]Institute of Cognitive Neurology and Dementia Research (IKND), Otto-Von-Guericke University, Magdeburg, Germany. [113]German Center for Neurodegenerative Diseases (DZNE), Magdeburg, Germany. [114]Icelandic Heart Association, Kopovagur, Iceland. [115]Center for Neurosciences, Vrije Universiteit Brussel (VUB), Brussels, Belgium. [116]Reference Center for Biological Markers of Dementia (BIODEM), Institute Born-Bunge, University of Antwerp, Antwerp, Belgium. [117]Institute Born-Bunge, University of Antwerp, Antwerp, Belgium. [118]Department of Neurology, UZ Brussel, Brussels, Belgium. [119]Department of Medical and Molecular Genetics, Indiana University, Indianapolis, IN, USA. [120]Fondazione IRCCS, Istituto Neurologico Carlo Besta, Milan, Italy. [121]Department of Clinical Biochemistry, Herlev and Gentofte Hospital, Herlev, Denmark. [122]Sanders-Brown Center on Aging, Department of Biostatistics, University of Kentucky, Lexington, KY, USA. [123]Centre for Memory Disturbances, Lab of

Clinical Neurochemistry, Section of Neurology, University of Perugia, Perugia, Italy. [124]University of Milan, Milan, Italy. [125]Laboratory of Neurogenetics, Department of Internal Medicine, Texas Tech University Health Science Center, Lubbock, TX, USA. [126]Reta Lila Weston Research Laboratories, Department of Molecular Neuroscience, UCL Institute of Neurology, London, UK. [127]Faculty of Medicine, University of Lisbon, Lisbon, Portugal. [128]Department of Psychiatry, Social Medicine Center East- Donauspital, Vienna, Austria. [129]Institute of Clinical Medicine, University of Oslo, Oslo, Norway. [130]Glenn Biggs Institute for Alzheimer's & Neurodegenerative Diseases, University of Texas Health Sciences Center, San Antonio, TX, USA. [131]Dementia Research Centre, UCL Queen Square Institute of Neurology, London, UK. [132]Unidad de Demencias, Servicio de Neurología y Neurofisiología. Instituto de Biomedicina de Sevilla (IBiS), Hospital Universitario Virgen del Rocío/CSIC/Universidad de Sevilla, Seville, Spain. [133]Instituto de Investigacion Sanitaria 'Hospital la Paz' (IdIPaz), Madrid, Spain. [134]Centro de Biología Molecular Severo Ochoa (UAM-CSIC), Madrid, Spain. [135]Hospital Universitario la Paz, Madrid, Spain. [136]Department of Geriatric Psychiatry, Central Institute for Mental Health, Mannheim, University of Heidelberg, Heidelberg, Germany. [137]Department of Genetics and Genomic Sciences, Ronald M. Loeb Center for Alzheimer's Disease Icahn School of Medicine at Mount Sinai, New York, NY, USA. [138]Alzheimer Research Center & Memory Clinic, Andalusian Institute for Neuroscience, Málaga, Spain. [139]Hospital Universitario Ramon y Cajal, IRYCIS, Madrid, Spain. [140]Department of Psychiatry and Psychotherapy, Medical University of Vienna, Vienna, Austria. [141]Department of Biostatistics, Epidemiology, and Informatics Perelman School of Medicine, University of Pennsylvania, Philadelphia, PA, USA. [142]CAEBI, Centro Andaluz de Estudios Bioinformáticos, Sevilla, Spain. [143]Center for Alzheimer Research, Department NVS, Division of Neurogeriatrics, Karolinska Institutet, Stockholm, Sweden. [144]Unit for Hereditary Dementias, Karolinska University Hospital-Solna, Stockholm, Sweden. [145]Aging Research Center, Department of Neurobiology, Care Sciences and Society, Karolinska Institutet and Stockholm University, Stockholm, Sweden. [146]Institute of Public Health, University of Cambridge, Cambridge, UK. [147]Department of Child and Adolescent Psychiatry and Psychotherapy, University Hospital of Psychiatry Zurich, University of Zurich, Zurich, Switzerland. [148]Neuroscience Center Zurich, University of Zurich and ETH Zurich, Zurich, Switzerland. [149]Zurich Center for Integrative Human Physiology, University of Zurich, Zurich, Switzerland. [150]Icelandic Heart Association, Faculty of Medicine, University of Iceland, Reykjavik, Iceland. [151]Human Genetics, School of Life Sciences, Life Sciences Building, University Park, University of Nottingham, Nottingham, UK. [152]AI Virtanen Institute for Molecular Sciences, University of Eastern Finland, Kuopio, Finland. [153]Department of Neurology, Medical School, University of Cyprus, Nicosia, Cyprus. [154]The John P. Hussman Institute for Human Genomics, University of Miami, Miami, FL, USA. [155]GRC 21, Alzheimer Precision Medicine Initiative (APMI), Sorbonne University, AP-HP, Pitié-Salpêtrière Hospital, Paris, France. [156]Institute of Human Genetics, University of Bonn, School of Medicine & University Hospital Bonn, Bonn, Germany. [157]Institute of Clinical Medicine, Neurology, University of Eastern, Kuopio, Finland. [158]Institute of Clinical Medicine, Internal Medicine, University of Eastern Finland, Kuopio, Finland. [159]Clinical and Experimental Science, Faculty of Medicine, University of Southampton, Southampton, UK. [160]Nash Family Department of Neuroscience & Friedman Brain Institute, Icahn School of Medicine at Mount Sinai, New York, NY, USA. [161]Department of Neurology, UMC Utrecht Brain Center, Utrecht, the Netherlands. [162]Biostatistics, University of Kentucky College of Public Health, Lexington, KY, USA. [163]Department of Biology, Brigham Young University, Provo, UT, USA. [164]Translational Health Sciences, Bristol Medical School, University of Bristol, Bristol, UK. [165]Division of Clinical Geriatrics, Center for Alzheimer Research, Care Sciences and Society (NVS), Karolinska Institutet, Stockholm, Sweden. [166]Institute of Public Health and Clinical Nutrition, University of Eastern Finland, Kuopio, Finland. [167]Neuroepidemiology and Ageing Research Unit, School of Public Health, Imperial College London, London, UK. [168]Research & Development, UnitStockholms Sjukhem, Stockholm, Sweden. [169]Department of Neurology, Kuopio University Hospital, Kuopio, Finland. [170]Department of Neurosciences, University of Helsinki and Department of Geriatrics, Helsinki University Hospital, Helsinki, Finland. [171]Department of Psychiatry and Psychotherapy, Universitätsklinikum Erlangen, and Friedrich-Alexander Universität Erlangen-Nürnberg, Erlangen, Germany. [172]Laboratory of Cognitive Neuroscience, School of Psychology, Aristotle University of Thessaloniki, Thessaloniki, Greece. [173]Department of Epidemiology, University of Washington, Seattle, WA, USA. [174]Neurology Service, Marqués de Valdecilla University Hospital (University of Cantabria and IDIVAL), Santander, Spain. [175]Stockholm Gerontology Research Center, Stockholm, Sweden. [176]Laboratory of Epidemiology, Demography, and Biometry, National Institute of Aging, The National Institutes of Health, Bethesda, MD, USA. [177]Intramural Research Program/National Institute on Aging/National Institutes of Health, Bethesda, MD, USA. [178]Public Health Promotion Unit, Finnish Institute for Health and Welfare, Helsinki, Finland. [179]Memory Clinic, Department of Neurology, Charles University, 2nd Faculty of Medicine and Motol University Hospital, Praha, Czechia. [180]International Clinical Research Center, St. Anne's University Hospital Brno, Brno, Czechia. [181]Departments of Neurology and Epidemiology, University of Washington, Seattle, WA, USA. [182]Department of Neurology, Hospital Universitario Donostia, OSAKIDETZA-Servicio Vasco de Salud, San Sebastian, Spain. [183]Department of Neurology, Columbia University, New York, NY, USA. [184]School of Health Sciences, Bangor University, Bangor, UK. [185]Institute of Neurology, Catholic University of the Sacred Heart, Rome, Italy. [186]Gertrude H. Sergievsky Center, Columbia University, New York, NY, USA. [187]MRC Prion Unit at UCL, UCL Institute of Prion Diseases, London, UK. [188]Unidad de Trastornos del Movimiento, Servicio de Neurología y Neurofisiología. Instituto de Biomedicina de Sevilla (IBiS), Hospital Universitario Virgen del Rocío/CSIC/Universidad de Sevilla, Seville, Spain. [189]Institute for Urban Public Health, University Hospital of University Duisburg-Essen, Essen, Germany. [190]Neurological Tissue Bank of the Biobanc-Hospital Clinic-IDIBAPS, Institut d'Investigacions Biomèdiques August Pi i Sunyer, Barcelona, Spain. [191]Alzheimer's Disease and Other Cognitive Disorders Unit, Neurology Department, Hospital Clinic, Barcelona, Spain. [192]Laboratory of Brain Aging and Neurodegeneration, FIL-CONICET, Buenos Aires, Argentina. [193]Human Genetics, School of Life Sciences, University of Nottingham, Nottingham, UK. [194]Memory Impairment and Neurodegenerative Dementia (MIND) Center, University of Mississippi Medical Center, Jackson, MS, USA. [195]Laboratorio de Bioquímica Molecular, Facultad de Medicina, Instituto de Investigaciones Médicas A. Lanari, UBA, C.A.B.A, Buenos Aires, Argentina. [196]Department of Medicine, University of Washington, Seattle, WA, USA. [197]IRCCS Fondazione Don Carlo Gnocchi, Florence, Italy. [198]Department of Clinical Medicine, University of Copenhagen, Copenhagen, Denmark. [199]DIMEC, University of Parma, Parma, Italy. [200]Resources and Research Memory Center (MRRC) of Distalz, LicendUniversity of Lille, INSERM, CHU Lille, UMR1172, Lille, France. [201]Institut de Biomedicina de València-CSIC CIBERNED, València, Spain. [202]Unitat Mixta de de Neurología y Genética, Institut d'Investigació Sanitària La Fe, València, Spain. [203]US 41-UMS 2014-PLBS, bilille, Université de Lille, CNRS, INSERM, CHU Lille, Institut Pasteur de Lille, Lille, France. [204]Institute of Psychiatry and Psychotherapy, Charité-Universitätsmedizin Berlin, Freie Universität Berlin, Humboldt-Universität Zu Berlin, and Berlin Institute of Health, Berlin, Germany. [205]German Center for Neurodegenerative Diseases (DZNE), Berlin, Germany. [206]Department of Biomedical Sciences, University of Cagliari, Cagliari, Italy. [207]CHUV, Old Age Psychiatry, Department of Psychiatry, Lausanne, Switzerland. [208]Old Age Psychiatry, Department of Psychiatry, Lausanne University Hospital, Lausanne, Switzerland. [209]Department of Geriatric Psychiatry, University Hospital of Psychiatry Zürich, Zürich, Switzerland. [210]Department of Neuropsychiatry and Laboratory of Molecular Psychiatry, Charité, Charitéplatz 1, Berlin, Germany. [211]Department of Clinical Biochemistry, Rigshospitalet, Copenhagen, Denmark. [212]Department of Psychiatry & Neuropsychologie, Maastricht University, Alzheimer Center Limburg, Maastricht, the Netherlands. [213]Depatamento de Especialidades Quirúrgicas Bioquímica e Inmunología, Facultad de Medicina, Universidad de Málaga, Málaga, Spain. [214]Delft Bioinformatics Lab, Delft University of Technology, Delft, the Netherlands. [215]Taub Institute, Columbia University, New York, NY, USA. [216]Bioinformatics, College of Life Sciences, Brigham Young University, Provo, UT, USA. [217]Institute of Social Medicine, Occupational Health and Public Health, University of Leipzig, Leipzig, Germany. [218]Center of Mental Health, Clinic and Policlinic of Psychiatry, Psychosomatics and Psychotherapy, University Hospital of Würzburg, Wuerzburg, Germany. [219]Department of Research and Innovation, Helse Fonna, Haugesund Hospital, Haugesund, Norway. [220]Institute of Clinical Medicine (K1), The University of Bergen, Bergen, Norway. [221]Departamento de Especialidades Quirúrgicas, Bioquímicas e Inmunología, School of Medicine, University of Málaga, Málaga, Spain. [222]Department of Neuroscience and Mental Health, AOU Città della Salute e della Scienza di Torino, Torino, Italy. [223]Athens Association of Alzheimer's Disease and Related Disorders, Athens, Greece. [224]Department of Geriatrics, St. Olav's Hospital, Trondheim

University Hospital, Trondheim, Norway. [225]Department of Immunology, Hospital Universitario Doctor Negrín, Las Palmas de Gran Canaria, Las Palmas, Spain. [226]Neurology Department-Hospital Clínic, IDIBAPS, Universitat de Barcelona, Barcelona, Spain. [227]First Department of Neurology, Aiginition Hospital, National and Kapodistrian University of Athens, Medical School, Athens, Greece. [228]LVR-Hospital Essen, Department of Psychiatry and Psychotherapy, Medical Faculty, University of Duisburg-Essen, Essen, Germany. [229]Department of Primary Medical Care, University Medical Centre Hamburg-Eppendorf, Hamburg, Germany. [230]Institute of Medical Biometry, Informatics and Epidemiology, University Hospital of Bonn, Bonn, Germany. [231]Department of Geriatric Medicine, Oslo University Hospital, Oslo, Norway. [232]Laboratory for Advanced Hematological Diagnostics, Department of Hematology and Stem Cell Transplant, Vito Fazzi Hospital, Lecce, Italy. [233]Centro de Investigación Biomédica en Red de Diabetes y Enfermedades Metabólicas Asociadas, CIBERDEM, Hospital Clínico San Carlos, Madrid, Spain. [234]Department of Neurology, University of Bonn, Bonn, Germany. [235]Department of Biomedical Sciences, Section of Neuroscience and Clinical Pharmacology, University of Cagliari, Cagliari, Italy. [236]Department of Psychiatry, Namsos Hospital, Namsos, Norway. [237]Department of Internal Medicine and Biostatistics, Erasmus MC, Rotterdam, the Netherlands. [238]Neurodegenerative Brain Diseases Group, VIB Center for Molecular Neurology, VIB, Antwerp, Belgium. [239]Department of Neurology, ErasmusMC, Rotterdam, the Netherlands. [240]Laboratory for Cognitive Neurology, Department of Neurosciences, University of Leuven, Leuven, Belgium. [241]Neurology Department, University Hospitals Leuven, Leuven, Belgium. [242]Department of Psychiatry and Psychotherapy, University Medical Center Goettingen, Goettingen, Germany. [243]Department of Psychiatry, Harvard Medical School, McLean Hospital, Belmont, MA, USA. [244]Department of Neurology and CNR-MAJ, F 76000, Normandy Center for Genomic and Personalized Medicine, Normandie University, UNIROUEN, INSERM U1245, CHU Rouen, Rouen, France. [245]German Center for Neurodegenerative Diseases (DZNE), Goettingen, Germany. [246]Medical Science Department, iBiMED, Aveiro, Portugal. [247]Department of Nutrition and Diatetics, Harokopio University, Athens, Greece. [248]Department of Medicine (Biomedical Genetics), Boston University School of Medicine, Boston, MA, USA. [249]Neurosciences Area, Instituto Biodonostia, San Sebastian, Spain. [250]Department of Health Service, University of Washington, Seattle, WA, USA. [251]Excellence Cluster on Cellular Stress Responses in Aging-Associated Diseases (CECAD), University of Cologne, Cologne, Germany. [567]These authors contributed equally: Céline Bellenguez, Fhari Kuçukali, Iris Jansen, Luca Kleineidam, Sonia Moreno-Grau, Najaf Amin, Adam Naj and Rafael Campos-Martin. [568]These authors jointly supervised this work: Mikko Hiltunen, Kristel Sleegers, Gerard Schellenberg, Cornelia van Duijn, Rebecca Sims, Wiesje van der Flier, Agustin Ruiz, Alfredo Ramirez and Jean-Charles Lambert. ✉e-mail: celine.bellenguez@pasteur-lille.fr; Jean-Charles.Lambert@pasteur-lille.fr

## EADB

Jan Laczo[179,180], Vaclav Matoska[252], Maria Serpente[124], Francesca Assogna[48], Fabrizio Piras[48], Federica Piras[48], Valentina Ciullo[48], Jacob Shofany[48], Carlo Ferrarese[38,39], Simona Andreoni[38], Gessica Sala[38], Chiara Paola Zoia[38], Maria Del Zompo[235], Alberto Benussi[66], Patrizia Bastiani[253], Mari Takalo[254], Teemu Natunen[254], Tiina Laatikainen[166,178], Jaakko Tuomilehto[166,178], Riitta Antikainen[255,256], Timo Strandberg[255,257], Jaana Lindström[178], Markku Peltonen[178], Richard Abraham[258], Ammar Al-Chalabi[259], Nicholas J. Bass[260], Carol Brayne[261], Kristelle S. Brown[262], John Collinge[187], David Craig[263], Pangiotis Deloukas[264], Nick Fox[265], Amy Gerrish[265], Michael Gill[266], Rhian Gwilliam[264], Denise Harold[267], Paul Hollingworth[258], Jarret A. Johnston[268], Lesley Jones[258], Brian Lawlor[266], Gill Livingston[260], Simon Lovestone[269], Michelle Lupton[270,271], Aoibhinn Lynch[266], David Mann[272], Bernadette McGuinness[268], Andrew McQuillin[260], Michael C. O'Donovan[258], Michael J. Owen[258], Peter Passmore[268], John F. Powell[270,271], Petra Proitsi[270,271], Martin Rossor[265], Christopher E. Shaw[259], A. David Smith[273], Hugh Gurling[274], Stephen Todd[275], Catherine Mummery[276], Nathalie Ryan[276], Giordano Lacidogna[98], Ad Adarmes-Gómez[11,77], Ana Mauleón[10], Ana Pancho[10], Anna Gailhajenet[10], Asunción Lafuente[10], D. Macias-García[11,77], Elvira Martín[10], Esther Pelejà[10], F. Carrillo[11,77], Isabel Sastre Merlín[11,134], L. Garrote-Espina[11,77], Liliana Vargas[10], M. Carrion-Claro[11,77], M. Marín[96], Ma Labrador[11,77], Mar Buendia[10], María Dolores Alonso[277], Marina Guitart[10], Mariona Moreno[10], Marta Ibarria[10], Mt Periñán[11,77], Nuria Aguilera[10], P. Gómez-Garre[11,77], Pilar Cañabate[10], R. Escuela[11,77], R. Pineda-Sánchez[1,77], R. Vigo-Ortega[11,77], S. Jesús[11,77], Silvia Preckler[10], Silvia Rodrigo-Herrero[132], Susana Diego[10], Alessandro Vacca[67], Fausto Roveta[67], Nicola Salvadori[112], Elena Chipi[112], Henning Boecker[9,278], Christoph Laske[279,280], Robert Perneczky[81,281], Costas Anastasiou[247], Daniel Janowitz[78], Rainer Malik[78], Anna Anastasiou[35], Kayenat Parveen[7], Carmen Lage[282], Sara López-García[282], Anna Antonell[226], Kalina Yonkova Mihova[283], Diyana Belezhanska[52], Heike Weber[284], Silvia Kochen[285], Patricia Solis[285], Nancy Medel[285], Julieta Lisso[285], Zulma Sevillano[285], Daniel G. Politis[285,286], Valeria Cores[285,286], Carolina Cuesta[285,286], Cecilia Ortiz[287], Juan Ignacio Bacha[287], Mario Rios[288], Aldo Saenz[288], Mariana Sanchez Abalos[289], Eduardo Kohler[290], Dana Lis Palacio[291], Ignacio Etcheparreborda[291], Matias Kohler[291], Gisela Novack[292], Federico Ariel Prestia[292], Pablo Galeano[292], Eduardo M. Castaño[292], Sandra Germani[293],

Carlos Reyes Toso[293], Matias Rojo[293], Carlos Ingino[293], Carlos Mangone[293], David C. Rubinsztein[294], Stefan Teipel[295], Nathalie Fievet[1], Vincent Deramerourt[200], Charlotte Forsell[143,144], Håkan Thonberg[143,144], Maria Bjerke[89], Ellen De Roeck[89], María Teresa Martínez-Larrad[296] and Natividad Olivar[293]

[252]Department of Clinical Biochemistry, Hematology and Immunology, Na Homolce Hospital, Prague, Czechia. [253]Institute of Gerontology and Geriatrics, Department of Medicine, University of Perugia, Perugia, Italy. [254]Insitute of Biomedicine, University of Eastern Finland, Kuopio, Finland. [255]Center for Life Course Health Research, University of Oulu, Oulu, Finland. [256]Medical Research Center Oulu, Oulu University Hospital, Oulu, Finland. [257]University of Helsinki and Helsinki University Hospital, Helsinki, Finland. [258]Division of Psychological Medicine and Clinial Neurosciences, MRC Centre for Neuropsychiatric Genetics and Genomics, Cardiff University, Cardiff, UK. [259]Institute of Psychiatry, Psychology and Neuroscience, Kings College London, London, UK. [260]Division of Psychiatry, University College London, London, UK. [261]Institute of Public Health, University of Cambridge, Cambridge, UK. [262]Institute of Genetics, Queens Medical Centre, University of Nottingham, Nottingham, UK. [263]Ageing Group, Centre for Public Health, School of Medicine, Dentistry and Biomedical Sciences, Queen's University Belfast, Belfast, UK. [264]The Wellcome Trust Sanger Institute, Wellcome Trust Genome Campus, Hinxton, Cambridge, UK. [265]Dementia Research Centre, Department of Neurodegenerative Disease, UCL Institute of Neurology, London, UK. [266]Mercer's Institute for Research on Ageing, St James' Hospital, Dublin, Ireland. [267]School of Biotechnology, Dublin City University, Dublin, Ireland. [268]Centre for Public Health, School of Medicine, Dentistry and Biomedical Sciences, Queens University, Belfast, UK. [269]Department of Psychiatry, University of Oxford, Oxford, UK. [270]Genetic Epidemiology, QIMR Berghofer Medical Research Institute, Herston, Queensland, Australia. [271]Department of Basic and Clinical Neuroscience, Institute of Psychiatry, Psychology and Neuroscience, Kings College London, London, UK. [272]Division of Neuroscience and Experimental Psychology, School of Biological Sciences, Faculty of Biology, Medicine and Health, University of Manchester, Manchester Academic Health Science Centre, Manchester, UK. [273]Oxford Project to Investigate Memory and Ageing (OPTIMA), University of Oxford, Level 4, John Radcliffe Hospital, Oxford, UK. [274]Department of Mental Health Sciences, University College London, London, UK. [275]Ageing Group, Centre for Public Health, School of Medicine, Dentistry and Biomedical Sciences, Queen's University Belfast, Belfast, UK. [276]Dementia Research Centre, UCL, London, UK. [277]Servei de Neurologia Hospital Clínic, Universitari de València, Valencia, Spain. [278]Department of Radiology, University Hospital Bonn, Bonn, Germany. [279]German Center for Neurodegenerative Diseases (DZNE), Tübingen, Germany. [280]Section for Dementia Research, Department of Psychiatry, Hertie Institute for Clinical Brain Research, Tübingen, Germany. [281]Department of Psychiatry and Psychotherapy, University Hospital, LMU Munich, Munich, Germany. [282]Service of Neurology, University Hospital Marqués de Valdecilla, IDIVAL, University of Cantabria, Santander, Spain. [283]Molecular Medicine Center, Department of Medical chemistry and biochemistry, Medical University of Sofia, Sofia, Bulgaria. [284]Department of Psychiatry, Psychosomatics and Psychotherapy, Center of Mental Health, University Hospital of Würzburg, Würzburg, Germany. [285]ENYS (Estudio en Neurociencias y Sistemas Complejos) CONICET- Hospital El Cruce "Nestor Kirchner"- UNAJ, Buenos Aires, Argentina. [286]HIGA Eva Perón, Buenos Aires, Argentina. [287]Neurología Clinica, Buenos Aires, Argentina. [288]Dirección de Atención de Adultos Mayores del Min. Salud Desarrollo Social y Deportes de la Pcia. de Mendoza, Mendoza, Argentina. [289]Laboratorio de Genética Forense del Ministerio Público de la Pcia de La Pampa, La Pampa, Argentina. [290]Fundacion Sinapsis, Santa Rosa, Argentina. [291]Hospital Dr. Lucio Molas, Santa Rosa; Fundacion Ayuda Enfermo Renal y Alta Complejidad (FERNAC), Santa Rosa, Argentina. [292]Laboratory of Brain Aging and Neurodegeneration (FIL), Buneos Aires, Argentina. [293]Centro de Neuropsiquiatría y Neurología de la Conducta (CENECON), Facultad de Medicina, Universidad de Buenos Aires (UBA), C.A.B.A., Buenos Aires, Argentina. [294]Cambridge Institute for Medical Research and UK Dementia Research Institute, University of Cambridge, Cambridge, UK. [295]German Center for Neurodegenerative Diseases (DZNE), Rostock, Germany. [296]Centro de Investigación Biomédica en Red de Diabetes y Enfermedades Metabólicas Asociadas, CIBERDEM, Hospital Clínico San Carlos, Madrid, Spain.

## GR@ACE

Nuria Aguilera[10,11], Mar Buendia[10], Amanda Cano[10], Pilar Cañabate[10,11], Susana Diego[10], Anna Gailhajenet[10], Marina Guitart[10], Marta Ibarria[10], Asunción Lafuente[10], Juan Macias[97], Olalla Maroñas[297], Elvira Martín[10], Mariona Moreno[10], Raúl Nuñez-Llaves[10], Clàudia Olivé[10], Ana Pancho[10], Ester Pelejá[10], Silvia Preckler[10] and Liliana Vargas[10]

[297]Grupo de Medicina Xen´omica, Centro Nacional de Genotipado (CEGEN-PRB3-ISCIII), Universidad de Santiago de Compostela, Santiago de Compostela, Spain.

## DEGESCO

Astrid D. Adarmes-Gómez[11,188], María Dolores Alonso[298], Guillermo Amer-Ferrer[299], Martirio Antequera[37], Juan Andrés Burguera[49], Fátima Carrillo[11,188], Mario Carrión-Claro[11,188], María José Casajeros[139], Marian Martinez de Pancorbo[300], Rocío Escuela[11,188], Lorena Garrote-Espina[11,188], Pilar Gómez-Garre[11,188], Saray Hevilla[138], Silvia Jesús[11,188], Miguel Angel Labrador Espinosa[11,188], Agustina Legaz[37], Sara López-García[11,174], Daniel Macias-García[11,188], Salvadora Manzanares[299,301], Marta Marín[132], Juan Marín-Muñoz[37], Tamara Marín[138], Begoña Martínez[37], Victoriana Martínez[37], Pablo Martínez-Lage Álvarez[302], Maite Mendioroz Iriarte[303], María Teresa Periñán-Tocino[11,188], Rocío Pineda-Sánchez[11,188],

Diego Real de Asúa[304], Silvia Rodrigo[132], Isabel Sastre[11,134], Maria Pilar Vicente[37], Rosario Vigo-Ortega[11,188] and Liliana Vivancos[37]

[298]Servei de Neurologia, Hospital Clínic Universitari de València, Valencia, Spain. [299]Department of Neurology, Hospital Universitario Son Espases, Palma, Spain. [300]BIOMICs, País Vasco; Centro de Investigación Lascaray, Universidad del País Vasco UPV/EHU, Vitoria-Gasteiz, Spain. [301]Fundación para la Formación e Investigación Sanitarias de la Región de Murcia, Palma, Spain. [302]Centro de Investigación y Terapias Avanzadas, Fundación CITA-Alzheimer, San Sebastian, Spain. [303]Navarrabiomed, Pamplona, Spain. [304]Hospital Universitario La Princesa, Madrid, Spain.

## EADI

Jacques Epelbaum[305], Didier Hannequin[244], Dominique campion[92], Vincent Deramecourt[200], Nathalie Fievet[1], Christophe Tzourio[101], Alexis Brice[306] and Bruno Dubois[307]

[305]UMR 7179 CNRS/MNHN, Brunoy, France. [306]Sorbonne Université, Paris Brain Institute, APHP, INSERM, CNRS, Paris, France. [307]Department of Neurology, Institute of Memory and Alzheimer's Disease (IM2A), Pitié-Salpêtrière Hospital, AP-HP, Boulevard de l'Hôpital, Paris, France.

## GERAD

Denise Harold[267], Paul Hollingworth[258], Amy Gerrish[265], Amy Williams[16], Charlene Thomas[16], Chloe Davies[16], William Nash[16], Kimberley Dowzell[16], Atahualpa Castillo Morales[16,108], Mateus Bernardo-Harrington[16,108], James Turton[308], Jenny Lord[308], Kristelle Brown[262], Emma Vardy[309], Elizabeth Fisher[310], Jason D. Warren[310], Martin Rossor[29], Natalie S. Ryan[276], Rita Guerreiro[310], James Uphill[187], John Collinge[187], Michelle Lupton[270,271], Ammar Al-Chalabi[259], Christopher E. Shaw[259], Nick Bass[260], Richard Abraham[258], Reinhard Heun[311], Heike Kölsch[312], Britta Schürmann[312], André Lacour[9], Christine Herold[9], Simon Lovestone[269], Bernadette McGuinness[268], David Craig[263], Janet A. Johnston[268], Michael Gill[266], Peter Passmore[263], Stephen Todd[275], John Powell[270,271], Petra Proitsi[270,271], Yogen Patel[313], Angela Hodges[311], Tim Becker[9,314], A. David Smith[273], Donald Warden[273], Gordon Wilcock[273], Robert Clarke[310], Aoibhinn Lynch[266], Brian Lawlor[266], Andrew McQuillin[260], Gill Livingston[260], David C. Rubinsztein[294], Carol Brayne[261], Rhian Gwilliam[264], Panagiotis Deloukas[264], Yoav Ben-Shlomo[315], David Mann[272], Nigel M. Hooper[272], Stuart Pickering-Brown[272], Rebecca Sussams[316], Nick Warner[317], Anthony Bayer[318], Isabella Heuser[319], Dmitriy Drichel[320], Norman Klopp[321], Manuel Mayhaus[322], Matthias Riemenschneider[322], Sabrina Pinchler[322], Thomas Feulner[322], Wei Gu[322], Hendrik van den Bussche[229], Michael Hüll[323], Lutz Frölich[324], H-Erich Wichmann[321], Karl-Heinz Jöckel[325], Michael O'Donovan[258], Lesley Jones[258] and Michael Owen[258]

[308]Institute of Genetics, Queen's Medical Centre, University of Nottingham, Nottingham, UK. [309]Institute for Ageing and Health, Newcastle University, Campus for Ageing and Vitality, Newcastle upon Tyne, UK. [310]Department of Neurodegenerative Disease, UCL Institute of Neurology, London, UK. [311]Department of Old Age Psychiatry, Institute of Psychiatry, Psychology and Neuroscience, King's College London, London, UK. [312]Department of Psychiatry and Psychotherapy, University of Bonn, Bonn, Germany. [313]Department of Basic and Clinical Neuroscience, Institute of Psychiatry, Psychology and Neuroscience, King's College London, London, UK. [314]Institute for Medical Biometry, Informatics and Epidemiology, University of Bonn, Bonn, Germany. [315]Population Health Sciences, Bristol Medical School, University of Bristol, Bristol, UK. [316]Division of Clinical Neurosciences, School of Medicine, University of Southampton, Southampton, UK. [317]Somerset Partnership NHS Trust, Somerset, UK. [318]Institute of Primary Care and Public Health, Cardiff University, University Hospital of Wales, Cardiff, UK. [319]Department of Psychiatry and Psychotherapy, Charité University Medicine, Berlin, Germany. [320]Cologne Center for Genomics, University of Cologne, Cologne, Germany. [321]Institute of Epidemiology, Helmholtz Zentrum München, German Research Center for Environmental Health, Neuherberg, Munich, Germany. [322]Department of Psychiatry and Psychotherapy, University Hospital, Saarland, Germany. [323]Department of Psychiatry, University of Freiburg, Freiburg, Germany. [324]Central Institute of Mental Health, Medical Faculty Mannheim, University of Heidelberg, Heidelberg, Germany. [325]Institute for Medical Informatics, Biometry and Epidemiology, University Hospital of Essen, University Duisburg-Essen, Essen, Germany.

## Demgene

Shahram Bahrami[46,326], Ingunn Bosnes[327,328], Per Selnes[329] and Sverre Bergh[330]

326Division of Mental Health and Addiction, Oslo University Hospital, Oslo, Norway. 327Department of Mental Health, Faculty of Medicine and Health Sciences, Norwegian University of Science and Technology, Trondheim, Norway. 328Department of Psychiatry, Hospital Namsos, Nord-Trøndelag Health Trust, Namsos, Norway. 329Department of Neurology, Akershus University Hospital, Lørenskog, Norway. 330Centre for Old Age Psychiatry Research, Innlandet Hospital Trust, Ottestad, Norway.

## FinnGen

Aarno Palotie[331], Mark Daly[331], Howard Jacob[332], Athena Matakidou[333], Heiko Runz[334], Sally John[334], Robert Plenge[335], Mark McCarthy[336], Julie Hunkapiller[336], Meg Ehm[337], Dawn Waterworth[337], Caroline Fox[338], Anders Malarstig[339], Kathy Klinger[340], Kathy Call[340], Tim Behrens[341], Patrick Loerch[342], Tomi Mäkelä[343], Jaakko Kaprio[331], Petri Virolainen[344], Kari Pulkki[344], Terhi Kilpi[345], Markus Perola[345], Jukka Partanen[346], Anne Pitkäranta[347], Riitta Kaarteenaho[348], Seppo Vainio[348], Miia Turpeinen[348], Raisa Serpi[348], Tarja Laitinen[349], Johanna Mäkelä[350], Veli-Matti Kosma[351], Urho Kujala[352], Outi Tuovila[353], Minna Hendolin[353], Raimo Pakkanen[353], Jeff Waring[332], Bridget Riley-Gillis[332], Jimmy Liu[334], Shameek Biswas[335], Dorothee Diogo[338], Catherine Marshall[339], Xinli Hu[339], Matthias Gossel[340], Robert Graham[341], Beryl Cummings[342], Samuli Ripatti[331], Johanna Schleutker[344], Mikko Arvas[346], Olli Carpén[347], Reetta Hinttala[348], Johannes Kettunen[348], Arto Mannermaa[351], Jari Laukkanen[352], Valtteri Julkunen[354], Anne Remes[354], Reetta Kälviäinen[354], Jukka Peltola[355], Pentti Tienari[356], Juha Rinne[357], Adam Ziemann[332], Jeffrey Waring[332], Sahar Esmaeeli[332], Nizar Smaoui[332], Anne Lehtonen[332], Susan Eaton[334], Sanni Lahdenperä[334], Janet van Adelsberg[335], John Michon[336], Geoff Kerchner[336], Natalie Bowers[336], Edmond Teng[336], John Eicher[336], Vinay Mehta[338], Padhraig Gormley[338], Kari Linden[339], Christopher Whelan[339], Fanli Xu[337], David Pulford[337], Martti Färkkilä[356], Sampsa Pikkarainen[356], Airi Jussila[358], Timo Blomster[359], Mikko Kiviniemi[360], Markku Voutilainen[357], Bob Georgantas[332], Graham Heap[332], Fedik Rahimov[332], Keith Usiskin[335], Tim Lu[6], Danny Oh[336], Kirsi Kalpala[339], Melissa Miller[339], Linda McCarthy[337], Kari Eklund[356], Antti Palomäki[357], Pia Isomäki[358], Laura Pirilä[357], Oili Kaipiainen-Seppänen[360], Johanna Huhtakangas[359], Apinya Lertratanakul[332], Marla Hochfeld[335], Nan Bing[339], Jorge Esparza Gordillo[337], Nina Mars[331], Margit Pelkonen[360], Paula Kauppi[356], Hannu Kankaanranta[355], Terttu Harju[359], David Close[333], Steven Greenberg[335], Hubert Chen[336], Jo Betts[337], Soumitra Ghosh[337], Veikko Salomaa[361], Teemu Niiranen[361], Markus Juonala[357], Kaj Metsärinne[357], Mika Kähönen[358], Juhani Junttila[359], Markku Laakso[354], Jussi Pihlajamäki[354], Juha Sinisalo[356], Marja-Riitta Taskinen[356], Tiinamaija Tuomi[356], Ben Challis[333], Andrew Peterson[336], Audrey Chu[338], Jaakko Parkkinen[339], Anthony Muslin[340], Heikki Joensuu[356], Tuomo Meretoja[356], Lauri Aaltonen[356], Johanna Mattson[356], Annika Auranen[355], Peeter Karihtala[359], Saila Kauppila[359], Päivi Auvinen[359], Klaus Elenius[357], Relja Popovic[332], Jennifer Schutzman[336], Andrey Loboda[338], Aparna Chhibber[338], Heli Lehtonen[339], Stefan McDonough[339], Marika Crohns[340], Diptee Kulkarni[337], Kai Kaarniranta[354], Joni A. Turunen[356], Terhi Ollila[356], Sanna Seitsonen[356], Hannu Uusitalo[355], Vesa Aaltonen[357], Hannele Uusitalo-Järvinen[355], Marja Luodonpää[359], Nina Hautala[359], Stephanie Loomis[334], Erich Strauss[336], Hao Chen[336], Anna Podgornaia[338], Joshua Hoffman[337], Kaisa Tasanen[359], Laura Huilaja[359], Katariina Hannula-Jouppi[356], Teea Salmi[358], Sirkku Peltonen[356], Leena Koulu[356], Ilkka Harvima[354], Ying Wu[9], David Choy[336], Pirkko Pussinen[356], Aino Salminen[356], Tuula Salo[356], David Rice[356], Pekka Nieminen[356], Ulla Palotie[356], Maria Siponen[354], Liisa Suominen[354], Päivi Mäntylä[354], Ulvi Gursoy[357], Vuokko Anttonen[359], Kirsi Sipilä[359], Justin Wade Davis[332], Danjuma Quarless[332], Slavé Petrovski[333], Eleonor Wigmore[333], Chia-Yen Chen[334], Paola Bronson[334], Ellen Tsai[334], Yunfeng Huang[334], Joseph Maranville[335], Elmutaz Shaikho[335], Elhaj Mohammed[335],

Samir Wadhawan[362], Erika Kvikstad[362], Minal Caliskan[362], Diana Chang[336], Tushar Bhangale[336], Sarah Pendergrass[336], Emily Holzinger[338], Xing Chen[339], Åsa Hedman[339], Karen S. King[337], Clarence Wang[340], Ethan Xu[340], Franck Auge[340], Clement Chatelain[340], Deepak Rajpal[340], Dongyu Liu[340], Katherine Call[340], Tai-he Xia[340], Matt Brauer[341], Mitja Kurki[331], Juha Karjalainen[331], Aki Havulinna[331], Anu Jalanko[331], Priit Palta[331], Pietro della Briotta Parolo[331], Wei Zhou[363], Susanna Lemmelä[331], Manuel Rivas[364], Jarmo Harju[331], Arto Lehisto[331], Andrea Ganna[331], Vincent Llorens[331], Hannele Laivuori[331], Sina Rüeger[331], Mari E. Niemi[331], Taru Tukiainen[331], Mary Pat Reeve[331], Henrike Heyne[331], Nina Mars[331], Kimmo Palin[365], Javier Garcia-Tabuenca[366], Harri Siirtola[366], Tuomo Kiiskinen[331], Jiwoo Lee[331], Kristin Tsuo[331], Amanda Elliott[331], Kati Kristiansson[345], Kati Hyvärinen[367], Jarmo Ritari[367], Miika Koskinen[347], Katri Pylkäs[348], Marita Kalaoja[348], Minna Karjalainen[348], Tuomo Mantere[348], Eeva Kangasniemi[350], Sami Heikkinen[351], Eija Laakkonen[352], Csilla Sipeky[368], Samuel Heron[368], Antti Karlsson[344], Dhanaprakash Jambulingam[368], Venkat Subramaniam Rathinakannan[368], Risto Kajanne[331], Mervi Aavikko[331], Manuel González Jiménez[331], Pietro della Briotta Parola[331], Arto Lehistö[331], Masahiro Kanai[363], Mari Kaunisto[331], Elina Kilpeläinen[331], Timo P. Sipilä[331], Georg Brein[331], Ghazal Awaisa[331], Anastasia Shcherban[331], Kati Donner[331], Anu Loukola[347], Päivi Laiho[345], Tuuli Sistonen[345], Essi Kaiharju[345], Markku Laukkanen[345], Elina Järvensivu[345], Sini Lähteenmäki[345], Lotta Männikkö[345], Regis Wong[345], Hannele Mattsson[345], Tero Hiekkalinna[345], Teemu Paajanen[345], Kalle Pärn[331] and Javier Gracia-Tabuenca[366]

[331]Institute for Molecular Medicine Finland, HiLIFE, University of Helsinki, Helsinki, Finland. [332]AbbVie, Chicago, IL, USA. [333]Astra Zeneca, Cambridge, UK. [334]Biogen, Cambridge, MA, USA. [335]Celgene, Summit, NJ, USA. [336]Genentech, San Francisco, CA, USA. [337]GlaxoSmithKline, Brentford, UK. [338]Merck, Kenilworth, NJ, USA. [339]Pfizer, New York, NY, USA. [340]Sanofi, Paris, France. [341]Maze Therapeutics, San Francisco, CA, USA. [342]Janssen Biotech, Beerse, Belgium. [343]HiLIFE, University of Helsinki, Helsinki, Finland. [344]Auria Biobank, University of Turku, Hospital District of Southwest Finland, Turku, Finland. [345]THL Biobank, The National Institute of Health and Welfare Helsinki, Helsinki, Finland. [346]Finnish Red Cross Blood Service, Finnish Hematology Registry and Clinical Biobank, Helsinki, Finland. [347]Helsinki Biobank, Helsinki University and Hospital District of Helsinki and Uusimaa, Helsinki, Finland. [348]Northern Finland Biobank Borealis, University of Oulu, Northern Ostrobothnia Hospital District, Oulu, Finland. [349]Oxford Healthy Aging Project, Clinical Trial Service Unit, University of Oxford, Oxford, UK. [350]Finnish Clinical Biobank Tampere, University of Tampere, Pirkanmaa Hospital District, Tampere, Finland. [351]Biobank of Eastern Finland, University of Eastern Finland / Northern Savo Hospital District, Kuopio, Finland. [352]Central Finland Biobank, University of Jyväskylä, Central Finland Health Care District, Jyväskylä, Finland. [353]Business Finland, Helsinki, Finland. [354]Northern Savo Hospital District, Kuopio, Finland. [355]Pirkanmaa Hospital District, Tampere, Finland. [356]Hospital District of Helsinki and Uusimaa, Helsinki, Finland. [357]Hospital District of Southwest Finland, Turku, Finland. [358]Pirkanmaa Hospital District, Tampere, Finland. [359]Northern Ostrobothnia Hospital District, Oulu, Finland. [360]Northern Savo Hospital District, Kuopio, Finland. [361]The National Institute of Health and Welfare Helsinki, Helsinki, Finland. [362]Bristol Myers Squibb, New York, NY, USA. [363]Broad Institute, Cambridge, MA, USA. [364]University of Stanford, Stanford, CA, USA. [365]University of Helsinki, Helsinki, Finland. [366]University of Tampere, Tampere, Finland. [367]Finnish Red Cross Blood Service, Helsinki, Finland. [368]University of Turku, Turku, Finland.

## ADGC

Erin Abner[369], Perrie M. Adams[370], Alyssa Aguirre[371], Marilyn S. Albert[372], Roger L. Albin[373,374,375], Mariet Allen[376], Lisa Alvarez[377], Liana G. Apostolova[378,379], Steven E. Arnold[380], Sanjay Asthana[381,382,383], Craig S. Atwood[381,382,383], Gayle Ayres[371], Clinton T. Baldwin[248], Robert C. Barber[377], Lisa L. Barnes[384,385,386], Sandra Barral[50,183,186], Thomas G. Beach[387], James T. Becker[388], Gary W. Beecham[154], Duane Beekly[389], Jennifer E. Below[93,390], Penelope Benchek[57], Bruno A. Benitez[391], David Bennett[384,386], John Bertelson[392], Flanagan E. Margaret[393,394], Thomas D. Bird[181,395], Deborah Blacker[396,397], Bradley F. Boeve[398], James D. Bowen[399], Adam Boxer[400], James Brewer[401], James R. Burke[402], Jeffrey M. Burns[403], Will S. Bush[57], Joseph D. Buxbaum[103,104,106,160], Nigel J. Cairns[404], Chuanhai Cao[405], Christopher S. Carlson[406], Cynthia M. Carlsson[382,383], Regina M. Carney[407], Minerva M. Carrasquillo[376], Scott Chasse[408], Marie-Francoise Chesselet[409], Hung-Hsin Chen[93], Alessandra Chesi[14], Nathaniel A. Chin[381,382], Helena C. Chui[410], Jaeyoon Chung[248], Suzanne Craft[411],

Paul K. Crane[196], David H. Cribbs[412], Elizabeth A. Crocco[413], Carlos Cruchaga[414,415], Michael L. Cuccaro[88,154], Munro Cullum[370], Eveleen Darby[416], Barbara Davis[417], Philip L. De Jager[418], Charles DeCarli[419], John DeToledo[420], Malcolm Dick[421], Dennis W. Dickson[376], Beth A. Dombroski[14], Rachelle S. Doody[416], Ranjan Duara[422], Nilüfer Ertekin-Taner[376,423], Denis A. Evans[424], Kelley M. Faber[119], Thomas J. Fairchild[425], Kenneth B. Fallon[426], David W. Fardo[122], Martin R. Farlow[427], John J. Farrell[248], Victoria Fernandez-Hernandez[391], Steven Ferris[428], Tatiana M. Foroud[119], Matthew P. Frosch[429], Brian Fulton-Howard[430], Douglas R. Galasko[401], Adriana Gamboa[431,432], Marla Gearing[433,434], Daniel H. Geschwind[414], Bernardino Ghetti[435], John R. Gilbert[88,154], Thomas J. Grabowski[181,436], Neill R. Graff-Radford[377,423], Struan F. A. Grant[437,438,439], Robert C. Green[440], John H. Growdon[441], Jonathan L. Haines[56,57], Hakon Hakonarson[442,443], James Hall[377], Ronald L. Hamilton[444], Kara L. Hamilton-Nelson[154], Oscar Harari[415], Lindy E. Harrell[445], Jacob Haut[14], Elizabeth Head[446], Victor W. Henderson[447,448], Michelle Hernandez[420], Timothy Hohman[93,449], Lawrence S. Honig[50], Ryan M. Huebinger[450], Matthew J. Huentelman[451], Christine M. Hulette[452], Bradley T. Hyman[441], Linda S. Hynan[370,453,454], Laura Ibanez[415,455], Gail P. Jarvik[196,456], Suman Jayadev[181], Lee-Way Jin[457], Kim Johnson[420], Leigh Johnson[431], M. Ilyas Kamboh[388,458,459], Anna M. Karydas[400], Mindy J. Katz[460], Jeffrey A. Kaye[461,462], C. Dirk Keene[463], Aisha Khaleeq[416], Ronald Kim[446], Janice Knebl[431], Neil W. Kowall[20,464], Joel H. Kramer[465], Pavel P. Kuksa[14], Frank M. LaFerla[466], James J. Lah[467], Eric B. Larson[468], Chien-Yueh Lee[14], Edward B. Lee[14], Alan Lerner[57], Yuk Yee Leung[14], James B. Leverenz[469], Allan I. Levey[467], Mingyao Li[14], Andrew P. Lieberman[470], Richard B. Lipton[460], Mark Logue[248,397,471], Constantine G. Lyketsos[472], John Malamon[14], Douglas Mains[431,432], Daniel C. Marson[445], Frank Martiniuk[473], Deborah C. Mash[474], Eliezer Masliah[401,475], Paul Massman[416], Arjun Masurkar[428], Wayne C. McCormick[196], Susan M. McCurry[476], Andrew N. McDavid[406], Stefan McDonough[477], Ann C. McKee[20,464], Marsel Mesulam[393,394], Jesse Mez[20], Bruce L. Miller[478], Carol A. Miller[479], Joshua W. Miller[457], Thomas J. Montine[480], Edwin S. Monuki[446], John C. Morris[404,415,445,481], Amanda J. Myers[413], Trung Nguyen[453], Sid O'Bryant[482], John M. Olichney[483], Marcia Ory[484], Raymond Palmer[485], Joseph E. Parisi[486], Henry L. Paulson[373,375], Valory Pavlik[416], David Paydarfar[371], Victoria Perez[420], Elaine Peskind[487], Ronald C. Petersen[398], Jennifer E. Phillips-Cremins[439,488], Aimee Pierce[412], Marsha Polk[489], Wayne W. Poon[421], Huntington Potter[490], Liming Qu[14], Mary Quiceno[491,492], Joseph F. Quinn[461,462], Ashok Raj[405], Murray Raskind[487], Eric M. Reiman[451,493,494,495], Barry Reisberg[428,489], Joan S. Reisch[417], John M. Ringman[410], Erik D. Roberson[445], Monica Rodriguear[416], Ekaterina Rogaeva[496], Howard J. Rosen[400], Roger N. Rosenberg[453], Donald R. Royall[485], Mark A. Sager[382], Mary Sano[106], Andrew J. Saykin[119,497], Julie A. Schneider[384,386,498], Lon S. Schneider[410,499], William W. Seeley[400], Jin Sha[14], Susan H. Slifer[154], Scott Small[50,186], Amanda G. Smith[405], Janet P. Smith[417], Yeunjoo E. Song[56,57], Joshua A. Sonnen[463], Salvatore Spina[435], Peter St George-Hyslop[500,501], Robert A. Stern[20], Alan B. Stevens[484,502,503], Stephen M. Strittmatter[504], David Sultzer[505], Russell H. Swerdlow[403], Rudolph E. Tanzi[441], Jeffrey L. Tilson[506], John Q. Trojanowski[14], Juan C. Troncoso[507], Debby W. Tsuang[395,487], Otto Valladares[14], Vivianna M. Van Deerlin[14], Linda J. van Eldik[122], Robert Vassar[393,394], Harry V. Vinters[410,508], Jean-Paul Vonsattel[50], Sandra Weintraub[509], Kathleen A. Welsh-Bohmer[402,510], Patrice L. Whitehead[154], Ellen M. Wijsman[196,456,511], Kirk C. Wilhelmsen[408], Benjamin Williams[411], Jennifer Williamson[50], Henrik Wilms[420], Thomas S. Wingo[467], Thomas Wisniewski[512,513], Randall L. Woltjer[514],

Martin Woon[392], Clinton B. Wright[515], Chuang-Kuo Wu[420], Steven G. Younkin[376,423], Chang-En Yu[196], Lei Yu[384,386], Yuanchao Zhang[442], Yi Zhao[48] and Xiongwei Zhu[516]

[369]Sanders-Brown Center on Aging, Department of Epidemiology, College of Public Health, University of Kentucky, Lexington, KY, USA. [370]Department of Psychiatry, University of Texas Southwestern Medical Center, Dallas, TX, USA. [371]Department of Neurology, Dell Medical School, University of Texas at Austin, Austin, TX, USA. [372]Department of Neurology, Johns Hopkins University, Baltimore, MD, USA. [373]Department of Neurology, University of Michigan, Ann Arbor, MI, USA. [374]Geriatric Research, Education and Clinical Center (GRECC), VA Ann Arbor Healthcare System (VAAAHS), Ann Arbor, MI, USA. [375]Michigan Alzheimer's Disease Center, University of Michigan, Ann Arbor, MI, USA. [376]Department of Neuroscience, Mayo Clinic, Jacksonville, FL, USA. [377]Department of Pharmacology and Neuroscience, University of North Texas Health Science Center, Fort Worth, TX, USA. [378]Departments of Neurology, Radiology, and Medical and Molecular Genetics, Indiana University School of Medicine, Indianapolis, IN, USA. [379]Indiana Alzheimer's Disease Research Center, Indiana University School of Medicine, Indianapolis, IN, USA. [380]Department of Psychiatry, Perelman School of Medicine, University of Pennsylvania, Philadelphia, PA, USA. [381]Geriatric Research, Education and Clinical Center (GRECC), University of Wisconsin, Madison, WI, USA. [382]Department of Medicine, University of Wisconsin, Madison, WI, USA. [383]Wisconsin Alzheimer's Disease Research Center, Madison, WI, USA. [384]Department of Neurological Sciences, Rush University Medical Center, Chicago, IL, USA. [385]Department of Behavioral Sciences, Rush University Medical Center, Chicago, IL, USA. [386]Rush Alzheimer's Disease Center, Rush University Medical Center, Chicago, IL, USA. [387]Civin Laboratory for Neuropathology, Banner Sun Health Research Institute, Phoenix, AZ, USA. [388]Departments of Psychiatry, Neurology, and Psychology, University of Pittsburgh School of Medicine, Pittsburgh, PA, USA. [389]National Alzheimer's Coordinating Center, University of Washington, Seattle, WA, USA. [390]Human Genetics Center, Department of Epidemiology, Human Genetics, and Environmental Sciences, School of Public Health, The University of Texas Health Science Center at Houston, Houston, TX, USA. [391]Department of Psychiatry and Hope Center Program on Protein Aggregation and Neurodegeneration, Washington University School of Medicine, St. Louis, MO, USA. [392]Department of Psychiatry, University of Texas at Austin/Dell Medical School, Austin, TX, USA. [393]Department of Pathology, Northwestern University Feinberg School of Medicine, Chicago, IL, USA. [394]Cognitive Neurology and Alzheimer's Disease Center, Northwestern University Feinberg School of Medicine, Chicago, IL, USA. [395]VA Puget Sound Health Care System/GRECC, Seattle, WA, USA. [396]Department of Epidemiology, Harvard School of Public Health, Boston, MA, USA. [397]Department of Psychiatry, Massachusetts General Hospital/ Harvard Medical School, Boston, MA, USA. [398]Department of Neurology, Mayo Clinic, Rochester, MN, USA. [399]Swedish Medical Center, Seattle, WA, USA. [400]Department of Neurology, University of California San Francisco, San Francisco, CA, USA. [401]Department of Neurosciences, University of California San Diego, La Jolla, CA, USA. [402]Department of Medicine, Duke University, Durham, NC, USA. [403]University of Kansas Alzheimer's Disease Center, University of Kansas Medical Center, Kansas City, KS, USA. [404]Department of Pathology and Immunology, Washington University, St. Louis, MO, USA. [405]USF Health Byrd Alzheimer's Institute, University of South Florida, Tampa, FL, USA. [406]Fred Hutchinson Cancer Research Center, Seattle, WA, USA. [407]Mental Health and Behavioral Science Service, Bruce W. Carter VA Medical Center, Miami, FL, USA. [408]Department of Genetics, University of North Carolina at Chapel Hill, Chapel Hill, NC, USA. [409]Neurogenetics Program, University of California, Los Angeles, Los Angeles, CA, USA. [410]Department of Neurology, University of Southern California, Los Angeles, CA, USA. [411]Section of Gerontology and Geriatric Medicine Research, Wake Forest School of Medicine, Winston-Salem, NC, USA. [412]Department of Neurology, University of California, Irvine, Irvine, CA, USA. [413]Department of Psychiatry and Behavioral Sciences, Miller School of Medicine, University of Miami, Miami, FL, USA. [414]NeuroGenomics and Informatics, Washington University in St. Louis, St. Louis, MO, USA. [415]Department of Psychiatry, Washington University in St. Louis, St Louis, MO, USA. [416]Alzheimer's Disease and Memory Disorders Center, Baylor College of Medicine, Houston, TX, USA. [417]Department of Population and Data Sciences, University of Texas Southwestern Medical Center, Dallas, Texas, USA. [418]Center for Translational and Computational Neuroimmunology, Department of Neurology, Columbia University Medical Center, New York, NY, USA. [419]Department of Neurology, University of California, Davis, Sacramento, CA, USA. [420]Departments of Neurology, Pharmacology and Neuroscience, Texas Tech University Health Science Center, Lubbock, TX, USA. [421]Institute for Memory Impairments and Neurological Disorders, University of California, Irvine, Irvine, CA, USA. [422]Wien Center for Alzheimer's Disease and Memory Disorders, Mount Sinai Medical Center, Miami Beach, FL, USA. [423]Department of Neurology, Mayo Clinic, Jacksonville, FL, USA. [424]Rush Institute for Healthy Aging, Department of Internal Medicine, Rush University Medical Center, Chicago, IL, USA. [425]Office of Strategy and Measurement, University of North Texas Health Science Center, Fort Worth, TX, USA. [426]Department of Pathology, University of Alabama at Birmingham, Birmingham, AL, USA. [427]Department of Neurology, Indiana University, Indianapolis, IN, USA. [428]Department of Psychiatry, New York University, New York, NY, USA. [429]C.S. Kubik Laboratory for Neuropathology, Massachusetts General Hospital, Charlestown, MA, USA. [430]Department of Neuroscience, Ronald M. Loeb Center for Alzheimer's Disease, Icahn School of Medicine at Mount Sinai, New York, NY, USA. [431]Department of Health Behavior and Health Systems, University of North Texas Health Science Center, Fort Worth, TX, USA. [432]Department of Health Management and Policy, School of Public Health, University of North Texas Health Science Center, Fort Worth, TX, USA. [433]Department of Pathology and Laboratory Medicine, Emory University, Atlanta, GA, USA. [434]Emory Alzheimer's Disease Center, Emory University, Atlanta, GA, USA. [435]Department of Pathology and Laboratory Medicine, Indiana University, Indianapolis, IN, USA. [436]Department of Radiology, University of Washington, Seattle, WA, USA. [437]Center for Spatial and Functional Genomics, Division of Human Genetics, Children's Hospital of Philadelphia, Philadelphia, PA, USA. [438]Department of Pediatrics, Perelman School of Medicine, University of Pennsylvania, Philadelphia, PA, USA. [439]Department of Genetics, Perelman School of Medicine, University of Pennsylvania, Philadelphia, PA, USA. [440]Division of Genetics, Department of Medicine and Partners Center for Personalized Genetic Medicine, Brigham and Women's Hospital and Harvard Medical School, Boston, MA, USA. [441]Department of Neurology, Massachusetts General Hospital/Harvard Medical School, Boston, MA, USA. [442]Center for Applied Genomics, Children's Hospital of Philadelphia, Philadelphia, PA, USA. [443]Division of Human Genetics, Department of Pediatrics, Perelman School of Medicine, University of Pennsylvania, Philadelphia, PA, USA. [444]Department of Pathology (Neuropathology), University of Pittsburgh, Pittsburgh, PA, USA. [445]Department of Neurology, University of Alabama at Birmingham, Birmingham, AL, USA. [446]Department of Pathology and Laboratory Medicine, University of California Irvine, Irvine, CA, USA. [447]Department of Epidemiology and Population Health, Stanford University, Stanford, CA, USA. [448]Department of Neurology and Neurological Sciences, Stanford University, Stanford, CA, USA. [449]Vanderbilt Memory and Alzheimer's Center, Department of Neurology, Vanderbilt University Medical Center, Nashville, TN, USA. [450]Department of Surgery, University of Texas Southwestern Medical Center, Dallas, TX, USA. [451]Neurogenomics Division, Translational Genomics Research Institute, Phoenix, AZ, USA. [452]Department of Pathology, Duke University, Durham, NC, USA. [453]Department of Neurology, University of Texas Southwestern Medical Center, Dallas, TX, USA. [454]Department of Neurological Surgery, University of Texas Southwestern Medical Center, Dallas, TX, USA. [455]Hope Center Program on Protein Aggregation and Neurodegeneration, Washington University School of Medicine, St. Louis, MO, USA. [456]Department of Genome Sciences, University of Washington, Seattle, WA, USA. [457]Department of Pathology and Laboratory Medicine, University of California, Davis, Sacramento, CA, USA. [458]Department of Human Genetics, University of Pittsburgh, Pittsburgh, PA, USA. [459]Alzheimer's Disease Research Center, University of Pittsburgh, Pittsburgh, PA, USA. [460]Department of Neurology, Albert Einstein College of Medicine, New York, NY, USA. [461]Department of Neurology, Oregon Health and Science University, Portland, OR, USA. [462]Department of Neurology, Portland Veterans Affairs Medical Center, Portland, OR, USA. [463]Department of Laboratory Medicine and Pathology, University of Washington, Seattle, WA, USA. [464]Department of Pathology, Boston

University, Boston, MA, USA. [465]Department of Neuropsychology, University of California San Francisco, San Francisco, CA, USA. [466]Department of Neurobiology and Behavior, University of California Irvine, Irvine, CA, USA. [467]Department of Neurology, Emory University, Atlanta, GA, USA. [468]Kaiser Permanente Washington Health Research Institute, Seattle, WA, USA. [469]Cleveland Clinic Lou Ruvo Center for Brain Health, Cleveland Clinic, Cleveland, OH, USA. [470]Department of Pathology, University of Michigan, Ann Arbor, MI, USA. [471]National Center for PTSD at Boston VA Healthcare System, Boston, MA, USA. [472]Department of Psychiatry, Johns Hopkins University, Baltimore, MD, USA. [473]Department of Medicine (Pulmonary), New York University, New York, NY, USA. [474]Department of Neurology, Miller School of Medicine, University of Miami, Miami, FL, USA. [475]Department of Pathology, University of California San Diego, La Jolla, CA, USA. [476]School of Nursing Northwest Research Group on Aging, University of Washington, Seattle, WA, USA. [477]Pfizer Worldwide Research and Development, New York, NY, USA. [478]Weill Institute for Neurosciences, Memory and Aging Center, University of California, San Francisco, San Francisco, CA, USA. [479]Department of Pathology, University of Southern California, Los Angeles, CA, USA. [480]Department of Pathology, Stanford University School of Medicine, Stanford, CA, USA. [481]Department of Neurology, Washington University at St. Louis, St. Louis, MO, USA. [482]Institute for Translational Research, University of North Texas Health Science Center, Fort Worth, TX, USA. [483]Center for Mind and Brain and Department of Neurology, University of California, Davis, Sacramento, CA, USA. [484]Center for Population Health and Aging, Texas A&M University Health Science Center, Lubbock, TX, USA. [485]Department of Family and Community Medicine, University of Texas Health Science Center San Antonio, San Antonio, TX, USA. [486]Department of Laboratory Medicine and Pathology, Mayo Clinic, Rochester, MN, USA. [487]Department of Psychiatry and Behavioral Sciences, University of Washington School of Medicine, Seattle, WA, USA. [488]Department of Bioengineering, University of Pennsylvania, Philadelphia, PA, USA. [489]Alzheimer's Disease Center, New York University, New York, NY, USA. [490]Department of Neurology, University of Colorado School of Medicine, Aurora, CO, USA. [491]Department of Internal Medicine and Geriatrics, University of North Texas Health Science Center, Fort Worth, TX, USA. [492]Department of Medical Education, TCU/UNTHSC School of Medicine, Fort Worth, TX, USA. [493]Arizona Alzheimer's Consortium, Phoenix, AZ, USA. [494]Banner Alzheimer's Institute, Phoenix, AZ, USA. [495]Department of Psychiatry, University of Arizona, Phoenix, AZ, USA. [496]Tanz Centre for Research in Neurodegenerative Disease, University of Toronto, Toronto, Ontario, Canada. [497]Department of Radiology and Imaging Sciences, Indiana University, Indianapolis, IN, USA. [498]Department of Pathology (Neuropathology), Rush University Medical Center, Chicago, IL, USA. [499]Department of Psychiatry, University of Southern California, Los Angeles, CA, USA. [500]Cambridge Institute for Medical Research, University of Cambridge, Cambridge, UK. [501]Faculty of Medicine, Department of Medicine (Neurology), University of Toronto, Toronto, Ontario, Canada. [502]Center for Applied Health Research, Baylor Scott & White Health, Temple, TX, USA. [503]College of Medicine, Texas A&M University Health Science Center, College Station, TX, USA. [504]Program in Cellular Neuroscience, Neurodegeneration and Repair, Yale University School of Medicine, New Haven, CT, USA. [505]Department of Psychiatry and Human Behavior, University of California, Irvine, Irvine, CA, USA. [506]Renaissance Computing Institute, University of North Carolina at Chapel Hill, Chapel Hill, NC, USA. [507]Department of Pathology, Johns Hopkins University, Baltimore, MD, USA. [508]Department of Pathology and Laboratory Medicine, University of California, Los Angeles, Los Angeles, CA, USA. [509]Department of Psychiatry and Behavioral Sciences, Northwestern University Feinberg School of Medicine, Chicago, IL, USA. [510]Department of Psychiatry and Behavioral Sciences, Duke University, Durham, NC, USA. [511]Department of Biostatistics, University of Washington, Seattle, WA, USA. [512]Department of Psychiatry, New York University Grossman School of Medicine, New York, NY, USA. [513]Center for Cognitive Neurology and Departments of Neurology and Pathology, New York University Grossman School of Medicine, New York, NY, USA. [514]Department of Pathology, Oregon Health and Science University, Portland, OR, USA. [515]Evelyn F. McKnight Brain Institute, Department of Neurology, Miller School of Medicine, University of Miami, Miami, FL, USA. [516]Department of Pathology, Case Western Reserve University, Cleveland, OH, USA.

## CHARGE

**Hieab Adams[1], Rufus O. Akinyemi[517], Muhammad Ali[518], Nicola Armstrong[519], Hugo J. Aparicio[109], Maryam Bahadori[130], James T. Becker[520], Monique Breteler[521], Daniel Chasman[522], Ganesh Chauhan[523], Hata Comic[12], Simon Cox[524], Adrienne L. Cupples[21], Gail Davies[525], Charles S. DeCarli[526], Marie-Gabrielle Duperron[527], Josée Dupuis[21], Tavia Evans[12], Frank Fan[528], Annette Fitzpatrick[529], Alison E. Fohner[23], Mary Ganguli[528], Mirjam Geerlings[530], Stephen J. Glatt[531], Hector M. Gonzalez[532], Monica Goss[130], Hans Grabe[533], Mohamad Habes[534], Susan R. Heckbert[535], Edith Hofer[536], Elliot Hong[537], Timothy Hughes[130], Xueqiu Jian[130], Tiffany F. Kautz[538], Maria Knol[12], William Kremen[12], Paul Lacaze[539], Jari Lahti[540], Quentin Le Grand[527], Elizabeth Litkowski[541], Shuo Li[21], Dan Liu[521], Xuan Liu[21], Marisa Loitfelder[542], Alisa Manning[543], Pauline Maillard[544], Riccardo Marioni[545], Bernard Mazoyer[546], Debora Melo van Lent[130], Hao Mei[547], Aniket Mishra[527], Paul Nyquist[548], Jeffrey O'Connell[537], Yash Patel[549], Tomas Paus[550], Zdenka Pausova[551], Katri Raikkonen-Talvitie[540], Moeen Riaz[539], Stephen Rich[539], Jerome Rotter[552], Jose Romero[553], Gena Roshchupkin[12], Yasaman Saba[554], Murali Sargurupremraj[554], Helena Schmidt[554], Reinhold Schmidt[555], Joshua M. Shulman[556], Jennifer Smith[557], Hema Sekhar[527], Reddy Rajula[527], Jean Shin[558], Jeannette Simino[559], Eeva Sliz[559], Alexander Teumer[560], Alvin Thomas[561], Adrienne Tin[559], Elliot Tucker-Drob[562], Dina Vojinovic[12], Yanbing Wang[9], Galit Weinstein[563], Dylan Williams[564], Katharina Wittfeld[565], Lisa Yanek[548] and Yunju Yang[566]**

[517]Centre for Genomic and Precision Medicine, College of Medicine, UI, Ibadan, Nigeria. [518]Washington University at St. Louis, St. Louis, MO, USA. [519]Mathematics and Statistics, Curtin University, Perth, Western Australia, Australia. [520]Departments of Psychiatry, Neurology, and Psychology, University of Pittsburgh, Pittsburgh, PA, USA. [521]Population Health Sciences, German Center for Neurodegenerative Diseases (DZNE), Bonn, Germany. [522]Brigham and Women's Hospital, Harvard University, Boston, MA, USA. [523]INSERM U1219, University of Bordeaux, Bordeaux, France. [524]University of Edinburgh,

Edinburgh, UK. [525]Centre for Cognitive Ageing and Cognitive Epidemiology, University of Edinburgh, Edinburgh, UK. [526]Department of Neurology and Center for Neuroscience, University of California, Davis, Davis, CA, USA. [527]Bordeaux Population Health Research Center, Team VIN-TAGE, UMR 1219, University of Bordeaux, INSERM, Bordeaux, France. [528]University of Pittsburgh, Pittsburgh, PA, USA. [529]Department of Family Medicine, University of Washington, Seattle, WA, USA. [530]University Medical Center Utrecht, Utrecht, the Netherlands. [531]Psychiatric Genetic Epidemiology & Neurobiology Laboratory (PsychGENe Lab), Department of Psychiatry and Behavioral Sciences, SUNY Upstate Medical University, Syracuse, NY, USA. [532]University of California, San Diego, San Diego, CA, USA. [533]Department of Psychiatry and Psychotherapy, University Medicine Greifswald, Greifswald, Germany. [534]Department of Radiology, University of Texas Health Science Center at San Antonio, San Antonio, TX, USA. [535]School of Public Health, University of Texas Health Science Center at Houston, Houston, TX, USA. [536]Clinical Division of Neurogeriatrics, Department of Neurology, Medical University of Graz, Graz, Austria. [537]University of Maryland, College Park, MD, USA. [538]Department of General Medicine, University of Texas Health Science Center, San Antonio, TX, USA. [539]Monash University Clayton Campus, Mebourne, Victoria, Australia. [540]University of Helsinki, Helsinki, Finland. [541]University of Colorado Anschutz Medical Center, Aurora, CO, USA. [542]Medical University of Graz, Graz, Austria. [543]Massachusetts General Hospital, Harvard University, Cambridge, MA, USA. [544]Imaging of Dementia and Aging (IDeA) Laboratory, Department of Neurology, University of California, Davis, Davis, CA, USA. [545]University of Staffmail, Edinburgh, UK. [546]University of Bordeaux, IMN, Bordeaux, France. [547]University of Mississippi Medical Center, Jackson, MS, USA. [548]GeneSTAR Research Program, Department of Neurology, Johns Hopkins University School of Medicine, Baltimore, MD, USA. [549]University of Toronto, Toronto, Ontario, Canada. [550]Departments of Psychiatry & Neuroscience, Centre Hospitalier Universitaire Saint-Justine, University of Montreal, Montreal, Quebec, Canada. [551]Hospital for Sick Children, University of Toronto, Toronto, Ontario, Canada. [552]Institute for Translational Genomics and Population Sciences, Los Angeles Biomedical Research Institute and Pediatrics at Harbor-UCLA Medical Center, Torrance, CA, USA. [553]Boston Medical Center, Boston, MA, USA. [554]Gottfried Schatz Research Center, Department of Molecular Biology and Biochemistry, Medical University of Graz, Graz, Austria. [555]Clinical Division of Neurogeriatrics, Department of Neurology, Medical University of Graz, Graz, Austria. [556]Departments of Neurology, Molecular & Human Genetics, and Neuroscience and Program in Developmental Biology, Baylor College of Medicine, Houston, TX, USA. [557]Department of Epidemiology, School of Public Health, University of Michigan, Ann Arbor, MI, USA. [558]Hospital for Sick Children, University of Toronto, Toronto, Ontario, Canada. [559]University of Mississippi Medical Center, Jackson, MS, USA. [560]Institute for Community Medicine, University Medicine Greifswald, Greifswald, Germany. [561]University of North Carolina, Chapel Hill, NC, USA. [562]University of Texas, Austin, TX, USA. [563]Clinical Division of Neurogeriatrics, Department of Neurology, Medical University of Graz, Graz, Austria. [564]Karolinska Institute, Stockholm, Sweden. [565]German Center for Neurodegenerative Diseases (DZNE), Site Rostock/Greifswald, Greifswald, Germany. [566]Institute of Molecular Medicine, University of Texas Health Science Center at Houston McGovern Medical School, Houston, TX, USA.

## Methods

**Samples.** All of our stage I meta-analysis samples came from the following consortia/datasets: EADB, GR@ACE, EADI, GERAD/PERADES, DemGene, Bonn, the Rotterdam study, the CCHS study, NxC and the UKBB. In the UKBB, individuals who did not report dementia or any family history of dementia were used as controls; the analysis included 2,447 diagnosed cases, 46,828 proxy cases of dementia and 338,440 controls. All individuals included in stage I are of European ancestry; demographic data on these case–control studies are summarized in Supplementary Table 1, and more detailed descriptions are available in the Supplementary Note. Stage II samples are from the ADGC, CHARGE and FinnGen consortia (Supplementary Table 1 and Supplementary Note) and are described in detail elsewhere[5,6,9,10,54–56]. Written informed consent was obtained from study participants or, for those with substantial cognitive impairment, a caregiver, legal guardian or other proxy. Study protocols for all cohorts were reviewed and approved by the appropriate institutional review boards.

**Quality control and imputation.** A standard quality control was performed on variants and samples from all datasets individually. The samples were then imputed with the TOPMed reference panel[57,58]. The Haplotype Reference Consortium (HRC) panel[59] was also used for some datasets (Supplementary Table 2). For the UKBB, we used the provided imputed data generated from a combination of the 1000 Genomes, HRC and UK10K reference panels (Supplementary Note).

**Stage I analyses.** Tests of the association between clinical or proxy-ADD status and autosomal genetic variants were conducted separately in each dataset by using logistic regression and an additive genetic model, as implemented in SNPTEST 2.5.4-beta3 (ref. [60]) or PLINK v1.90 (ref. [4]). However, a logistic mixed model (as implemented in SAIGE v0.36.4 (ref. [61])) was considered for the UKBB data. We analyzed the genotype probabilities in SNPTEST (using the newml method) and dosages in PLINK and SAIGE. Analyses were adjusted for PCs and genotyping centers, when necessary (Supplementary Table 2). For the UKBB dataset, only variants with a MAF above 0.01% and a minor allele count (MAC) above 3 were analyzed, and effect sizes and standard errors were corrected by a factor of two, because proxy cases were analyzed[7]. This approach is appropriate for variants with a moderate-to-high frequency and a small effect size. For all datasets, we filtered out duplicated variants and variants with (1) missing data on the effect size, standard error or P value; (2) an absolute effect size above 5; (3) an imputation quality below 0.3; and (4) a value below 20 for the product of the MAC and the imputation quality (MAC-info score). For datasets not imputed with the TOPMed reference panel, we also excluded (1) variants for which conversion of position or alleles from the GRCh37 assembly to the GRCh38 assembly was not possible or problematic or (2) variants with very large difference of frequency between the TOPMed reference panel and the reference panels used to perform imputation.

Results were then combined across studies in a fixed-effect meta-analysis with an inverse-variance weighted approach, as implemented in METAL v2011-03-25 software[62]. We filtered out (1) variants with a heterogeneity P value below $5 \times 10^{-8}$, (2) variants analyzed in less than 20% of the total number of cases and (3) variants with frequency amplitude above 0.4 (defined as the difference between the maximum and minimum frequencies across all the studies). We also excluded variants not analyzed in the EADB-TOPMed dataset.

The genomic inflation factor lambda was computed with the GenABEL 1.8-0 R package[63] and a median approach after exclusion of the *APOE* region (44–46 Mb on chromosome 19 in GRCh38). The LD score regression intercept was computed with LDSC v1.0.1 software using the 'baselineLD' LD scores built from 1000 Genomes phase 3 (ref. [64]). The analysis was restricted to HapMap 3 variants and excluded multiallelic variants, variants without an rs ID and variants in the *APOE* region.

**Definition of associated loci.** A region of ±500 kb was defined around each variant with a stage I P value below $1 \times 10^{-5}$. These regions were then merged (using bedtools v2.27.0 software; https://bedtools.readthedocs.io/en/latest/) to define nonoverlapping regions. The region corresponding to the *APOE* locus was excluded. We then used the PLINK clumping procedure to define independent hits in each region. An iterative clumping procedure was applied to all variants with a stage I P value below $1 \times 10^{-5}$, starting with the variant with the lowest P value (referred to as the index variant). Variants with a stage I P value below $1 \times 10^{-5}$, located within 500 kb of this index variant and in LD with the index variant ($r^2$ above 0.001) were assigned to the index variant's clump. The clumping procedure was then applied until all the variants had been clumped. LD in the EADB-TOPMed dataset was computed using high-quality (probability ≥0.8) imputed genotypes.

**Stage II analyses.** Variants with a stage I P value below $1 \times 10^{-5}$ were followed up (Supplementary Note). Results were combined across all stage I and II studies in a fixed-effect meta-analysis with an inverse variance weighted approach, as implemented in METAL. In each clump, we then reported the variants with positive follow-up results (i.e., the same direction of effect in stage I and stage II, and a stage II P value below 0.05) and the lowest P value in the meta-analysis. Those variants were considered to be associated at the genome-wide significance

level if they had a P value below $5 \times 10^{-8}$ in the stage I and II meta-analysis. However, we excluded the chr6:32657066:G:A variant, because its frequency amplitude was high.

**Pathway analysis.** A total of 10,271 gene sets were considered for analysis (Supplementary Note). Gene set enrichment analyses were performed in MAGMA v1.08 (refs. [65,66]), with correction for the number of variants in each gene, LD between variants and LD between genes. LD was computed from the EADB-TOPMed dataset using high-quality (probability ≥0.9) imputed genotypes. The measure of pathway enrichment was the MAGMA 'competitive' test (in which the association statistic for genes in the pathway is compared with those for all other protein-coding genes), as recommended by De Leeuw et al.[67]. We used the 'mean' test statistic, which uses the sum of −log(variant P value) across all genes. The primary analysis assigned variants to genes if they lay within the gene boundaries, although a secondary analysis used a window of 35 kb upstream and 10 kb downstream to assign variants to genes (as in Kunkle et al.[5]). The primary analysis included all variants with an imputation quality above 0.8. We used q values[68] to account for multiple testing.

**Expression in various cell types.** The expression of genes was assigned to specific cell classes of the adult brain, as described previously[69]. Briefly, middle temporal gyrus single-nucleus transcriptomes from the Allen Brain Atlas dataset (49,555 total nuclei derived from 8 human tissue donors aged 24–66 years) were used to annotate and select six main cell classes using Seurat 3.1.1 (ref. [70]): glutamatergic neurons, GABAergic neurons, astrocytes, oligodendrocytes, microglia and endothelial cells. Enrichment analyses were performed by using the mean gene expression per nucleus for each cell type relative to the total expression summed across cell types as a quantitative covariate in a MAGMA gene property analysis.

**Functional interpretation of GWAS signals and gene prioritization.** To prioritize candidate genes in the new loci, we systematically searched for evidence for these genes in seven different domains: (1) variant annotation, (2) eQTL-GWAS integration, (3) sQTL-GWAS integration, (4) protein QTL (pQTL)-GWAS integration, (5) mQTL-GWAS integration, (6) histone acetylation QTL (haQTL)-GWAS integration and (7) APP metabolism. On the basis of this evidence, we then defined a gene prioritization score of between 0 and 100 for each candidate gene (Supplementary Fig. 34). Detailed information on the domains, categories (e.g., the tissue or cell type for QTL-GWAS integration domains) and subcategories (for the type of evidence) is given in Supplementary Table 19. A brief summary of how evidence was assessed in each domain is provided below, together with a detailed description of the gene prioritization strategy.

*Candidate genes.* We considered protein-coding candidate genes within a ±1-Mb window of the new lead variants. The genes in overlapping loci (i.e., L28, L30 and L37) were assigned to their respective loci based on proximity to the lead variants, and the distal genes were not considered for gene prioritization in the investigated loci. Moreover, we did not perform gene prioritization in the complex IGH gene cluster locus (L27), as this telomeric region contains complex splicing events (spanning a high number of IGH genes) that probably result from known fusion events[18].

*The variant annotation domain.* In this domain, we determined whether the candidate gene was the nearest protein-coding gene to the lead variant and/or whether the lead variant was a rare variant (MAF < 1%) and/or protein-altering variant of the investigated candidate gene.

*Molecular QTL–GWAS integration domains.* To study the downstream effects of new ADD-associated variants on molecular phenotypes (i.e., expression, splicing, protein expression, methylation and histone acetylation) in various AD-relevant tissues, cell types and brain regions, molecular *cis*-QTL information (i.e., the genetic variants that regulate these molecular phenotypes) was integrated with the stage I ADD GWAS results in genetic colocalization analyses, TWASs and a genetically driven DNA methylation scan. These molecular QTLs include eQTLs, sQTLs, pQTLs, mQTLs and haQTLs. We mapped and prepared eQTL/sQTL catalogs in AD-relevant bulk brain regions from AMP-AD cohorts[71–74] and in LCLs from the EADB Belgian cohort. We used additional eQTL/sQTL information in AD-relevant bulk brain regions from GTEx[75] and microglia from the MiGA study[76]. Furthermore, eQTLs in monocytes and macrophages from various datasets[77–82] (as prepared by eQTL Catalogue[83]) were included in the analyses. Data on pQTLs[84], mQTLs[85] and haQTLs[85] were available for DLPFC. Using each molecular QTL catalogue, the effect of the lead variants was queried and significant associations were reported. Moreover, genetic colocalization studies were conducted by comparing ADD association signals with the eQTL/sQTL signals from AMP-AD bulk brain, MiGA microglia and EADB LCL cohorts. We also conducted eTWASs and splicing TWASs (sTWAS) of the ADD risk, along with fine mapping of the eTWAS results. To this end, we trained functional expression and splicing reference panels based on the AMP-AD bulk brain and EADB LCL cohorts, and we leveraged precalculated reference panel weights[86] for the GTEx dataset[75] in tissues and cells of interest. Lastly, for the mQTL-GWAS integration

domain, we also tested for associations between ADD and genetically driven DNA methylation (MetaMeth analysis) in blood (with blood–brain methylation correlation estimates obtained from BECon[87]) using the procedures described by Freytag et al.[88] and Barbeira et al.[89]. A detailed description of the datasets and methods used for each of these analyses is given in the Supplementary Note.

*APP metabolism domain.* We assessed the functional impact of gene underexpression on APP metabolism for all candidate genes based on a genome-wide high-content short interfering RNA screen[17] (Supplementary Note).

*Gene prioritization score.* We computed a gene prioritization score for each candidate gene as the weighted sum of the evidence identified in the seven domains. We specified a weight for each type of evidence, as detailed in Supplementary Table 19. For the molecular QTL-GWAS integration domains, we gave more weight to replicated hits (i.e., evidence in several datasets) than to single hits. We also gave more weight to hits observed in brain (the bulk brain and microglia datasets) than to hits observed in other tissues/cell types (LCLs, monocytes, macrophages and blood). To avoid score inflation, several specific rules were applied: (1) for the results of sQTL- and mQTL-based analyses, multiple splice junctions or CpGs annotated for the same genes were aggregated prior to weighting due to correlated data; (2) if we observed a fine-mapped eTWAS association for a gene, its other significant (but not fine-mapped) eTWAS associations were not considered; (3) for genes having several significant CpGs (prior to aggregation) in MetaMeth analyses, the associated CpGs with a low (<75% percentile) blood–brain methylation correlation estimate were not considered if the gene also had associated CpGs with a high (≥75% percentile) blood–brain methylation correlation estimate.

*Gene prioritization strategy.* After obtaining a total weighted score per gene, we ranked genes per locus according to their prioritization scores and compared the relative score differences between the highest ranked gene and other genes in the investigated locus. If this relative difference was at least 20% and the gene prioritization score for the highest ranked gene was ≥4, then we classified this gene as a tier 1 prioritized gene in the investigated locus (i.e., a greater likelihood of being the true risk gene responsible for the ADD signal). If this absolute threshold was not met, then the highest ranked gene was classified as a tier 2 prioritized gene (i.e., a lower level of confidence and absence of the minimum level of evidence for a true risk gene). Furthermore, other genes in a locus harboring a tier 1 gene were classified as tier 2 prioritized genes if the relative score difference versus the highest ranked (tier 1) gene was between 20% and 50%. Lastly, when the relative score difference between the highest ranked gene and other genes in the same locus was <20%, then both the highest ranked gene and all genes with a score difference <20% were classified as tier 2 prioritized genes in the investigated locus; based on the current evidence, it is difficult to prioritize two or more similarly scored genes. The gene prioritization strategy is summarized in Supplementary Fig. 34. Detailed descriptions and discussions of prioritized genes and tier levels in each investigated new locus can be found in the Supplementary Note.

*GRS analysis.* Eight longitudinal MCI cohorts and seven population-based studies were included in the analysis and are fully described in the Supplementary Note and Supplementary Table 33. The GRSs were calculated as previously described[90]. Briefly, we considered variants with genome-wide significant evidence of association with ADD in our study. We did not include any *APOE* variants in the GRS. Variants were directly genotyped or imputed ($R^2 \geq 0.3$). Imputation was performed using the HRC panel[59] for subcohorts from the Rotterdam study and the TOPMed panel for the other cohorts[57]. For HRC-imputed data, LD proxies were considered for variants that were not available in this reference panel. The GRS was calculated as the weighted average of the number of risk-increasing alleles for each variant, using dosages. Weights were based on the respective log(OR) obtained in stage II. The GRS was then multiplied by the number of included variants. Thus, the HR measured the effect of carrying one additional average risk allele.

To assess whether the new variants in this study contribute to the risk of conversion to AD (in addition to known AD genes), we calculated two GRSs: one based solely on variants known before this study (GRS$_{known}$, $n=39$; Table 1) and another based on variants identified in the present study (GRS$_{novel}$, $n=44$; Table 2). These GRSs were calculated in the same way as the GRS encompassing all the variants.

The association between the GRS and the risk of progression to dementia in individuals from population-based cohorts or patients with MCI from memory clinics was tested statistically using Cox proportional hazards models. The models were adjusted for age, sex, the first four PCs (to correct for potential population stratification) and the number of *APOE*-ε4 and *APOE*-ε2 alleles (assuming an additive effect). In the FHS study, the generation was used as an additional covariate. In the 3C study, the analysis was adjusted for age, sex, the number of *APOE* alleles, the two first PCs and center. The PCs used were generated for each cohort, using the same variants as in the case/control study's PC analysis. The number of *APOE*-ε4 alleles was obtained from direct genotyping or, if missing, the genotypes (with probability >0.8) derived from the TOPMed imputations. The interaction between the GRS and the number of *APOE*-ε4 alleles was tested on

the multiplicative scale. In the primary analysis, conversion to AD was used as the outcome (conversions to non-AD dementias were coded as being censored at time of conversion), but analyses were repeated using all-cause dementia as the outcome.

To quantify the effect size of the potential association between the GRS and conversion to dementia regarding predictive performance, we computed three different indices measuring different aspects of the predictive performance of the GRS in our prospective, longitudinal cohort studies[91]: the continuous version of the C-index,[92,93] the continuous NRI[94] and IPA[95] (Supplementary Note). For all indices, we provide point estimates and 95% CIs.

In the main analysis, indices were computed at the time point for which all cohorts in a specific setting (i.e., population-based studies or memory clinics, respectively) provided follow-up observations (that is 5 years for population-based cohorts and 3 years for MCI cohorts). In a sensitivity analysis, indices for longer or shorter follow-up periods were also derived (that is 3 years and 10 years for population-based cohorts and 5 years for MCI cohorts). Standard errors for indices were derived by non-parametric bootstrapping with 1,000 samples.

To determine the average effect of the GRS across the various cohorts examined, individual cohort results were subjected to both inverse-variance weighted meta-analyses (primary analyses) and random effects meta-analysis (Supplementary Note). To facilitate comparisons of results for different time points, cohorts with longer follow-up periods were meta-analyzed separately. Furthermore, two memory clinic cohorts with a limited sample size ($N<50$) were excluded to assess their impact on the final meta-analysis results. Meta-analyses were performed using the 'metafor' (3.0.2) R package[96].

To further illustrate the clinical relevance of the GRS, we pooled computed GRSs across four population-based cohorts (3C, AgeCoDe, VITA and MAS) and computed deciles of the GRS distribution for use as a common reference for all cohorts. We then computed the increase in risk when augmenting the GRS value from the first decile (GRS = 50.76) to the ninth decile (GRS = 59.74) of the distribution. To represent this risk increase in the HR, we rescaled the HR derived from our meta-analyses results using the equation $e^{\log(HR)*(GRS9th_{decile}-GRS1st_{decile})}$. Importantly, this approach yields exactly the same results as transforming the GRS so that a one unit increment corresponds to the increase from the lowest decile to the highest decile.

Furthermore, we approximated the probability of conversion to AD at 3 and 5 years in memory clinic patients with MCI by using Cox models implemented in the 'PredictCox' function from the 'riskRegression' (2020.12.8) R package[97]. We did not derive AD conversion probabilities for two cohorts with very small sample sizes ($N<50$). Predicted AD conversion probabilities were derived and averaged for all patients in each of the groups formed by the decile of the GRS distribution in each cohort. The difference between the groups with the highest and lowest GRSs was computed in each cohort. We report the median (range) results in each group formed by the GRS deciles.

**Reporting Summary.** Further information on research design is available in the Nature Research Reporting Summary linked to this article.

## Data Availability

Genome-wide summary statistics have been deposited to the European Bioinformatics Institute GWAS Catalog (https://www.ebi.ac.uk/gwas/) under accession no. GCST90027158.

The significant eQTLs/sQTLs mapped and eTWAS/sTWAS functional reference panel weights generated for this study (in AD-relevant bulk brain regions from AMP-AD cohorts and in LCLs from the EADB Belgian cohort) are publicly available at https://doi.org/10.5281/zenodo.5745927 and https://doi.org/10.5281/zenodo.5745929.

Anonymized aligned reads of the amplicon-based long-read Nanopore cDNA sequencing experiment conducted for the *TSPAN14* splicing analysis are available through the European Nucleotide Archive under accession PRJEB49234.

Moreover, the following data used in the gene prioritization are publicly available: AMP-AD rnaSeqReprocessing Study (https://www.synapse.org/#!Synapse:syn9702085);

MayoRNAseq whole-genome sequencing variant call formats (WGS VCFs) (https://www.synapse.org/#!Synapse:syn11724002);

ROSMAP WGS VCFs (https://www.synapse.org/#!Synapse:syn11724057);

MSBB WGS VCFs (https://www.synapse.org/#!Synapse:syn11723899);

eQTLGen (https://www.eqtlgen.org/);

eQTL Catalogue database (https://www.ebi.ac.uk/eqtl/);

Brain xQTL serve (http://mostafavilab.stat.ubc.ca/xqtl/);

GTEx v8 eQTL and sQTL catalogs (https://www.gtexportal.org/);

GTEx v8 expression and splicing prediction models (http://predictdb.org/);

MiGA eQTLs (https://doi.org/10.5281/zenodo.4118605);

MiGA sQTLs (https://doi.org/10.5281/zenodo.4118403);

MiGA meta-analysis (https://doi.org/10.5281/zenodo.4118676); and

Wingo et al.[84] pQTL data (https://www.synapse.org/#!Synapse:syn23627957).

## Code availability

We used publicly available software for all analyses. The software are listed in the Supplementary Note with their appropriate citations and/or URLs.

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

## Acknowledgements

We thank the many study participants, researchers and staff for collecting and contributing to the data, the high-performance computing service at the University of Lille and the staff at CEA-CNRGH for their help with sample preparation and genotyping and excellent technical assistance. We thank Antonio Pardinas for his help. We thank the Netherlands Brain Bank. This research was conducted using the UKBB resource (application number 61054). This work was funded by a grant (EADB) from the EU Joint Programme – Neurodegenerative Disease Research. INSERM UMR1167 is also funded by the INSERM, Institut Pasteur de Lille, Lille Métropole Communauté Urbaine and French government's LABEX DISTALZ program (development of innovative strategies for a transdisciplinary approach to AD). Full consortium acknowledgements and funding are in the Supplementary Note.

## Author contributions

EADB coordination: K. Mather, F.J., M.T., R.F.-S., J. Clarimon, J.-F. Deleuze, O.A.A., M.I., M. Hiltunen, K.S., C.M.v.D., R.S., W.M.v.d.F., A. Ruiz, A. Ramirez and J.-C.L. Data analyses: C. Bellenguez, F.K., I.E.J., L.K., S.M.-G., N.A., R.C., B.G.-B., V. Andrade, P.A.H., R.C.-M., V.D., S.J.v.d.L., M.R.C., T.K., I.R., J. Chapuis and P.G.-G. ADGC analysis and coordination: A.C.N., W.S.B., L.A.F., J.L.H., K.L.H.-N., P.P.K., B.W.K., C.-Y.L. and E.R.M., R. Mayeux, M.A.P.-V., J.S., L.-S.W., Y.Z. and G.D.S. Charge analysis and coordination: Q.Y., J.C.B., A.D.S., C.S., B.M.P., R.W., O. Lopez. and S. Seshadri. FinnGen analysis: T.K. and M. Hiltunen. Rotterdam analysis: A.Y., I.P.-N, M. Ghanbari and M.A.I. Sample contribution: S. Ahmad, V. Giedraitis, D. Aarsland, P.V.Ã., D.G.-G., C. Abdelnour, E.A.-M., D. Alcolea, M. Alegret, I. Alvarez, V. Alvarez, N.J.A., A. Tsolaki, C. Antúnez, I. Appollonio, M. Arcaro, S. Archetti, A.A.P., B.A., L.A., H. Bailly, N.B., M. Baquero, S. Barral, A. Beiser, A.B.P., J.E.B., P. Benchek, L.B., C. Berr, C. Besse, V. Bessi, G. Binetti, A. Bizarro, R.B., M. Boada, E.B., B.B., S. Boschi, P. Bossù, G. Bråthen, J.B., C. Bresner, H. Brodaty, K.J.B., L.I.B., D.B.-R., K.B., V. Burholt, W.S.B., M.C., L.B.C., G.C., J. Chung, M.L.C., Ã.C., R.C., L.C.-C., C. Charbonnier, H.-H.C., C. Chilotti, S.C., J.A.C., C. Clark, E. Conti, A.C.-G., E. Costantini, C. Custodero, D.D., M.C.D., A. Daniele, E. Dardiotis, J-F. Dartigues, P.P.d.D., K.d.P.L., L.D.d.W., S. Debette, J.D., T.d.S., N.D., A. DeStefano, M.D., J.D.-S., M.D.-F., P.D.R., S. Djurovic, E. Duron, E. Duzel, C.D., G.E., S.E., V.E.-P., A.E., M.E., K.M.F., T. Fabrizio, S.F.N., D.W.F., L. Farotti, C.F., M.F.-F., R.F., C.B.F., E.F., B. Fin, P.F., T. Fladby, K.F., B. Fongang, M.F., J.F., T.M.F., S.F., N.C.F., E.F.-M., M.J.B., A.F.-G., L. Froelich, B.F.-H., D.G., J.M.G.-A., S.G.-M., G.G.-R., R.G., I.G., G. Giorgio, A.M.G., O.G., D.G.-F., A.G.-P., C.G., G. Grande, E. Green, T.G., E. Grunblatt, M. Grunin, V. Gudnason, T.G.-B., A.H., G.H., J.L.H., K.L.H.-N., H. Hampel, O.H., J. Hardy, A.M.H., L.H., J. Harwood, S.H.-H., S.H., M.T.H., I.H., M.J.H., P.H., C.H., H. Holstege, R.H.V., M. Hulsman, J. Humphrey, G.J.B., X.J., C.J., G.R.J., Y.K., J. Kauwe, P.G.K., L. Kilander, A.K.S., M.K., A.K., J. Kornhuber, M.H.K., W.A.K., P.P.K., B.W.K., A.B.K., C.L., E.J.L., L. Launer, A. Lauria, C.-Y.L., J.L., O.Ler., A. Lleó, W.L.J., O. Lopez, A.L.d.M., S.L., M.L., L. Luckcuck, K.L.L., Y.M., J.M., C.A.M., W.M., F. Mangialasche, M. Spallazzi, M. Marquié, R. Marshall, E.R.M., A.M.M., C.M.R., C. Masullo, R. Mayeux, S. Mead, P. Mecocci,

M. Menéndez-González, A.M., S. Mehrabian, S. Mendoza, M.M.-G., P. Mir, S. Moebus, M. Mol, L.M.-P., L. Montrreal, L. Morelli, F. Moreno, K. Morgan, T. Mosley, M.M.N., C. Muchnik, S. Mukherjee, B.N., T.N., G.N., B.G.N., R.O., A.O., M.O., G.O., A.P., C. Paollo., G. Papenberg, L.P., F.P., P. Pastor, G. Peloso, A.P.-C., J.P.-T., P. Pericard, O.P., Y.A.P., J.A.P., G.P.-R., C. Pisanu, T.P., J. Popp, D.P., J. Priller, R.P., O.Q., I.Q., J.Q.T., A. Rábano, I. Rainero, F.R., I. Ramakers, L.M.R., M.J.R., C.R., D.R.-D., P. Ridge, S.R.-H., P. Riederer, N.R., E.R.-R., A. Rongve, I.R.A., M.R.-R., J.L.R., E.R., D.R., M.E.S., P. Sakka, I.S., Ā.S., M.B.S.-A., F.S.-G., P.S.J., R.S.-V., S.B.S., C.S., C.L.S., M. Scamosci, N. Scarmeas, E. Scarpini, P. Scheltens, N. Scherbaum, M. Scherer, M. Schmid, A. Schneider, J.M.S., G. Selbæk, D.S., M. Serrano, J.S., A.A.S., O.S., S. Slifer, G.J.L.S., H.S., V.S., A. Solomon, Y.S., S. Sorbi, O.S.-G., G. Spalletta, A. Spottke, A. Squassina, E. Stordal, J.P.T., L. Tárraga, N.T., A. Thalamuthu, T.T., G.T., L. Traykov, L. Tremolizzo, A.T.-H., A. Uitterlinden, A. Ullgren, I.U., S.V., O.V., C.V.B., J. Vance, B.N.V., A.v.d.L., J.V.D., J.v.R., J.v.S., R.V., F.V., J.-S.V., J. Vogelgsang, M.V., M.W., D.W., L.-S.W., R.W., L.W., J. Wiltfnag, G.W., B.W., M.Y., H.Z., Y.Z., X.Z., C.Z., M.Z., L.A.F., B.M.P., M. Ghanbari, T.R., P. Sachdev, K. Mather, F.J., M.A.I., A.d.M., J. Hort, M.T. and M.A.P.-V. Core writing group: C. Bellenguez, F.K., V. Andrade, B.G.-B., P.A.H., R.C.-M., L.K., S.J.v.d.L., K.S., A. Ruiz, A. Ramirez and J.-C.L.

## Competing interests

H. Hampel is an employee of Eisai. The present article was initiated and prepared as part of his academic position at Sorbonne University (Paris, France), and it reflects entirely and exclusively his own opinion. He serves as Senior Associate Editor for the *Alzheimers & Dementia* journal and has not received any fees or honoraria since May 2019. Before May 2019, H. Hampel received lecture fees from Servier, Biogen and Roche; research grants from Pfizer, Avid and MSD Avenir (paid to the institution); travel funding from Eisai, Functional Neuromodulation, Axovant, Eli Lilly and Company, Takeda, Zinfandel Pharmaceuticals, GE Healthcare and Oryzon Genomics; and consultancy fees from Qynapse, Jung Diagnostics, Cytox, Axovant, Anavex, Takeda, Zinfandel Pharmaceuticals, GE Healthcare, Oryzon Genomics and Functional Neuromodulation. He served as a scientific advisory board member for Functional Neuromodulation, Axovant, Eisai, Eli Lilly and Company, Cytox, GE Healthcare, Takeda and Zinfandel, Oryzon Genomics and Roche Diagnostics. The remaining authors declare no competing interests

## Additional information

**Correspondence and requests for materials** should be addressed to Céline Bellenguez or Jean-Charles Lambert.

**Peer review file** *Nature Genetics* thanks the anonymous reviewers for their contribution to the peer review of this work. Peer reviewer reports are available.

                            Jean-Charles Lmabert

# Reporting Summary

Nature Research wishes to improve the reproducibility of the work that we publish. This form provides structure for consistency and transparency in reporting. For further information on Nature Research policies, see our Editorial Policies and the Editorial Policy Checklist.

## Statistics

For all statistical analyses, confirm that the following items are present in the figure legend, table legend, main text, or Methods section.

| n/a | Confirmed | |
|---|---|---|
| ☐ | ☒ | The exact sample size (*n*) for each experimental group/condition, given as a discrete number and unit of measurement |
| ☐ | ☒ | A statement on whether measurements were taken from distinct samples or whether the same sample was measured repeatedly |
| ☐ | ☒ | The statistical test(s) used AND whether they are one- or two-sided <br> *Only common tests should be described solely by name; describe more complex techniques in the Methods section.* |
| ☐ | ☒ | A description of all covariates tested |
| ☐ | ☒ | A description of any assumptions or corrections, such as tests of normality and adjustment for multiple comparisons |
| ☐ | ☒ | A full description of the statistical parameters including central tendency (e.g. means) or other basic estimates (e.g. regression coefficient) AND variation (e.g. standard deviation) or associated estimates of uncertainty (e.g. confidence intervals) |
| ☐ | ☒ | For null hypothesis testing, the test statistic (e.g. *F*, *t*, *r*) with confidence intervals, effect sizes, degrees of freedom and *P* value noted <br> *Give P values as exact values whenever suitable.* |
| ☒ | ☐ | For Bayesian analysis, information on the choice of priors and Markov chain Monte Carlo settings |
| ☒ | ☐ | For hierarchical and complex designs, identification of the appropriate level for tests and full reporting of outcomes |
| ☒ | ☐ | Estimates of effect sizes (e.g. Cohen's *d*, Pearson's *r*), indicating how they were calculated |

*Our web collection on statistics for biologists contains articles on many of the points above.*

## Software and code

Policy information about availability of computer code

| Data collection | No software was used. |
|---|---|
| Data analysis | Bedtools 2.27.0 <br> bcftools 1.9 <br> bwa 0.7.17 <br> coloc 4.0.4 <br> Eagle 2.4; 2.0.5 <br> EIGENSOFT 7.2.1 <br> EIGENSTRAT <br> Enhanced FastQTL 2.184_gtex <br> EstiMeth 1.1 <br> FlashPCA 2.0 <br> FOCUS 0.7 <br> GCTA-COJO from gcta 1.93.2beta <br> GENESIS 2.14.4 <br> GenomeStudio 2.0.3 <br> Gentrain 3.0 <br> ggplot2 3.3.3 <br> Guppy 3.2.4 <br> GWAF  2.2 <br> GWASTools 1.30.1 <br> HIBAG 1.4 <br> IGV 2.4.17 |

```
LDSC 1.0.1
Ldstore 2.0
Leafcutter 0.2.9
Liftover
MAGMA 1.08
METAL v2011-03-25
MetaXcan 0.6.12
Michigan Imputation Server 1.2.4
Minimac 3 and 4-1.0.2
minimap 2.17
mosdepth 0.2.9
NanoStat 1.1.2
NCBI remap
PBWT 3.1
Picard 2.22.6
PLINK 1.9 and 2.0
pygenometracks 3.5
qcat 1.0.1
R 3.6.0, 3.6.1 and 3.6.3
RegTools 0.5.1
RNASeQC 2.3.5
R package GenABEL 1.8-0
R package haplo.stats 1.8.6
R package ieugwasr 0.1.5
R package metafor 3.0.2
R package pec 2020.11.17
R package riskRegression 2020.12.8
R package stats 3.6.2
R package survIDINRI 1.1.1
R package survival 3.2.11
SAIGE  0.36.3.2 and 0.36.4
SamJdk 9750c96
Samtools 1.9
SeqMeta  1.6.7
SNPRelate 1.18.1
SNPTEST 2.5.3, 2.5.4-beta3 and 2.5.6
STAR 2.7.3a
STRING v11
vcffilterjdk v9750c96
```

For manuscripts utilizing custom algorithms or software that are central to the research but not yet described in published literature, software must be made available to editors and reviewers. We strongly encourage code deposition in a community repository (e.g. GitHub). See the Nature Research guidelines for submitting code & software for further information.

# Data

Policy information about availability of data

All manuscripts must include a data availability statement. This statement should provide the following information, where applicable:

- Accession codes, unique identifiers, or web links for publicly available datasets
- A list of figures that have associated raw data
- A description of any restrictions on data availability

Genome-wide summary statistics have been deposited to the European Bioinformatics Institute GWAS Catalog (https://www.ebi.ac.uk/gwas/) under accession no. GCST90027158.
The significant e/sQTLs mapped and e/sTWAS functional reference panel weights generated for this study (in AD-relevant bulk brain regions from AMP-AD cohorts and in LCLs from the EADB Belgian cohort) are publicly available at https://doi.org/10.5281/zenodo.5745927 and https://doi.org/10.5281/zenodo.5745929.
Anonymized aligned reads of the amplicon-based long-read nanopore cDNA sequencing experiment conducted for the TSPAN14 splicing analysis are available through ENA under accession PRJEB49234.
Moreover, the following data used in the gene prioritization are publicly available:
AMP-AD rnaSeqReprocessing Study: https://www.synapse.org/#!Synapse:syn9702085
MayoRNAseq WGS VCFs: https://www.synapse.org/#!Synapse:syn11724002
ROSMAP WGS VCFs: https://www.synapse.org/#!Synapse:syn11724057
MSBB WGS VCFs: https://www.synapse.org/#!Synapse:syn11723899
eQTLGen: https://www.eqtlgen.org/
eQTL Catalogue database: https://www.ebi.ac.uk/eqtl/
Brain xQTL serve: http://mostafavilab.stat.ubc.ca/xqtl/
GTEx v8 eQTL and sQTL catalogues: https://www.gtexportal.org/
GTEx v8 expression and splicing prediction models: http://predictdb.org/
MiGA eQTLs: https://doi.org/10.5281/zenodo.4118605
MiGA sQTLs: https://doi.org/10.5281/zenodo.4118403
MiGA Meta-analysis: https://doi.org/10.5281/zenodo.4118676
Wingo et al. pQTL data: https://www.synapse.org/#!Synapse:syn23627957

# Field-specific reporting

Please select the one below that is the best fit for your research. If you are not sure, read the appropriate sections before making your selection.

☒ Life sciences ☐ Behavioural & social sciences ☐ Ecological, evolutionary & environmental sciences

# Life sciences study design

All studies must disclose on these points even when the disclosure is negative.

| | |
|---|---|
| Sample size | Raw data used in this study was collected by the EADB consortia and summary statistics were recruited by external sources used for meta-analysis. Sample size was not pre-determined and was chosen based on all known available cohorts with relevant data collected to date, after quality control steps were performed in each cohort (described in detail in Supplementary Information) in particular to avoid any sample duplications. The sample size was calculated as the number of individuals summed across all studies in the meta-analysis, N=487,511. |
| Data exclusions | We exluded samples and variants based on standard quality control procedures for GWAS ( Samples: Heterozygosity and missingness, Population outliers, Sex-check, Relatedness, Possibly problematic chips batch; Variants: Missingness and Hardy-Weinberg equilibrium, Frequency checks, Ambiguous variants, Duplicated variants). Complete details of our quality control procedures are provided in the methods and supplementary information section of the manuscript. |
| Replication | The meta-analysis strategy includes replication by default, as it weights the reported test statistics by the evidence of association across multiple samples. Further, SNP-based replication was carried out for the top GWAS association signals in an independent sample (N= 25,392 Alzheimer's disease cases and 276,086 controls; see Methods and supplementary information). |
| Randomization | Samples were randomized by case and control status on plates during genotyping at their independent study sites. |
| Blinding | genotyping was done blind without knowing the status of the individuals. The analysts were not blinded to the status of the individuals because QC procedures require knowing case and control status. |

# Reporting for specific materials, systems and methods

We require information from authors about some types of materials, experimental systems and methods used in many studies. Here, indicate whether each material, system or method listed is relevant to your study. If you are not sure if a list item applies to your research, read the appropriate section before selecting a response.

## Materials & experimental systems

| n/a | Involved in the study |
|---|---|
| ☒ | ☐ Antibodies |
| ☒ | ☐ Eukaryotic cell lines |
| ☒ | ☐ Palaeontology and archaeology |
| ☒ | ☐ Animals and other organisms |
| ☐ | ☒ Human research participants |
| ☒ | ☐ Clinical data |
| ☒ | ☐ Dual use research of concern |

## Methods

| n/a | Involved in the study |
|---|---|
| ☒ | ☐ ChIP-seq |
| ☒ | ☐ Flow cytometry |
| ☒ | ☐ MRI-based neuroimaging |

# Human research participants

Policy information about studies involving human research participants

| | |
|---|---|
| Population characteristics | We used multiple independent sets of participants in this study. We adjusted the analysis for principal components. Sample sizes, age and gender characteristics for our sample can be found per cohort and overall in Supplementary Tables 1 and Supplementary Information. |
| Recruitment | Participants from case-control studies were primarily recruited from clinics, nursing homes, disease registries, and hospitals, with controls being drawn from various ongoing studies and screened to exclude dementia/cognitive decline (see description of the samples in the supplmentary information). Cases were recruited according to clincal diagnsosis and defined as probable AD cases with a potential risk of misdiagnosis (estimated between 10 and 20% in the litterature). Controls included in the study were free of cognitive decline but a large part of them did not have any follow-up wit the possiblity that they developed dementia years later.
The UK Biobank recruited adult volunteers from national health registration records. UK Biobank participants are healthier than the general population, but since the data used in this study referred to parental diagnoses, the impact of selection bias should be minor. |

Ethics oversight

Written informed consent was obtained from study participants or, for those with substantial cognitive impairment, from a caregiver, legal guardian, or other proxy, and the study protocols for all populations were reviewed and approved by the appropriate local Institutional review boards (see description of the samples in the supplementary information).

Note that full information on the approval of the study protocol must also be provided in the manuscript.

