## [Peer Review File. · Nature Genetics]

Peer Review Information

Manuscript Title: New insights into the genetic etiology of Alzheimer's disease and related dementia

Corresponding author name(s): Dr Jean-Charles Lambert

Editorial Notes: This manuscript has been previously reviewed at another journal. This document only contains reviewer comments, rebuttal and decision letters for versions considered at Nature Genetics.

Decision Letter, initial version:

6th Aug 2021

Dear Dr Lambert,

Your Article, "New insights on the genetic etiology of Alzheimer's and related dementia" has now been seen by 2 referees. You will see from their comments below that while they find your work of interest, some important points are raised. We are interested in the possibility of publishing your study in Nature Genetics, but would like to consider your response to these concerns in the form of a revised manuscript before we make a final decision on publication.

As you will see, both reviewers state the manuscript has improved and that many of the previous concerns have been addressed. However, neither is fully satisfied. Reviewer #1 thinks that you need to justify the gene prioritization approach used and improve the language. Reviewer #2 asks about brain regions and also mentions that the manuscript should be edited for clarity.

We therefore invite you to revise your manuscript taking into account all reviewer and editor comments. Please highlight all changes in the manuscript text file. At this stage we will need you to upload a copy of the manuscript in MS Word .docx or similar editable format.

*1) Include a "Response to referees" document detailing, point-by-point, how you addressed each

referee comment. If no action was taken to address a point, you must provide a compelling argument. This response will be sent back to the referees along with the revised manuscript.

*2) If you have not done so already please begin to revise your manuscript so that it conforms to our Article format instructions, available [here](http://www.nature.com/ng/authors/article_types/index.html). Refer also to any guidelines provided in this letter.

[REDACTED]

We hope to receive your revised manuscript within four to eight weeks. If you cannot send it within this time, please let us know.

Thank you very much.

All the best,

Catherine

Catherine Potenski, PhD
Chief Editor
Nature Genetics
1 NY Plaza, 47th Fl.
New York, NY 10004
catherine.potenski@us.nature.com
<https://orcid.org/0000-0002-4843-7071>

Referee expertise:

Referee #1: statistical genetics

Referee #2: aging/neurogenetics

Reviewers' Comments:

Reviewer #1:

Remarks to the Author:

The revised manuscript is a great deal stronger. I have a few remaining concerns.

First, I appreciate that the authors constructed a systematic way of prioritizing genes, but it is complex and involves a very large number of seemingly arbitrary weights. Is there any way to simplify it and/or justify the very many choices that went into the system? Supp Table 19 and Supp Fig 34 are clarifying, but it shouldn't be necessary to dig through the supplement to understand fundamentally what makes a gene a Tier 1 gene. I have two more specific concerns as well. First, do the authors have a justification for giving added points for the variant being low frequency? Is there a reference for the claim that non-coding rare variants are more likely than common variants to impact the closest gene? Second, I was concerned by this sentence, which didn't seem to match the description of the overall gene prioritization pipeline: "Of note, LILRB2 in locus 39 (L39) was the only gene that we upgraded from Tier 2 to Tier 1 based on strong bibliographic evidence (see Supplementary Information), compared to the other Tier 2 gene in this locus, i.e. MYADM." If bibliographic evidence is being used to choose genes then that would bias subsequent enrichment analyses and undermine claims of being systematic.

Also, I don't understand the authors' reluctance to visualize the characteristics of the prioritized genes. I don't think this is critical, but there is a lot of information in the discussion, and I think visual representation of all of the claims and hypotheses from the discussion would be an aid to the reader.

I also found that the writing remained unclear in places. For example, this sentence took me a long time to parse and I'm still only mostly sure I know what the authors are trying to say: "Even if these data may suggest a functional relationship between microglia and APP/A β pathways, this observation rather reinforces implication of endocytosis in microglia in AD, mechanism which is also strongly involved in the APP metabolism."

Reviewer #2:

Remarks to the Author:

This manuscript has been transferred to Nature Genetics after I provided a review at another journal.

Overall, the authors have addressed many of many of my concerns and have proven the novelty/significance of their findings. I disagree with them about the necessity of future GWAS focused on common variants, but appreciate their rebuttal and see room for disagreement. See below for specific commentary and feedback:

A) Summary of key results: Largely unchanged

B) Originality and significance: The novelty of this manuscript is enhanced in the revised manuscript

C) Data and methodology: The clarity of data sources and methodology is much improved in the revised manuscript

D) Appropriateness of statistics and treatment of uncertainties: Improved in the revised manuscript

E) Conclusions: Largely unchanged compared to the initially submitted manuscript

F) Suggested improvements:

The manuscript is substantially improved since last reviewed and I appreciate the authors' changes. I have a questions/suggestions to consider:

Major:

1) Supplemental table 16 and ABA analyses. The authors mention using brain from multiple brain regions, but AD has a distinct neuroanatomic signature that differentiates it from other forms of dementia. Does the microglial signal remain the only signal in areas more specific to AD (e.g. hippocampus, parietal lobe, entorhinal cortex, etc).

Minor:

1) The manuscript still has many typos and incomplete sentences. For instance: "Even if these data may suggest a functional relationship between microglia and APP/A β pathways, this observation rather reinforces implication of endocytosis in microglia in AD, mechanism which is also strongly involved in the APP metabolism." I find it very hard to follow this sentence and think it is missing words and could be rephrased as "...reinforces *the* implication of endocytosis in microglia in AD, *a* mechanism...". Close proofreading and revision would be required if published.

G) References: Updated and appropriate

H) Clarity and context: Improved

Author Rebuttal to Initial comments

Dear Dr. Potenski,

Thank you very much for considering a new revised version of our manuscript entitled « New insights into the genetic etiology of Alzheimer's disease and related dementias» in Nature Genetics.

We were very happy to read that the modifications we made to our manuscript following the constructive comments of the two reviewers were particularly appreciated by them. We thank them again for helping us to improve our work and for continuing to do so by indicating a few more points for discussion.

Importantly, we redid several of our PRS analyses excluding FHS because this cohort is part of CHARGE and therefore included in the stage II from which we took the ORs to compute the PRS. This might potentially lead to overfitting. Even if this particular issue was not raised by the two reviewers, we decided to perform sensitivity analyses by removing this study when necessary. Our results indicate that FHS does not drive any of our findings. These results are briefly mentioned in the main result section (line 684-685) and added in the supplementary Tables 36, 37, 39, 40, 42 and 43.

Referee #1.

The revised manuscript is a great deal stronger. I have a few remaining concerns. First, I appreciate that the authors constructed a systematic way of prioritizing genes, but it is complex and involves a very large number of seemingly arbitrary weights. Is there any way to simplify it and/or justify the very many choices that went into the system? Supp Table 19 and Supp Fig 34 are clarifying, but it shouldn't be necessary to dig through the supplement to understand fundamentally what makes a gene a Tier 1 gene.

We considered several ways to approach the systematic gene prioritization. First, we tested the methodology that the reviewer illustrated as an example, which consisted of grouping the genes into tiers by considering three sets of evidence, where a gene with hits in all three sets of evidence was classified as Tier 1, etc. However, due to relatively high number of significant associations in TWAS and/or QTL overlap analyses, this led to an inflation of prioritized genes, resulting in ~90 genes that were prioritized as Tier 1 or Tier 2, in addition to 51 genes with Tier 3 classification. This inflation was especially prominent in the complex loci with large LD blocks of common association signals spanning numerous genes. We therefore proceeded with an approach of gene prioritization in each locus separately, as indicated in Supp. Fig. 34. This extra step is necessary to account for the fact that loci may have different characteristics in terms of frequency and LD of the association signal and gene density and complexity.

Furthermore, we found that the prioritization scheme needed additional layers, because it involved levels of evidence from input data of diverse methodology, cell- and tissue types with varying strength of evidence and relevance, which could lead to bias in a simplified prioritization scheme. These additional layers are reflected in the scoring system, which we started by assigning the lowest weight to the most peripheral cell type and the weakest molecular evidence (lead variant - LCL sQTL overlap category: weight 0.5), and then built up from there, adhering to a predefined mathematical relationship with other classes of evidence, as also mentioned in the manuscript:

- a) Brain and microglia related associations are weighted 2X compared to others peripheral cell type related associations in LCL, monocyte, and macrophage

b) Weights for colocalization and TWAS are 2X higher than lead variant - QTL overlap in the same tissue/cell (and in the case of the availability of TWAS fine-mapping option for expression TWAS results, we give 2X higher weights for the fine-mapped associations, and 1.5X higher for the significant TWAS associations that were not in the fine-mapping credible set)

c) When data from multiple independent QTL catalogues are available, we add weight in case of replication, in which case we give more weight for replication in brain and microglia (+1) compared to LCL, monocyte, and macrophage (+0.5)

d) Evidence from PWAS and MetaMeth are also categorized and ranked accordingly

We adapted the idea of the reviewer to facilitate the interpretation of gene prioritization visually, in a way that the reader can understand what type of profile the prioritized gene has in terms of the assays and tissues/cells in which evidence was observed (and how strong) for its prioritization (Fig. 3A and Supp. Fig. 34). We do think that our methodology is ranking the available evidence for each gene based on each category and domain fairly, and when applied to each locus each gene is tested equally with the same type of post-GWAS analyses, without having bias from different locus characteristics. Overall, this allowed us to nominate candidate risk genes in each locus, that will need to be further investigated in future functional follow-up studies.

We agree with the reviewer that it would be helpful for the reader to include Supp. Fig. 34 as a figure in the main text. We thus incorporated this Figure in the main text (Figure 2) hoping that the editor will agree with this modification (now 8 items in the main manuscript).

I have two more specific concerns as well. First, do the authors have a justification for giving added points for the variant being low frequency? Is there a reference for the claim that non-coding rare variants are more likely than common variants to impact the closest gene?

As the molecular QTL studies typically test common (MAF > 1%) variants for molecular phenotype associations, for analysis of rare association signals they have limited utility for gene prioritization. For prioritization of such rare variant loci, the variation annotation domain is the decisive domain.

There are multiple studies that support our methodology to give extra weight for the closest gene when the lead variant is a rare variant. First, the frequently used gene-based rare variant association testing model, sequence kernel association test (SKAT¹) and its extension SKAT-O², recommend upweighting of the effect of rare variants (typically at MAF < 1%, and with beta weights of 1 for common and 25 for rare variants) based on simulation and real-data analyses. Second, another study showed that the candidate large-effect non-coding rare variants are enriched near transcription start site (TSS) of their cis-eQTL genes; and importantly, their effect sizes decrease as a function of distance to TSS³, that was also later demonstrated by other following studies conducted using GTEx data⁴⁻⁶. Moreover, when considering the high confidence lead variant - causal gene relationships in the GWAS Gold Standards repository (<https://github.com/opentargets/genetics-gold-standards>, curated by the Open Targets Platform), we found that the rare noncoding lead variants are more accurate in predicting the causal gene as the nearest annotated gene (72.7%), compared to common noncoding lead variants (62.6%). Of note, rare protein-altering variants were fully accurate to identify the high confidence causal genes, and this was true for 97.4% of the cases for the common protein-altering variant, an observation that also supported our considerably higher weight for protein-altering common and rare variants.

Taken together, we think that these references and observations are in line with our approach to give additional points for a noncoding rare association in the gene prioritization, compared to noncoding common association.

1. Wu, M. C. *et al.* Rare-variant association testing for sequencing data with the sequence kernel association test. *Am. J. Hum. Genet.* **89**, 82–93 (2011).
2. Lee, S. *et al.* Optimal unified approach for rare-variant association testing with application to small-sample case-control whole-exome sequencing studies. *Am. J. Hum. Genet.* **91**, 224–237 (2012).
3. Li, X. *et al.* Transcriptome sequencing of a large human family identifies the impact of rare noncoding variants. *Am. J. Hum. Genet.* **95**, 245–256 (2014).
4. Li, X. *et al.* The impact of rare variation on gene expression across tissues. *Nature* **550**, 239–243 (2017).
5. Ferraro, N. M. *et al.* Transcriptomic signatures across human tissues identify functional rare genetic variation. *Science (80-.)*. **369**, eaaz5900 (2020).
6. Li, J., Kong, N., Han, B. & Sul, J. H. Rare variants regulate expression of nearby individual genes in multiple tissues. *PLoS Genet.* **17**, 1–26 (2021).

Second, I was concerned by this sentence, which didn't seem to match the description of the overall gene prioritization pipeline: "Of note, LILRB2 in locus 39 (L39) was the only gene that we upgraded from Tier 2 to Tier 1 based on strong bibliographic evidence (see Supplementary Information), compared to the other Tier 2 gene in this locus, i.e. MYADM." If bibliographic evidence is being used to choose genes then that would bias subsequent enrichment analyses and undermine claims of being systematic.

We thank the reviewer for pointing this out and agree that this Tier upgrade based on literature evidence for one of the genes detracts from our systematic approach. In this revised version, we have changed the Tier status of *LILRB2* back to Tier 2, and have updated numbers of Tier 1 and 2 genes in abstract, main text (lines 576-578) and supplementary text (page 43 and 44). We have adapted Fig. 3A to reflect the change in Tier status for *LILRB2*. We have redone the STRING analyses (no major changes), which are based only on Tier 1 genes (lines 612-630 and supplementary Table 31). We also moved the (adapted) description of the locus in the Supplementary Results to the section on genes containing multiple Tier 2 genes (page 51). In the main text, we rephrased the sentence cited by the reviewer (lines 710-713).

Also, I don't understand the authors' reluctance to visualize the characteristics of the prioritized genes. I don't think this is critical, but there is a lot of information in the discussion, and I think visual representation of all of the claims and hypotheses from the discussion would be an aid to the reader.

The reviewer is correct in indicating that we are reluctant to provide a visualization of the novel genetic determinants of AD in a pathophysiological context. This is partly explained by the difficulty of privileging or not a particular function of a gene if the latter has pleiotropic properties. Even if the objective of a GWAS study is obviously to confirm/propose pathophysiological pathways involved in the disease, it is also not to bias the post-genomic studies which will be based on this work. This is all the truer since it took several years following the first GWAS works on AD for the statement of overarching

hypotheses based on our genetic knowledge. This was only possible following the publication of numerous studies deciphering the role of these genes in many biological contexts. Such a work is needed for the 42 new loci we have characterized.

Within this background, we decided to only focus in the discussion on obvious pathways such as the APP metabolism and the TNF- α signaling pathway mainly through the LUBAC genetic implication. Since the latter is a new finding and in order to take into account the reviewer's comment, we have added a supplementary figure describing the TNF- α complex, its regulation by LUBAC and the genetic risk factors involved (supplementary Fig. 47).

I also found that the writing remained unclear in places. For example, this sentence took me a long time to parse and I'm still only mostly sure I know what the authors are trying to say: "Even if these data may suggest a functional relationship between microglia and APP/A β pathways, this observation rather reinforces implication of endocytosis in microglia in AD, mechanism which is also strongly involved in the APP metabolism."

An English-native speaker completely edited the document when necessary to make it clearer.

Referee #2

Overall, the authors have addressed many of many of my concerns and have proven the novelty/significance of their findings. I disagree with them about the necessity of future GWAS focused on common variants, but appreciate their rebuttal and see room for disagreement. See below for specific commentary and feedback:

A) Summary of key results: Largely unchanged

B) Originality and significance: The novelty of this manuscript is enhanced in the revised manuscript

C) Data and methodology: The clarity of data sources and methodology is much improved in the revised manuscript

D) Appropriateness of statistics and treatment of uncertainties: Improved in the revised manuscript

E) Conclusions: Largely unchanged compared to the initially submitted manuscript

F) Suggested improvements:

The manuscript is substantially improved since last reviewed and I appreciate the authors' changes. I have a questions/suggestions to consider:

Major:

1) Supplemental table 16 and ABA analyses. The authors mention using brain from multiple brain regions, but AD has a distinct neuroanatomic signature that differentiates it from other forms of dementia. Does the microglial signal remain the only signal in areas more specific to AD (e.g. hippocampus, parietal lobe, entorhinal cortex, etc).

As requested by the reviewer, we performed analyses based on the different brain areas available in the Allen brain atlas dataset. Results were similar whatever the area studied and this observation has been added in the main result section (line 525) and in the supplementary Table 16.

Minor:

1) The manuscript still has many typos and incomplete sentences. For instance: "Even if these data may suggest a functional relationship between microglia and APP/A β pathways, this observation rather reinforces implication of endocytosis in microglia in AD, mechanism which is also strongly involved in the APP metabolism." I find it very hard to follow this sentence and think it is missing words and could be rephrased as "...reinforces *the* implication of endocytosis in microglia in AD, *a* mechanism...". Close proofreading and revision would be required if published.

As previously indicated, An English-native speaker completely edited the document when necessary to make it clearer.

We hope that our answers and modifications correspond to your expectations as well as those of reviewers.

Looking hearing forwards from you

Best regards

Jean-Charles Lambert on behalf of the EADB consortium

Inserm Research Director
EADB consortium coordinator
Inserm UMR1167, Lille, France

Decision Letter, first revision:

Our ref: NG-A57672R

13th Oct 2021

Dear Dr. Lambert,

Thank you for submitting your revised manuscript "New insights into the genetic etiology of Alzheimer's disease and related dementia" (NG-A57672R). It has now been seen by the original referees and their comments are below. The reviewers find that the paper has improved in revision, and therefore we'll be happy in principle to publish it in Nature Genetics, pending minor revisions to satisfy the referees' final requests and to comply with our editorial and formatting guidelines.

Thank you very much.

All the best,

Catherine

Catherine Potenski, PhD
Chief Editor
Nature Genetics
1 NY Plaza, 47th Fl.
New York, NY 10004
catherine.potenski@us.nature.com
<https://orcid.org/0000-0002-4843-7071>

Reviewer #1 (Remarks to the Author):

The authors have addressed my remaining concerns. In particular, I found that the analysis of rare and common closest-gene analysis in Open Targets was an important justification for the prioritization metric and I would encourage the authors to include this in the manuscript if they haven't already.

Reviewer #2 (Remarks to the Author):

The authors have addressed my concerns mentioned in the prior revision.

Author Rebuttal, first revision:

Dear Dr. Potenski,

We have finalized a new revised version of our manuscript entitled “**New insights into the genetic etiology of Alzheimer’s disease and related dementias**”.

As requested by reviewer 1, we have included the justification of the analysis of rare and common closest-gene analysis in Open Targets for the prioritization metric in the supplementary information.

Best regards

Jean-Charles Lambert on behalf of the EADB consortium

Inserm Research Director
EADB consortium coordinator
Inserm UMR1167, Lille, France

Final Decision Letter:

In reply please quote: NG-A57672R1 Lambert

27th Jan 2022

Dear Dr. Lambert,

I am delighted to say that your manuscript "New insights into the genetic etiology of Alzheimer’s disease and related dementia" has been accepted for publication in an upcoming issue of Nature Genetics.

Your paper will be published online after we receive your corrections and will appear in print in the next available issue. You can find out your date of online publication by contacting the Nature Press Office (press@nature.com) after sending your e-proof corrections. Now is the time to inform your Public Relations or Press Office about your paper, as they might be interested in promoting its publication. This will allow them time to prepare an accurate and satisfactory press release. Include your manuscript tracking number (NG-A57672R1) and the name of the journal, which they will need when they contact our Press Office.

Please note that *Nature Genetics* is a Transformative Journal (TJ). Authors may publish their research with us through the traditional subscription access route or make their paper immediately open access through payment of an article-processing charge (APC). Authors will not be required to make a final decision about access to their article until it has been accepted. [Find out more about Transformative Journals](https://www.springernature.com/gp/open-research/transformative-journals)

Authors may need to take specific actions to achieve [compliance](https://www.springernature.com/gp/open-research/funding/policy-compliance-faqs) with funder and institutional open access mandates. For submissions from January 2021, if your research is supported by a funder that requires immediate open access (e.g. according to [Plan S principles](https://www.springernature.com/gp/open-research/plan-s-compliance)) then you should select the gold OA route, and we will direct you to the compliant route where possible. For authors selecting the subscription publication route our standard licensing terms will need to be accepted, including our [self-archiving policies](https://www.springernature.com/gp/open-research/policies/journal-policies). Those standard licensing terms will supersede any other terms that the author or any third party may assert apply to any version of the manuscript.

Please note that Nature Research offers an immediate open access option only for papers that were first submitted after 1 January, 2021.

If you have any questions about our publishing options, costs, Open Access requirements, or our legal

forms, please contact ASJournals@springernature.com

If you have not already done so, we invite you to upload the step-by-step protocols used in this manuscript to the Protocols Exchange, part of our on-line web resource, natureprotocols.com. If you complete the upload by the time you receive your manuscript proofs, we can insert links in your article that lead directly to the protocol details. Your protocol will be made freely available upon publication of your paper. By participating in natureprotocols.com, you are enabling researchers to more readily reproduce or adapt the methodology you use. [Natureprotocols.com](http://natureprotocols.com) is fully searchable, providing your protocols and paper with increased utility and visibility. Please submit your protocol to <https://protocolexchange.researchsquare.com/>. After entering your [nature.com](http://www.nature.com) username and password you will need to enter your manuscript number (NG-A57672R1). Further information can be found at <https://www.nature.com/nprot/>.

Congratulations on the paper!

All the best,

Catherine

Catherine Potenski, PhD
Chief Editor
Nature Genetics
1 NY Plaza, 47th Fl.
New York, NY 10004

catherine.potenski@us.nature.com
<https://orcid.org/0000-0002-4843-7071>